# PlugMem: A Task-Agnostic Plugin Memory Module for LLM Agents

**Ke Yang** [* 1]   **Zixi Chen** [* 2]   **Xuan He** [* 1]   **Jize Jiang** [* 1]
**Michel Galley** [3]   **Chenglong Wang** [3]   **Jianfeng Gao** [3]   **Jiawei Han** [1]   **ChengXiang Zhai** [1]

## Abstract

Long-term memory is essential for large language model (LLM) agents operating in complex environments, yet existing memory designs are either task-specific and non-transferable, or task-agnostic but less effective due to low task-relevance and context explosion from raw memory retrieval. We propose PLUGMEM, a task-agnostic plugin memory module that can be attached to arbitrary LLM agents without task-specific redesign. Motivated by the fact that decision-relevant information is concentrated as abstract knowledge rather than raw experience, we draw on cognitive science to structure episodic memories into a compact, extensible knowledge-centric memory graph that explicitly represents propositional and prescriptive knowledge. This representation enables efficient memory retrieval and reasoning over task-relevant knowledge, rather than verbose raw trajectories, and departs from other graph-based methods like GraphRAG by treating *knowledge* as the unit of memory access and organization instead of entities or text chunks. We evaluate PLUGMEM unchanged across three heterogeneous benchmarks (long-horizon conversational question answering, multi-hop knowledge retrieval, and web agent tasks). The results show that PLUGMEM consistently outperforms task-agnostic baselines and exceeds task-specific memory designs, while also achieving the highest information density under a unified information-theoretic analysis. Code and data are available at https://github.com/TIMAN-group/PlugMem.

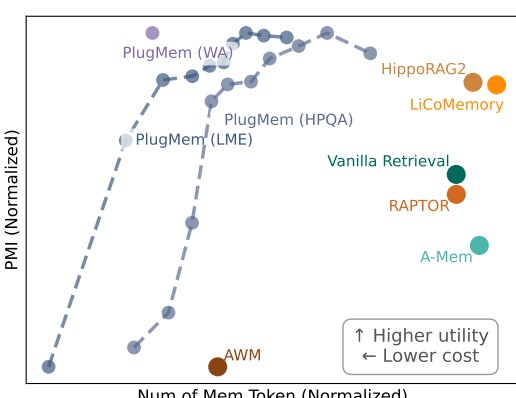

*Figure 1.* A utility–cost visualization of agentic memory approaches.[1] **PLUGMEM, evaluated unchanged across heterogeneous benchmarks requiring processing multiple memory types, achieves the highest decision-making utility of memory at the lowest agent-side memory cost.**

## 1. Introduction

Large language model (LLM) agents increasingly operate in settings that require long-term memory, where relevant information is distributed across long interaction histories and must be reused to support future decisions ([Wang et al., 2024a](); [Yang & Zhai, 2025]()). However, naively accumulating past interactions as raw context quickly leads to unbounded memory growth, high computational cost, and degraded performance ([Packer et al., 2024](); [Liu et al., 2023]()). To address this, many agent architectures rely on external memory modules to store and retrieve past experience. Yet most existing memory designs tightly couple memory representation and retrieval to specific tasks or benchmarks, relying on hand-crafted heuristics for what to store and how to use it ([Wang et al., 2024b](); [Gutiérrez et al., 2025]()). While effective in

---

[1]University of Illinois Urbana-Champaign [2]Tsinghua University [3]Microsoft Research. Correspondence to: Ke Yang <key4@illinois.edu>, ChengXiang Zhai <czhai@illinois.edu>.

*Proceedings of the $43^{rd}$ International Conference on Machine Learning*, Seoul, South Korea. PMLR 306, 2026. Copyright 2026 by the author(s).

[1]Each point corresponds to a memory method evaluated on

at least one of three benchmarks, i.e., LongMemEval ([Wu et al., 2024]()), HotpotQA ([Yang et al., 2018]()) and WebArena ([Zhou et al., 2024]()), plotted in a normalized utility–cost space to enable cross-task visualization and to accommodate baselines that are not applicable to all benchmarks. Both axes are min–max normalized, which linearly rescales values to a fixed range (e.g., [0, 1]) while preserving relative ordering across methods. Curves are obtained by sweeping the memory token budget. The normalization is applied solely for visualization; detailed benchmark-specific analyses using raw PMI values and token numbers are reported in Section 4 and Figure 5.

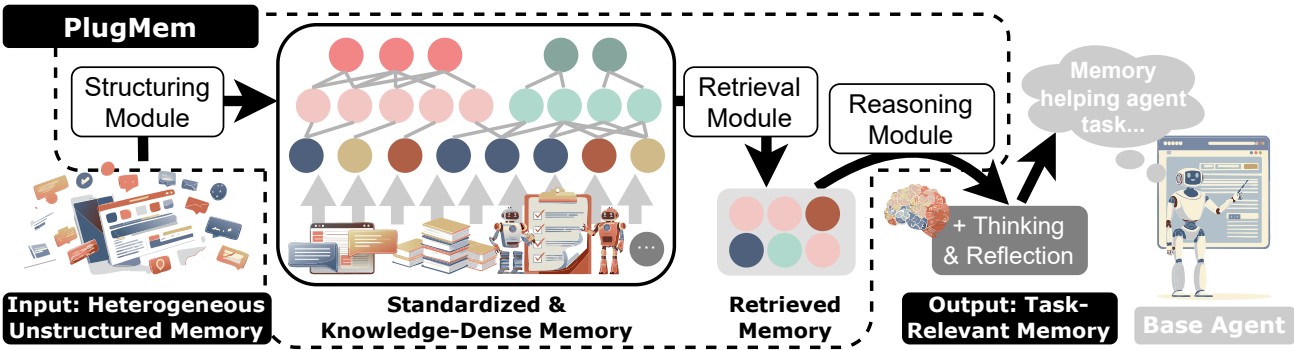

*Figure 2.* PLUGMEM organizes raw memory and outputs refined memory tokens to help the base agent's decision-making.

narrow settings, such task-specific memory modules fail to generalize: a memory system optimized for long-horizon conversations does not readily transfer to web navigation, and vice versa. This limitation motivates the need for a *task-agnostic plug-and-play* memory module that can be attached to arbitrary LLM agents without task-specific redesign, reducing per-task engineering overhead and enabling a single memory module to be reused across diverse agentic settings.

Designing such a general plugin memory module is challenging. The most straightforward task-agnostic approach is retrieval-based memory, where all past experiences are stored as raw text chunks and retrieved by relevance (optionally augmented with reasoning, as in RAG). However, this paradigm often fails in practice due to *knowledge sparsity*. Memories that are truly useful for decision-making are typically condensed and abstract forms of knowledge, whereas raw memories are verbose, episodic, and dominated by low-level information. For example, when recommending recipes, an agent benefits from knowing a user's dietary preferences and restrictions, which are compact factual propositions distilled from many interactions, rather than re-reading long conversation histories. Similarly, when shopping on a previously unseen web interface, an agent needs generalizable procedural knowledge, i.e., how to search, filter, and check out, not raw trajectories containing full-page observations with thousands of irrelevant tokens. Treating raw episodic memory as directly usable knowledge therefore imposes an unnecessary context burden and obscures the information most relevant for decision-making.

Prior work has attempted to bridge this gap by compressing or summarizing memory, but largely in task-specific ways. Memory modules are often engineered to perform well on a particular benchmark, such as conversational long-term memory or web-based agents, implicitly assuming a single dominant type of memory (Wang et al., 2024b; Gutiérrez et al., 2025). When applied to a different task domain, these designs rarely transfer without significant task-specific modifications. This raises a key challenge: how can we design a general-purpose memory module that simultaneously sup-

ports multiple memory types and adapts to the diverse demands of agentic tasks, and evaluate it in a way that jointly reflects decision utility and agent-side cost, while allowing cross-task comparability?

To address the design side of this challenge, we ground our approach in a principled account of memory organization from cognitive science. Decades of research suggest that the human brain makes a fundamental distinction between episodic memory (detailed records of experience) and knowledge-level memory, which can be further divided into semantic memory (knowing that; factual *propositions*) and procedural memory (knowing how; action-oriented *prescriptions*) (Tulving, 1972; Squire, 2004). Episodic memory serves as the source from which propositional and prescriptive knowledge are abstracted, while the latter forms are most directly useful for reasoning and decision-making. This perspective suggests that effective agent memory should not merely retrieve past experiences, but actively transform raw episodic memory into structured, knowledge-dense representations.

Based on these principles, we introduce a novel plugin memory module PLUGMEM, that performs *memory-to-knowledge abstraction* and supports the unified management of multiple key memory types across agentic tasks. As shown in Figure 2, our approach consists of: *i)* a *structuring module* that standardizes heterogeneous raw memories and extracts propositional and prescriptive knowledge through hierarchical abstraction, organizing them into a memory graph; *ii)* a *retrieval module* that selects task-relevant subgraphs; and *iii)* a *reasoning module* that further adapts and compresses retrieved knowledge for the base agent. Unlike conventional knowledge graphs that operate on entities and relations, our memory graph operates on *knowledge units* (i.e., propositions and prescriptions) which form the fundamental units of memory access and manipulation. PLUGMEM can be viewed as a knowledge-centric form of GraphRAG (Edge et al., 2025) tailored for memory management, where graph nodes are knowledge rather than entities or text chunks.

Complementary to the design of a general-purpose memory module, we contribute a novel utility-cost analysis framework that enables fair comparison between different memory designs across tasks, capturing both performance improvements and memory efficiency. Specifically, we measure the *information density* of memory, defined as the decision-relevant information gain provided to the base agent per memory token. We implement PLUGMEM and evaluate it on three heterogeneous and challenging benchmarks: long-horizon conversational question answering (Wu et al., 2024), multi-hop knowledge retrieval over Wikipedia (Yang et al., 2018), and web-based agent tasks (Zhou et al., 2024). *Using the same memory module implementation across all settings*, we demonstrate consistent performance gains over vanilla task-agnostic baselines and task-specific memory modules, while incurring lower agent-side memory cost, as illustrated in Figure 1. Ablation studies further clarify the roles of different components: First, since agentic memory ultimately serves decision-making, task performance is primarily determined by whether retrieval brings the most useful memory to bear at decision time. Second, our structuring module incrementally enhances retrieval by organizing heterogeneous experience into knowledge-centric memory units, yielding additional performance gains. Third, the reasoning module substantially improves efficiency, reducing memory token usage by one to two orders of magnitude through task-adaptive condensation. We provide illustrative benchmark examples in Appendix B.

In summary, our contributions are fourfold:

- **Design principles:** cognitively motivated principles for task-agnostic memory in LLM agents.
- **Evaluation framework:** an information-theoretic measure of memory utility and efficiency.
- **General memory module:** a plugin memory system applicable across heterogeneous agentic benchmarks.
- **Reproducibility:** released code and experimental results.

## 2. Related Work

**Cognitive Science Based Agent Memory**   Cognitive science characterizes long-term memory as persistent storage, and distinguishing episodic, semantic, and procedural memory (Atkinson & Shiffrin, 1968; Tulving, 1972; Squire, 2004). In agentic settings, interactions are naturally episodic, from which reusable abstractions can be derived, e.g., factual information such as user preferences can be distilled into semantic memory (Li et al., 2025), while action strategies can be abstracted as procedural memory (Wang et al., 2024b). These abstractions align with more general knowledge notions: propositional knowledge as factual statements and prescriptive knowledge as generalized goal-directed procedures (Mokyr, 2012).

Building on these insights, recent systems incorporate different memory types to support long-context understanding (Lee et al., 2024), long-term interactions (Li et al., 2025), structured memory organization (Anokhin et al., 2024; Rasmussen et al., 2025), and experience-driven strategy refinement (Zhao et al., 2024; Wang et al., 2024b; Fang et al., 2025). However, most approaches are task-specific and do not jointly support multiple memory types. In contrast, PLUGMEM standardizes heterogeneous episodic experience and organizes extracted knowledge into a propositional-prescriptive structure, enabling task-agnostic reuse.

*Table 1.* Comparison of representative agentic memory systems. **M2K** indicates if a system transforms working memories into *reusable, abstract knowledge* that's generalizable across tasks. **KaU** means if it adopts knowledge as the memory unit. **Mech.** characterizes the design of the memory modules, including **Str.** (structured memory representations), **Ret.** (refined retrieval strategies), and **Rea.** (post-retrieval reasoning or compression). **Mem.** denotes the memory type that the system supports, including **E**pisodic, **S**emantic, and **P**rocedural memory. **Agn.** indicates whether the memory design is task-agnostic.

| Meth. | M2K | KaU | Mech. | | | Mem. | | | Agn. |
|---|---|---|---|---|---|---|---|---|---|
| | | | Str. | Ret. | Rea. | E | S | P | |
| Vanilla Retrieval | ✗ | ✗ | ✗ | ✗ | ✓ | ✓ | ✗ | ✗ | ✓ |
| Vanilla RAG | ✗ | ✗ | ✗ | ✗ | ✗ | ✓ | ✗ | ✗ | ✓ |
| GraphRAG | ✓ | ✗ | ✓ | ✓ | ✓ | ✗ | ✓ | ✗ | ✓ |
| A-Mem | ✓ | ✓ | ✓ | ✗ | ✗ | ✓ | ✓ | ✗ | ✓ |
| Zep | ✗ | ✗ | ✓ | ✓ | ✗ | ✓ | ✓ | ✗ | ✗ |
| MemoryOS | ✗ | ✗ | ✗ | ✓ | ✗ | ✓ | ✓ | ✗ | ✗ |
| HippoRAG2 | ✓ | ✗ | ✓ | ✓ | ✓ | ✓ | ✓ | ✗ | ✗ |
| AWM | ✓ | ✓ | ✗ | ✗ | ✗ | ✗ | ✗ | ✓ | ✗ |
| ReasoningBank | ✓ | ✓ | ✗ | ✗ | ✗ | ✓ | ✗ | ✓ | ✗ |
| **PLUGMEM** | ✓ | ✓ | ✓ | ✓ | ✓ | ✓ | ✓ | ✓ | ✓ |

**Memory Module Designs**   Many agent memory systems adopt retrieval-based external memory, storing past interactions and retrieving them at inference time. Early task-agnostic methods rely on flat retrieval over unstructured episodic memory, such as vanilla retrieval and RAG (Lewis et al., 2021; Wang et al., 2024a), resulting in redundant and episode-specific memory reuse. Later work introduces structure over episodic memory, including hierarchical and graph-based retrieval (e.g., GraphRAG) to support multi-hop reasoning (Edge et al., 2025; Sarthi et al., 2024; Gutiérrez et al., 2025; Wang & Han, 2025). While improving access efficiency, these methods largely preserve episodic traces as the primary memory unit. In parallel, task-specific systems explicitly transform experience into higher-level representations, such as temporal knowledge graphs or workflow memories (Gutiérrez et al., 2025; Wang et al., 2024b; Ouyang et al., 2025). Although effective within fixed tasks, their abstractions are tightly coupled to task assumptions and limit transferability.

Overall, as shown in Table 1, prior work highlights a distinction between improving retrieval over episodic memory and transforming experience into reusable knowledge, with only

the latter supporting cross-task generalization. PLUGMEM adopts this principle by organizing memory around propositional and prescriptive knowledge units, enabling more effective retrieval of decision-relevant information, while retaining episodic traces as verifiable evidence.

For a more detailed discussion of related work, including the full versions of the above sections and additional discussion on memory benchmarks, see Appendix C.

# 3. Methodology

We present PLUGMEM, a task-agnostic plugin memory module to support long-term decision-making for LLM agents. Rather than treating past interactions as flat episodic text, PLUGMEM structures experience into knowledge-level representations that are more compact, generalizable, and directly relevant to downstream retrieval and reasoning.

As shown in Figure 2, PLUGMEM consists of three core components: *a structuring module* that standardizes heterogeneous episodic memories and induces propositional and prescriptive knowledge; *a retrieval module* that selects relevant knowledge from structured memory graphs via abstraction-aware retrieval; and *a reasoning module* that further adapts retrieved knowledge into actionable guidance for the base agent. In the following sections, we describe each component in detail.

## 3.1. Structuring Module

As shown in Figure 3, the structuring module serves as the foundation of PLUGMEM by transforming raw episodic experience, such as dialogue turns, document snippets, or episodic trajectories, into knowledge representations that are reusable, compact, and aligned with agent decision-making. Specifically, we structure memory to reflect the role different information plays in reasoning and action selection.

Our design is guided by three principles motivated by cognitive theories of human memory. First, episodic memory captures concrete interaction traces and serves primarily as verifiable evidence rather than directly actionable knowledge. Second, decision-relevant information is most effectively represented at the knowledge level, where semantic memory encodes factual propositions ("knowing that") and procedural memory encodes reusable strategies ("knowing how"). Third, effective long-term memory requires separating knowledge abstraction from task-specific execution details, enabling memory to generalize across heterogeneous environments. These principles imply that different memory types should be represented using structural units and organization logics aligned with their properties. Table 2 summarizes how episodic, semantic, and procedural memories are mapped to corresponding graph units and structuring mechanisms, reflecting their functional roles in abstraction,

retrieval, and verification.

Building on this design, the structuring module operationalizes memory abstraction in two stages: *i)* standardizing heterogeneous interaction traces into a unified episodic representation, and *ii)* inducing propositional and prescriptive knowledge that can be independently indexed and reused across tasks. We describe each stage as follows.

### 3.1.1. STANDARDIZE

Episodic memory constitutes the fundamental substrate from which semantic and procedural memories are derived (Tulving, 1972). For agents, episodic memories originate from heterogeneous sources, including user–agent interactions (Wu et al., 2024), factual documents (Yang et al., 2018), and action trajectories in complex environments (Zhou et al., 2024). This heterogeneity motivates a *unified, task-agnostic representation* that can support downstream knowledge induction.

**Episodic Formalization.** We represent a raw interaction trace as a sequence of observation-action pairs:

$$\tau = [(o_t, a_t)]_{t=1}^{T}$$

While episodic memory is widely used in agent systems, its internal structure is often treated as unstructured text. In contrast, we explicitly formalize episodic memory at the step level by mapping each interaction into a structured tuple. Specifically, each pair $(o_t, a_t)$ is standardized as:

$$e_t = (o_t, s_t, a_t, r_t, g_t)$$

where $s_t$ denotes the agent state at time $t$, $g_t$ denotes the subgoal associated with executing $a_t$, and $r_t$ denotes the reward of the action with respect to $g_t$.

The state $s_t$ is derived from $(s_{t-1}, a_{t-1}, o_t)$ via LLM-based information extraction. Both $g_t$ and $r_t$ are annotated by an LLM conditioned on the task instruction and local interaction context. Aggregating all standardized steps yields an episodic memory sequence

$$M_{\text{epi}} = [e_t]_{t=1}^{T}$$

Implementation details of episodic standardization, including the prompt template, are provided in Appendix D.1.

### 3.1.2. EXTRACT KNOWLEDGE

We focus this section on the design of the knowledge extraction and organization process. Implementation details, including model choices, parameter settings, and prompt configurations, are deferred to Appendix D.2.

Given standardized episodic memory $M_{\text{epi}}$, we induce two complementary forms of long-term memory: semantic

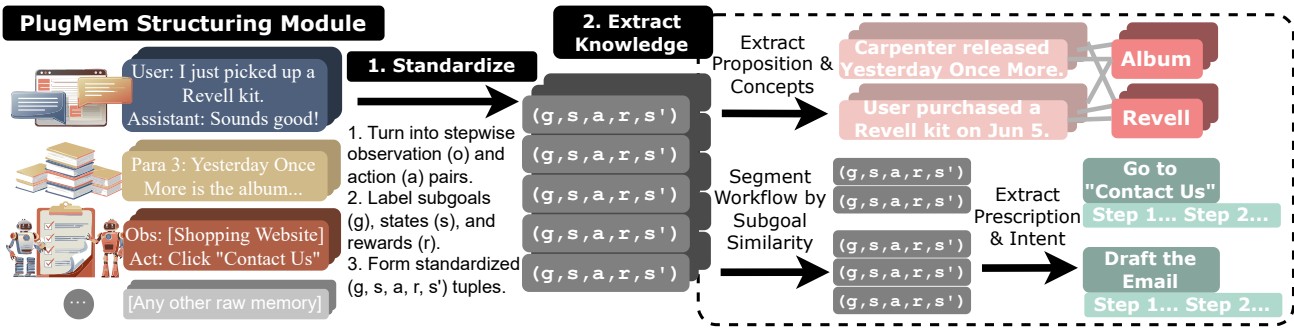

*Figure 3.* The structuring module in PLUGMEM transforms heterogeneous memory into a formalized knowledge-dense memory graph.

*Table 2.* **Mapping Memory Properties to Graph Architecture.** We analyze long-term memory properties to derive our memory graph design. The main rationale is **aligning memory structure with the granularity at which knowledge is acquired, stored, and reasoned over**, reflected here as ensuring core nodes represent complete, self-contained, and verifiable **knowledge blocks** (Propositions/Prescriptions), thus improving the efficiency and fidelity of downstream graph operations.

| Memory Type | Nature & Description | Knowledge-Dense Unit | Subgraph Structuring Logic |
|---|---|---|---|
| **Semantic** | **Declarative & Static** Context-independent concepts, facts, and world knowledge. | **The Proposition (Fact Block)** A complete statement conveying verifiable truth. | **Concept-Centric Semantic Mem Structuring** *Logic:* Concepts acts as lightweight indices pointing to heavy Proposition payloads. *Structure:* Concept $\xleftarrow{mentions}$ Proposition |
| **Procedural** | **Executive & Goal-Oriented** Dynamic "how-to" knowledge for problem solving. | **The Prescription (Workflow Block)** A full action sequence to execute a complex task. | **Intent-Centric Procedural Mem Structuring** *Logic:* Intents (User Goals) serve as keys to find holistic Solution blocks. *Structure:* Intent $\xleftarrow{solves}$ Prescription |
| **Episodic** | **Autobiographical & Linear** Raw record of past interactions and observations. Large volume. | **The Source Trace (Event Window)** A trajectory segment for grounding and verification. | **Episodic Mem as the Anchor** *Logic:* Episodic acts as the "ground truth" layer validating the abstract knowledge graphs. *Structure:* Knowledge $\xleftarrow{proves}$ Source |

memory and procedural memory. Both are extracted from episodic experience and organized as structured memory graphs with provenance.

**Semantic Memory.** The semantic memory module extracts and stores factual knowledge from episodic memory to support later retrieval. Given an episodic unit $e_t$, the module uses an LLM to extract a set of atomic propositions that describe salient facts implied by the interaction. Each proposition is accompanied by a set of associated concepts, which serve as semantic tags for indexing. For example, a proposition may be: *"Tam Sventon, known in Swedish as Ture Sventon, is a fictional private detective based in Stockholm."* The associated concept set is {*Tam Sventon, fictional private detective, Stockholm*}. To ensure extraction quality, we apply several constraints during LLM extraction, including coreference resolution, proposition deduplication, and length control.

The extracted propositions and concepts are stored in a semantic graph $G^S$. Each proposition and concept is instantiated as a node with a cached dense embedding. Two types of edges are constructed: *i) membership* edges linking proposi-

tions to their associated concepts, and *ii) provenance* edges linking propositions to their source episodic units in the episodic graph $G^E$. This design allows retrieved semantic knowledge to be traced back to its originating experience.

**Procedural Memory.** The procedural memory module extracts reusable action strategies from episodic trajectories to support future decision making. Given an episodic sequence $M_{\text{epi}}$, the module first segments the trajectory into coherent sub-trajectories by detecting boundaries where the similarity between adjacent subgoals $g_{t-1}$ and $g_t$ falls below a predefined threshold. For each trajectory segment, the module uses an LLM to induce a compact *(intent, prescription)* pair. The intent represents the objective pursued within the segment, while the prescription specifies an environment-agnostic action workflow that captures the key steps and cause-effect patterns required for successful execution. An example prescription is: *"To identify the lowest price of an item, search for the item using the search bar, sort the results by price, and verify the minimum across variants."* To enable quality-aware reuse, each induced prescription is assigned a scalar *return* score. The score is obtained using an LLM-based evaluator that assesses whether the intent is

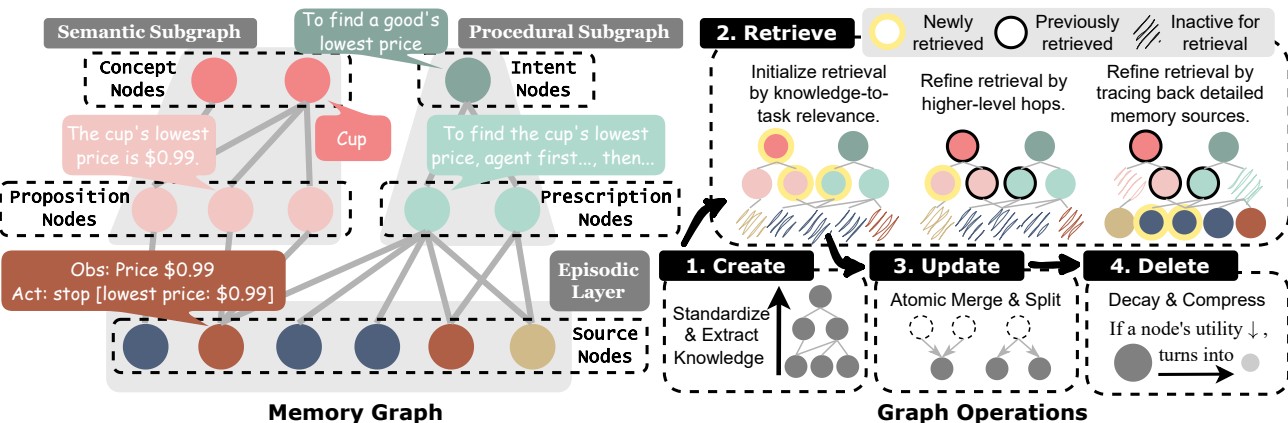

*Figure 4.* PLUGMEM's knowledge-centric memory graph design and the standard graph operations it supports.

achieved and how well the prescription is executed.

The extracted intents and prescriptions are stored in a procedural memory graph $G^P$. Each intent and prescription is instantiated as a node with a cached dense embedding. Edges in $G^P$ encode two types of relations. First, *hierarchical* edges link each high-level intent node to its associated low-level prescription nodes. Second, *provenance* edges link prescription nodes to their originating episodic units in the episodic graph $G^E$, enabling procedural knowledge to be traced back to concrete interaction experience.

### 3.2. Retrieval Module

This section describes the high-level retrieval process over semantic and procedural memory graphs. Detailed prompt templates, the step-by-step retrieval algorithm, and more technical details are deferred to Appendix D.3.

In the structuring stage, PLUGMEM constructs three inter-linked memory graphs: an episodic graph $G^E$, a semantic graph $G^S$, and a procedural graph $G^P$. Both $G^S$ and $G^P$ maintain explicit provenance links to $G^E$, enabling verifiable grounding of retrieved knowledge and experience. Figure 4 illustrates the overall memory organization.

Given a task description or query $Q$, an LLM-based retriever first determines which memory types to emphasize: *episodic*, *semantic*, or *procedural*. Retrieval primarily operates over $G^S$ and $G^P$ using an abstraction-specificity interleaving strategy. When episodic memory is prioritized, the same retrieval process (as will be introduced below) is applied, but the system ultimately returns provenance-linked episodic nodes in $G^E$.

Retrieval begins by encoding $Q$ into an embedding $q$ and scoring it against all low-level nodes (i.e., proposition or prescription nodes) to initialize a candidate set $C_0$. At hop $t$, the retriever conditions on $(Q, C_t)$ to generate an abstract query $q_t^a$. For $G^S$, $q_t^a$ is represented as a set of concepts,

while for $G^P$ it is represented as a set of intents.

The abstract query $q_t^a$ is matched against high-level (i.e., concept or intent nodes) nodes, which act as routing signals to activate adjacent low-level nodes that are added to $C_{t+1}$. Only low-level nodes are retained as candidates, while high-level nodes serve exclusively as intermediate traversal signals. When $|C_t|$ exceeds a predefined budget (e.g., top-$K$), candidates are re-ranked and pruned based on relevance and importance. This multi-hop retrieval process iterates until sufficient evidence is accumulated or a maximum hop limit is reached.

### 3.3. Reasoning Module

The reasoning module is a *test-time running module* that transforms retrieved memory into immediately actionable guidance for the playing agent. In many cases, retrieved memory may contain multiple overlapping or verbose descriptions of past interactions that are individually relevant but collectively redundant for the current decision. The reasoning module leverages the LLM to aggregate and condense such information into a compact, task-aligned representation, distilling the shared signal across messages into a single actionable summary. More technical details are included in Appendix D.4.

### 3.4. Summary and Supported Operations

As shown in Figure 3 and 4, starting from raw agent interactions, PLUGMEM standardizes episodic memory, extracts semantic and procedural knowledge, organizes them into structured memory graphs, and enables retrieval and reasoning over stored experience to support downstream decision making. At the system level, PLUGMEM supports a set of basic memory graph operations, including: *i) create*, which inserts newly observed episodic experience into structured memory, *ii) retrieve*, which retrieves relevant semantic, procedural, or episodic memory given a task or query, *iii)*

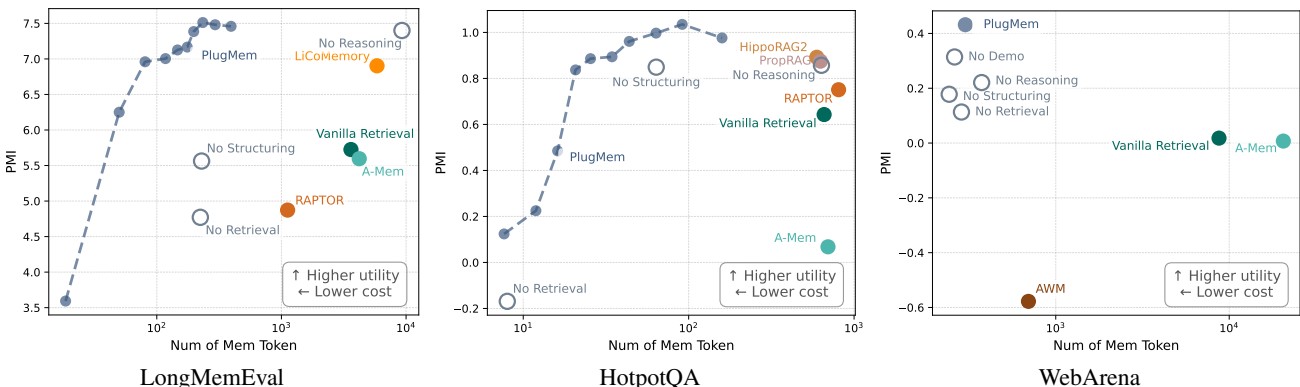

| LongMemEval | HotpotQA | WebArena |

*Figure 5.* Utility–cost analysis across benchmarks. Each point represents a memory method, with the x-axis indicating agent-side memory cost (in tokens) and the y-axis indicating decision-relevant utility (in bits). The slope of the line connecting a point to the origin corresponds to information density (bit per token). Curves are obtained by sweeping the memory token budget on a randomly sampled subset of benchmark tasks, illustrating how memory utility initially increases with budget, then saturates, and may eventually decline as additional memory becomes counterproductive, for example by introducing noise or interference in decision-making. **PLUGMEM consistently achieves a more favorable utility–cost trade-off, dominating prior approaches by providing higher decision-relevant utility under smaller memory budgets across benchmarks.**

*update*, which revises existing memory entries when new evidence becomes available, and *iv) delete*, which removes obsolete or low-utility memory.

The benchmark evaluations in the main paper primarily evaluate the *create* and *retrieve* operations. Additional experiments evaluating the effectiveness of the *update* and *delete* operations are provided in Appendix D.5.

## 4. Experiments

### 4.1. Evaluation Framework

We evaluate PLUGMEM using standard benchmark-wise metrics (e.g., accuracy, F1-score, success rate, etc.) to measure end-task performance. However, such metrics alone are insufficient for evaluating agentic memory, as they fail to capture the trade-off between decision-relevant utility and agent-side cost. We therefore propose an information-theoretic measure that quantifies the *decision-relevant information gain per memory token* contributed by the memory module. Specifically, for each decision instance with state $s$ and gold optimal action $a^*$, let the base agent's prior belief be $P_{\text{base}}(a^* \mid s)$ and the memory-augmented posterior be $P_{\text{mem}}(a^* \mid s, m)$ after consuming memory $m$. We define the *Decision Information Gain* as point-wise mutual information (PMI):

$$\text{PMI}(a^*; m \mid s) = \log_2 \frac{P_{\text{mem}}(a^* \mid s, m)}{P_{\text{base}}(a^* \mid s)} \tag{1}$$

and normalize by memory length $|m|$ (in tokens) to obtain *Memory Information Density* (bits / token):

$$\rho(a^*, m) = \frac{\text{PMI}(a^*; m \mid s)}{|m|} \tag{2}$$

Over a dataset, we report a global, amortized density via a ratio-of-sums:

$$\rho_{\text{global}} = \frac{\sum_i \text{PMI}(a_i^*; m_i \mid s_i)}{\sum_i |m_i|} \tag{3}$$

Measured in bits per token, our metric is task-agnostic and thus comparable across tasks. Cross-task variation in its magnitude reflects a utility–cost trade-off: higher density arises when memory yields larger decision-relevant gains or does so with fewer tokens, while lower density occurs when the base agent already solves the task well or when useful memory must be expressed verbosely. Appendix E details the complete analysis framework and additional components beyond the main-text description.

### 4.2. Common Experimental Setup

We evaluate PLUGMEM unchanged across three heterogeneous benchmarks that stress different aspects of agentic memory: *i)* LongMemEval (Wu et al., 2024) for long-horizon conversational memory, *ii)* HotpotQA (Yang et al., 2018) for multi-hop knowledge retrieval and reasoning, and *iii)* WebArena (Zhou et al., 2024) for interactive web-based decision-making. Across all benchmarks, we adopt a unified memory evaluation protocol to ensure fair comparison. Baselines are grouped into three categories: *i)* Vanilla, which do not rely on external memory; *ii)* Task-agnostic, which employ generic retrieval or agentic memory mechanisms not tailored to the benchmark; and *iii)* Task-specific, which incorporate benchmark-specific memory representations or retrieval heuristics.

Detailed experiment settings, prompt templates, and benchmark-level analysis are provided in Appendix F.

*Table 3.* Results on LongMemEval. #Tok$_{Avg.}$ is the average length of memory tokens. Experiments use NV-Embed-v2 (abbreviated as NVE) as the embedding model for retrieval, and Qwen2.5-32B (Q32) / gpt-4o (4o) as base LLMs for structuring and reasoning. * denotes results taken from previous work. † denotes methods evaluated on a subset of the full benchmark. Best is **bolded**.

| Method | Emb | LLM | Acc. | #Tok$_{Avg.}$ | Info. Density |
|---|---|---|---|---|---|
| Vanilla Baseline | | | | | |
| No Context | - | 4o | 8.8 | - | - |
| All Context | - | 4o | 63.0 | 115K | 4.8e-5 |
| Task-Agnostic | | | | | |
| Vanilla Retrieval | NVE | 4o | 65.2 | 3612.6 | 1.6e-3 |
| A-Mem† | NVE | 4o | 60.4 | 4211.4 | 1.3e-3 |
| Task-Specific | | | | | |
| RAPTOR† | NVE | 4o | 53.0 | 1042.2 | 4.7e-3 |
| Zep* | BGE-m3 | 4o | 71.2 | 1600 | - |
| LiCoMemory† | NVE | 4o | 74.1 | 5846.0 | 1.2e-3 |
| Ours | | | | | |
| PLUGMEM | NVE | Q32 + 4o | **82.8** | **233.9** | **3.2e-2** |

*Note.* More abbreviations: Llama3.3-70B (L70).

*Table 4.* Results on HotpotQA. EM means Exact Match. * denotes results taken from previous work. We underline the upper bound performance and **bold** the best.

| Mem. Method | Emb | LLM | EM | F1 | #Tok$_{Avg.}$ | Info. Density |
|---|---|---|---|---|---|---|
| Vanilla Baseline | | | | | | |
| No Context | - | Q32 | 22.1 | 31.0 | - | - |
| Gold Context | - | Q32 | 69.2 | 82.1 | 86.5 | 1.6e-1 |
| Task-Agnostic | | | | | | |
| Vanilla Retrieval | NVE | Q32 | 51.7 | 62.7 | 659.2 | 1.2e-2 |
| A-Mem | NVE | Q32 | 43.8 | 53.6 | 695.6 | 1.2e-2 |
| Task-Specific | | | | | | |
| GraphRAG* | NVE | L70 | 55.2 | 68.6 | - | - |
| RAPTOR | NVE | Q32 | 56.7 | 69.7 | 806.3 | 1.1e-2 |
| PropRAG | NVE | Q32 | 57.8 | 72.1 | 626.1 | 1.9e-2 |
| HippoRAG2 | NVE | Q32 | 60.0 | 73.3 | 595.1 | 1.9e-2 |
| Ours | | | | | | |
| PLUGMEM | NVE | Q32 | **61.4** | **74.1** | **81.6** | **1.4e-1** |

*Table 5.* Results on WebArena. SR means Success Rate. Each site-domain is split into (online/offline) sets. Shopping contains (38/149) tasks, GitLab contains (37/143), and Multi-set contains (10/38). * denotes results taken from previous work. AWM does not natively support Multi-site tasks. Best SR is **bolded**. Negative Info. Density indicates memory hurts performance.

| | | | SR % (on/off) | | | | |
|---|---|---|---|---|---|---|---|
| Method | Emb | Agent | Shopping | GitLab | Multi-site | #Tok$_{Avg.}$ | Info. Density |
| Vanilla Baseline | | | | | | | |
| AgentOccam* | - | 4o | 42.1/43.6 | 37.8/39.2 | **20.0**/15.8 | - | - |
| Task-Agnostic | | | | | | | |
| Van. Retrieval | NVE | Q32+4o | 43.0 /42.3 | 40.5/41.3 | 10.0/18.4 | 8733 | 2.0e-6 |
| A-Mem | NVE | Q32+4o | 44.7/44.3 | 37.8/38.5 | **20.0**/15.8 | 20516 | 3.4e-7 |
| Task-Specific | | | | | | | |
| AWM | - | 4o | 26.3/28.2 | 27.0/27.3 | - | 696 | -7.9e-4 |
| Ours | | | | | | | |
| PLUGMEM | NVE | Q32+4o | **52.6/58.4** | **51.4/55.2** | **20.0/21.6** | **301** | **1.4e-3** |

### 4.3. RQ1: Does PLUGMEM Improve Performance and Memory Efficiency Across Tasks?

Our first research question examines whether a single, task-agnostic memory module can consistently improve agent performance while reducing memory consumption across heterogeneous tasks. Results on LongMemEval, HotpotQA, and WebArena (Tables 3, 4, 5) show a consistent pattern despite large differences in task structure and interaction modality. First, PLUGMEM improves end-task performance over both task-agnostic and task-specific baselines. Second,

these gains are achieved with substantially fewer memory tokens injected into the agent context. Thus, PLUGMEM attains the highest *information-gain density* under the unified information-theoretic analysis introduced in Section 4.1. This trade-off is further illustrated by the utility–cost visualization in Figure 5, where PLUGMEM consistently shifts toward higher utility and lower agent-side cost across all three benchmarks.

The results indicate that PLUGMEM retrieves *more decision-relevant memory*. By abstracting raw experience into compact propositional and prescriptive knowledge, the memory module provides higher utility per token, enabling the base agent to reason more effectively under tight context budgets.

### 4.4. RQ2: What Is the Role of Each Component in PLUGMEM?

We next analyze the contribution of each component in PLUGMEM via ablations on all benchmarks (Tables 6, 7, 8). Removing retrieval leads to the most severe performance degradation across tasks, underscoring that memory is *only* useful when relevant experience can be accessed at decision time. However, this does not imply that retrieval alone drives performance gains. Rather, retrieval determines whether memory is operative at all, while its effectiveness is bounded by how memory is represented. The structuring module improves retrieval by organizing memory at appropriate abstraction levels, allowing the retriever to more effectively identify and access task-relevant knowledge. The reasoning module plays a complementary role, primarily affecting memory efficiency by controlling how retrieved knowledge is compressed and consumed.

Overall, retrieval determines *whether* memory helps, structuring determines *what* can be retrieved, and reasoning determines *how efficiently* retrieved memory can be used. Retrieval thus constitutes the defining bottleneck, while structuring and reasoning modulate effectiveness and efficiency once retrieval is in place.

### 4.5. RQ3: Knowledge Transfer and Memory Reuse

Our third research question evaluates whether agent memory can support transferable knowledge that generalizes across task instantiations and environments. To this end, we design a specialized evaluation protocol on WebArena: We focus on the Shopping, GitLab, and Multi-site subsets. Shopping and GitLab are procedure-heavy domains with large task volumes and relatively low baseline success rates (Yang et al., 2025b), avoiding saturation effects. The Multi-site subset further requires compositional skills across multiple websites, making it particularly challenging and well-suited for evaluating cross-task knowledge reuse.

To explicitly test memory evolution and reuse, we split tasks

*Table 6.* Ablation study on LongMemEval.

| Method | Emb | LLM | Acc. | #Tok$_{Avg.}$ | Info. Density |
|---|---|---|---|---|---|
| PLUGMEM | NVE | Q32 + 4o | 82.8 | 233.9 | 3.2e-2 |
| No Structuring | NVE | Q32 + 4o | 63.6 | 228.8 | 2.4e-2 |
| No Retrieval | - | Q32 + 4o | 55.8 | 223.6 | 2.1e-2 |
| No Reasoning | NVE | Q32 | 81.6 | 9267.9 | 8.0e-4 |

*Table 7.* Ablation study on HotpotQA.

| Method | Emb | LLM | EM | F1 | #Tok$_{Avg.}$ | Info. Density |
|---|---|---|---|---|---|---|
| PLUGMEM | NVE | Q32 | 61.4 | 74.1 | 81.6 | 1.4e-1 |
| No Structuring | NVE | Q32 | 51.4 | 62.0 | 116.7 | 6.8e-2 |
| No Retrieval | - | Q32 | 20.0 | 24.3 | 8.0[1] | -3.8e-1 |
| No Reasoning | NVE | Q32 | 59.3 | 71.8 | 635.1 | 1.7e-2 |

[1] In No Retrieval, we randomly sample corpus items to fit the reasoning module's context window. The sampled items are often irrelevant, so the reasoning module outputs little to no distilled context, yielding a much smaller #Tok$_{Avg.}$ (i.e., 8).

*Table 8.* Ablation Study on WebArena. No Human Demo means no human demonstrations are inserted into the memory graph between online and offline evaluation. We collect 23/18/5 demos for Shopping/GitLab/Multi-site.

| Method | Emb | Agent | SR % (on/off) | | | #Tok$_{Avg.}$ | Info. Density |
|---|---|---|---|---|---|---|---|
| | | | Shopping | GitLab | Multi-site | | |
| PLUGMEM | NVE | Q32+4o | 52.6/58.4 | 51.4/55.2 | 20.0/21.6 | 301 | 1.4e-3 |
| No Structuring | NVE | Q32+4o | 50.0/51.7 | 41.7/42.0 | 20.0/18.4 | 243 | 7.2e-4 |
| No Retrieval | NVE | Q32+4o | 42.1/46.3 | 45.8/44.0 | 20.0/15.8 | 286 | 3.8e-4 |
| No Reasoning | NVE | Q32+4o | 52.6/53.7 | 41.7/43.4 | 20.0/18.4 | 374 | 5.6e-4 |
| No Human Demo | NVE | Q32+4o | 52.6 /52.3 | 51.4 /51.0 | 20.0 /18.4 | 261 | 1.2e-3 |

into an *online* set and an *offline* set. For each WebArena's template, one instantiation is assigned to the online set, while the remaining instantiations form the offline set. The agent is first evaluated on the online set, during which PLUGMEM is allowed to insert memory. We then augment the memory module with a small number of high-quality human demonstrations, representing external sources of procedural knowledge analogous to tutorials or experience sharing. Finally, we evaluate on the offline set, where memory insertion is largely disabled. This protocol evaluates memory as reusable knowledge rather than episodic recall. The agent evaluated on the offline set can be viewed as a new agent that inherits a pre-built memory graph, testing whether accumulated procedural and semantic knowledge can mitigate cold-start issues. Additional experimental details and implementation specifics are provided in Appendix F.3.

As shown in Table 5, PLUGMEM significantly improves success rates on the offline set across domains, with strong gains on the Multi-site tasks. These results demonstrate effective reuse of accumulated procedural and semantic knowledge, mitigating cold-start issues and supporting compositional generalization.

### 4.6. Discussion: Why Can a Task-Agnostic Memory Outperform Task-Specific Designs?

A natural question is why a task-agnostic memory module can outperform systems tailored to individual benchmarks. Our results suggest that the key difference lies not in reject-

ing task-specific heuristics, but in prioritizing what fundamentally makes agentic memory effective. Task-specific designs often encode benchmark-specific insights through customized memory units or transformations, implicitly assuming that relevant memory will be available when needed. While effective within their scope, such approaches conflate memory transformation with memory utility. In contrast, our findings highlight that agentic memory is fundamentally retrieval-driven. Without effective retrieval, neither task-specific abstractions nor carefully engineered memory representations translate into performance gains, as consistently shown in our ablations. At the same time, retrieval alone is insufficient. Its effectiveness is bounded by how memory is structured, since structuring determines which aspects of experience can be indexed and recovered. Our knowledge-centric structuring enables retrieval over semantically meaningful and decision-relevant abstractions, allowing useful information to surface at decision time.

Importantly, PLUGMEM is designed as a task-agnostic memory backbone that targets the shared retrieval and representation challenges underlying agentic memory designs. From this perspective, task-specific memory approaches and heuristics can be naturally layered on top of PLUGMEM, rather than viewed as alternatives to it. Our framework therefore provides a common foundation on which task-specific adaptations can be applied to further improve performance. We empirically validate this view through additional task-adaptation experiments, where representative task-specific heuristics and memory transformation strategies from prior baselines are integrated into PLUGMEM. These adaptations consistently lead to further performance improvements beyond using PLUGMEM alone, indicating that task-specific techniques and our task-agnostic memory design are *complementary*. Detailed experimental setups and results are provided in Appendix I.

## 5. Conclusions

We presented PLUGMEM, a task-agnostic plugin memory module that organizes agent experience into knowledge-centric representations to enable effective retrieval of decision-relevant memory across diverse agentic tasks. Through extensive experiments, we demonstrate that PLUGMEM consistently improves end-task performance while reducing agent-side memory cost under a unified utility–cost evaluation framework. Beyond standalone usage, PLUGMEM serves as a general memory backbone that can be augmented with task-specific heuristics, with task-adaptation experiments showing further gains. Overall, these results position PLUGMEM as principled foundation for transferable and efficient memory in LLM agents, pointing toward more general and extensible memory systems for long-horizon decision-making.

## Impact Statement

This work proposes a task-agnostic plugin memory module for LLM agents that organizes long-term experience into reusable propositional and prescriptive knowledge. By improving memory efficiency and cross-task generalization, the proposed approach has the potential to support more capable and maintainable agent systems across a wide range of applications, including long-horizon interaction, knowledge-intensive reasoning, and tool-use agents.

At the same time, enhancing long-term memory and knowledge reuse in agents may introduce potential risks. Persistent memory abstractions could encode sensitive information if derived from personal interactions, and improved agent autonomy may amplify both beneficial and harmful uses of automated systems. In addition, abstracted knowledge may inherit biases or errors present in underlying experience.

This work does not involve the collection of sensitive personal data. Future applications should consider appropriate memory management, provenance tracking, and access control mechanisms to mitigate privacy and misuse risks. Overall, we believe this work contributes positively to the development of general and efficient agent memory systems while highlighting the importance of responsible use in downstream settings.

## Acknowledgments

This work is supported in part by the National Science Foundation (NSF) and the Institute of Education Sciences (IES) under Grants 2229612, 2433308, and 2505932. We thank Microsoft Research for providing LLM API support that facilitated this research.

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

# A. Motivation, Research Questions, and Novelty

## A.1. Background and Motivation

Recent advances in LLM-based agents have highlighted the critical role of external memory systems in enabling long-horizon reasoning and decision-making. However, existing memory paradigms in both academic research and engineering practice exhibit a fundamental limitation: **they adopt fragmented and incompatible definitions of memory** (Li et al., 2025; Kang et al., 2025; Chhikara et al., 2025).

Specifically, prior work typically focuses on only one of the following two forms of memory:

- **Factual (semantic) memory**: summaries of past experiences in the form of facts, such as user preferences, historical events, or world knowledge (Huang et al., 2026; Gutiérrez et al., 2025; Sarthi et al., 2024).

- **Procedural memory**: distilled knowledge of how to perform tasks, often represented as workflows, action sequences, or strategies derived from past executions (Wang et al., 2024b; Fang et al., 2025; Zhao et al., 2024).

These two types of memory are often treated in isolation. For example, conversational agents emphasize factual memory about users, while web agents and embodied agents prioritize procedural knowledge such as action workflows. As a result, existing systems are typically designed around a *single memory type*, leading to solutions that are inherently task-specific and difficult to generalize.

However, in real-world agent scenarios, **both types of memory are indispensable and inherently complementary** (Tulving, 1972; Squire, 2004). Consider a personal assistant tasked with purchasing a branded wallet on behalf of a user. To successfully complete this task, the agent must rely on **procedural knowledge**, such as how to navigate different shopping websites, understand their layouts, search for products, apply filters, and complete the checkout process; each of which may vary significantly across platforms. At the same time, the agent must also leverage **factual knowledge** about the user, including preferences for color, style (e.g., minimalist versus ornate), and acceptable price range. These user-specific facts are crucial for selecting a product that satisfies the user's expectations. Therefore, effective decision-making requires not only access to both types of memory, but also their **dynamic interaction** within a unified framework.

Despite their complementary nature, current memory systems lack a unified framework to jointly model, retrieve, and utilize these heterogeneous memory types. This fragmentation leads to a critical limitation: **existing methods cannot effectively handle scenarios where multiple memory types must be jointly leveraged**.

## A.2. Research Question

Motivated by this gap, we ask the following central research question:

> *Can we design a unified, task-agnostic memory module for LLM agents that simultaneously supports both factual and procedural memory, and enables their effective interaction for decision-making?*

This question is practically important for two reasons:

- Any non-trivial agent task requires both "knowing that" (facts) and "knowing how" (procedures).

- These two forms of knowledge must not only coexist, but also **interact dynamically** during task execution.

Importantly, the challenge is not merely to apply existing systems to multiple benchmarks. A possible misunderstanding is to attribute the limited applicability of prior methods to incomplete evaluation, i.e., that these systems could potentially generalize if only they were tested on more diverse benchmarks. However, this is not the case. Rather, prior methods are **practically constrained by their design**: they are typically built around a single type of memory, with representations, abstractions, and retrieval mechanisms tailored specifically to that setting (Wang et al., 2024b; Gutiérrez et al., 2025).

As a result, these systems are not simply under-evaluated on other memory types, but they are **structurally incompatible** with them. Even with substantial engineering effort to adapt such systems to a different class of memory tasks, their

performance remains inherently limited. This is because factual and procedural memory differ not only in surface form, but also in their underlying structure, usage patterns, and role in decision-making.

For example, consider a memory module designed for long-horizon conversational benchmarks, where the primary goal is to summarize a user's past interactions and extract persistent factual information such as preferences or personal history. Such systems are optimized for aggregating and updating user-specific facts from dialogue. In contrast, web agent tasks require a fundamentally different type of memory: the agent must extract and reuse successful action workflows from past trajectories, enabling it to perform better in future interactions by leveraging procedural knowledge. The capabilities evaluated in these two settings are inherently different, as they emphasize distinct forms of memory abstraction and usage. As a result, a memory system designed for one type of benchmark cannot be effectively transferred to the other, even with significant modification. In other words, methods optimized for one type cannot be easily repurposed to the other without breaking their core design assumptions.

Therefore, the limitation of prior work is not a matter of evaluation scope, but a consequence of **task-dependent memory definitions and architectures**, which prevent them from effectively supporting heterogeneous memory types within a unified framework.

### A.3. Cognitive Science Inspiration

To address this challenge, we draw inspiration from cognitive science, which provides a principled and widely validated framework for understanding human memory and knowledge (Anokhin et al., 2024).

A long-standing distinction in cognitive science characterizes knowledge into two fundamental types (Mokyr, 2012):

- **"Knowing that"**: declarative, factual knowledge (semantic memory)

- **"Knowing how"**: procedural, action-oriented knowledge (procedural memory)

This distinction has been extensively studied in philosophy, psychology, and related fields, and has been used to explain human learning, reasoning, and decision-making processes.

We further note that this perspective is not merely theoretical, but has also been reflected in modern analyses of technological and economic progress. For example, the *2025 Nobel Prize in Economic Sciences* (Nobel Prize Outreach, 2026), awarded to Joel Mokyr, Philippe Aghion, and Peter Howitt, highlights how long-term growth is driven by innovation and the process of *creative destruction*, where new technologies and methods replace outdated ones. In their frameworks, sustained progress relies not only on discovering *that* something works (factual understanding), but also on developing and refining *how* to apply it effectively (procedural capability). The Nobel committee explicitly emphasizes that successful innovation requires both the accumulation of knowledge about the world and the ability to operationalize that knowledge into improved production processes and actions.

This provides a concrete, real-world example supporting the generality of the "knowing that" and "knowing how" distinction, and motivates its application to the design of agent memory systems.

We adopt this perspective as a guiding principle: **agent memory should be organized around knowledge types that directly support decision-making, rather than raw experience alone**.

### A.4. Our Approach: Unified Memory via Knowledge Abstraction

Based on this principle, we propose a unified approach that treats memory as **knowledge extracted from experience**. Specifically, we:

- Transform heterogeneous raw experiences into structured knowledge representations via **memory-to-knowledge abstraction**.

- Represent both factual and procedural knowledge within a unified **knowledge-centric memory graph**.

- Enable **joint retrieval and reasoning** over these knowledge types, allowing them to interact dynamically during decision-making.

A key design requirement is that these knowledge types must **coexist without conflict** and be jointly usable. For example, procedural knowledge guides how to perform a task, while factual knowledge personalizes the execution. Our system ensures that both can be retrieved, combined, and summarized into task-relevant guidance.

## A.5. Interpretation of the Title

Our title reflects the core properties of the proposed system:

**Task-Agnostic** Existing memory systems are often designed for specific benchmarks or task settings, resulting in fragmented definitions of memory. In contrast, we define memory in a **task-independent manner**: any past experience that can be abstracted into useful knowledge to guide future decision-making should be supported. Our goal is to build a memory module that generalizes across diverse agentic tasks without redesign.

**Plugin** Practical usability is a central design goal. A memory module should be easily deployable across different agent systems without requiring task-specific redesign or intrusive modifications to the agent architecture. In this work, we emphasize an **extreme level of engineering simplicity and generality**: regardless of the agent's task (including both factual- and procedural-memory-intensive scenarios), our module can be integrated with minimal effort.

Concretely, our implementation supports a lightweight integration that requires only **a few lines of code (e.g., six lines)** to *i)* import the module, *ii)* initialize memory, *iii)* perform memory updates, and *iv)* enable retrieval and memory reasoning during agent execution. This means that developers can seamlessly augment an existing agent with a fully functional memory system without redesigning task pipelines or adapting the system to specific benchmarks.

This level of plug-and-play usability represents a **significant engineering contribution**, ensuring that the proposed memory module is not only theoretically general, but also practically deployable across a wide range of real-world agent systems.

**Memory Module** We emphasize that memory is not merely storage, but a **functional and dynamic process** that transforms experience into decision-relevant knowledge, retrieves it when needed, and adapts it to the current task context.

Our design reflects the view that a memory module must go beyond simply storing or aggregating different types of memory. Instead, it should provide a **principled and unified processing pipeline** that can handle heterogeneous memory types while enabling their interaction during decision-making. In particular, we identify three essential components:

- **Structuring**: A principled abstraction process that transforms raw experiences into a unified representation, organizing both factual ("knowing that") and procedural ("knowing how") memory within the same framework.

- **Retrieval**: A task-driven mechanism that is **agnostic to memory type**, allowing relevant knowledge to be accessed based on the current task requirements rather than predefined memory categories.

- **Reasoning**: A dynamic integration process that combines retrieved knowledge across different memory types, distilling them into a coherent, task-aligned representation to support decision-making.

Crucially, these components operate as a **cohesive and interdependent pipeline**, rather than a simple aggregation of separate systems for different memory types. The structuring stage establishes a unified representation, retrieval selects task-relevant knowledge without bias toward a specific type, and reasoning enables **dynamic interaction and fusion** of heterogeneous knowledge. Together, they form a memory module that is inherently **dynamic, compositional, and decision-oriented**, rather than a static repository or a loosely coupled combination of independent subsystems.

## A.6. Discussion of Key Design Aspects and Novelty

The novelty of our work lies in rethinking agent memory from a fundamentally unified and principled perspective, rather than introducing isolated improvements to existing paradigms. Prior work typically treats memory as either factual summaries or procedural traces, and builds systems tailored to one of these forms (Wang et al., 2024b; Gutiérrez et al., 2025). In contrast, we identify that this separation is itself the core limitation, and propose a unified formulation in which both factual ("knowing that") and procedural ("knowing how") knowledge are treated as first-class citizens within a single memory system. This is not merely a matter of combining two types of memory, but of designing a representation and processing pipeline that allows them to coexist, be retrieved under a common mechanism, and interact dynamically to support decision-making.

To realize this, we shift the unit of memory from raw experience or text fragments to structured knowledge. This knowledge-centric perspective enables memory to be more compact, reusable, and directly aligned with decision-making needs. Importantly, the system is designed such that retrieval is agnostic to memory type, and reasoning serves as a unifying stage where heterogeneous knowledge is dynamically integrated into task-relevant guidance. This stands in contrast to prior approaches where different memory types are either handled by separate systems or are implicitly entangled in ways that limit their usability across tasks.

Furthermore, our design explicitly targets task-agnosticism as a primary goal. Rather than optimizing memory representations for specific benchmarks, we define memory in a way that generalizes across heterogeneous agent tasks, ensuring that any form of past experience that can be abstracted into useful knowledge can be incorporated and utilized. This is complemented by a strong emphasis on practical deployability: the proposed module is implemented as a lightweight plugin that can be integrated into existing agent systems with minimal effort, enabling broad applicability in real-world settings.

Taken together, our contribution is a coherent framework that unifies memory representation, retrieval, and usage across heterogeneous knowledge types and tasks. This positions our work as a step toward a more general, flexible, and practically usable memory system for LLM agents.

# B. Real Benchmark Cases Illustrating Why PLUGMEM Outperforms Task-Agnostic and Task-Specific Baselines

In this section, we present a series of illustrative case studies to provide a concrete, mechanism-level understanding of how PLUGMEM operates in practice. The goal of this section is to make the behavior of the system transparent, by tracing how memory is constructed, retrieved, and utilized in real task scenarios.

Through these examples, we focus on two central questions that arise naturally from the main results. **First**, how can a *task-agnostic* memory module consistently outperform systems that are specifically designed for individual tasks? **Second**, what role does the structuring module play in enabling this behavior, and how does it differ from prior approaches that rely on summarization or note-taking-style memory abstraction and representations?

To answer these questions, we compare PLUGMEM with several representative baselines, including both task-specific memory systems and summarization-based memory abstraction approaches. We present the details below.

## B.1. Understanding the Advantage of Task-Agnostic Memory over Frontier Task-Specific Designs

### B.1.1. LONGMEMEVAL

Table 9 show an examples of PLUGMEM retrieving episodic memory hierrachically on LongMemEval.

The example highlights a key reason why PLUGMEM outperforms task-agnostic baselines on LongMemEval. Unlike Zep (Rasmussen et al., 2025), which contains episode nodes in its graph but only performs retrieval at the semantic level (semantic entities and facts), and LiCoMemory (Huang et al., 2026), which performs graph retrieval for original dialogue chunks solely based on entities, PLUGMEM retrieves episodic memories by leveraging semantic memory extracted from those episodic memories. Specifically, the retrieval module first identifies semantic memories relevant to the query. In the example, semantic memory 669, semantic memroy 141, and semantic memory 146 are all related to wedding, which is the topic of the question. It then locates the source episodic memory corresponding to each retrieved semantic memory and selects those containing a sufficient number of semantic memories retrieved. In the example, semantic memory 669 and some other retrieved memories are extracted from episodic memory session 24, so episodic memory session 24 is selected. So does episodic memory session 6. We can notice that, although question-related, semantic memory 141 and 146 do give sufficient information about which wedding they are describing, making them ambiguous when counting the number. But by tracing back to its source episodic memory, PLUGMEM successfully distinguishes them, showing the importance of episodic memory which is neglected in Zep. Also, although LiCoMemory retrieves back origin dialogue chunks, the retrieval process is based on semantic entities. Due to the fact that episodic memory related a single entity is definitely more than episodic memory related to a specific piece of semantic memory, it is less efficiency and may bring more noise.

*Table 9.* Example 1 of retrieval trace showing hierarchical retrieval for episodic memory.

| LongMemEval Case Study: Hierarchical Retrieval |
| --- |
| **Query.** *"How many weddings have I attended in this year?"* |
| **Semantic Memory Retrieved** |
| ▷ **Semantic Memory 669** : *User's sister's wedding was amazing, and User had a great time planning it with her, including choosing the dress and deciding on the menu.*
▷ **Semantic Memory 141** : *User is planning a wedding and dreams of having a small, outdoor ceremony at a beach or in a park. User recently attended their college roommate's wedding in the city which had a rooftop garden ceremony overlooking the skyline.*
▷ **Semantic Memory 146** : *Social media platforms like Instagram and Pinterest can be used to follow wedding planners, venues, and photographers. Reading reviews on Yelp or Google provides insights into the venue's reputation.*
... |
| **Episodic Memory Retrieved** |
| ▷ **Episodic Memory Session 24** : *Date: 2023/10/15 (Sun) 04:44 User: I'm planning my own wedding and I was wondering if you could give me some tips on how to choose the perfect venue. By the way, I just got back from a friend's wedding last weekend, and it was amazing - the bride, Jen, looked stunning in her bohemian-inspired dress, and her husband, Tom, was clearly smitten with her. It was at a rustic barn in the countryside, and it was so cozy and relaxed...*
▷ **Episodic Memory Session 6** : *Date: 2023/10/15 (Sun) 10:57 User: I'm planning my own wedding and I need some help with finding a venue. I've always dreamed of having a small, outdoor ceremony, maybe at a beach or in a park. Do you have any recommendations or resources for finding the perfect spot? By the way, I just got back from my college roommate's wedding in the city, and it was beautiful - they had a rooftop garden ceremony overlooking the skyline, and it was so romantic. My friend Emily finally got to tie the knot with her partner Sarah, and it was amazing to see them so happy after waiting for marriage equality...*
... |
| **Reasoning** |
| ▷ *To determine how many weddings you have attended this year, let's review the information from the relevant memories provided:*
*1. **Cousin's Wedding at a Vineyard (August)**: You mentioned attending your cousin Rachel's wedding at a vineyard in August.*
*2. **College Roommate's Wedding in the City**: You mentioned attending your college roommate's wedding, which had a rooftop garden ceremony.*
*3. **Friend Jen's Wedding at a Rustic Barn**: You mentioned attending your friend Jen's wedding at a rustic barn in the countryside last weekend...* |
| **Answer. "Therefore, you have attended 3 weddings this year.".** |

### B.1.2. HotpotQA

Table 10 and Table 11 show two examples of multi-hop retrieval on HotpotQA alternating abstract node like concept and specific node like semantic node.

Across both case studies, PLUGMEM succeeds by explicitly separating *what to retrieve next* from *where the evidence resides*. In Hop0, the retriever typically surfaces a small set of low-level propositions that are semantically close to the question but may only partially resolve it. Crucially, rather than repeatedly expanding within a local neighborhood around these propositions (as in many graph-based retrievers that rely on 1–2 hop adjacency), PLUGMEM elevates the intermediate result into an *abstract routing signal*, a bridge concept (e.g., *Jim Croce*) or a bridge entity/context (e.g., *Space Shuttle Columbia*). This abstraction step turns multi-hop QA into a sequence of targeted sub-queries: first identify the missing bridge, then retrieve the specific fact needed to answer.

Technically, this behavior is enabled by the bipartite organization of our memory graphs, where high-level nodes (concepts/intents) connect to many low-level nodes (propositions/prescriptions). Routing through the abstract layer allows the retriever to "jump" beyond the narrow vicinity of an initial seed proposition and activate a much broader set of candidate evidence, which is especially important on HotpotQA where supporting facts often lie in different articles or distant parts of the corpus. In both examples, the first hop retrieves a bridge-bearing statement; the next hop uses the bridge to directly access the answer-bearing statement (birth year or first-launch year), while irrelevant but superficially related candidates (e.g., other payload specialists, other singer-songwriters) are naturally deprioritized once the bridge entity is fixed.

Finally, the reasoning module plays a complementary role by compressing the retrieved pool into a minimal sufficient evidence set (typically two propositions in these examples), which improves robustness and token efficiency. Overall,

*Table 10.* Example 1 of retrieval trace showing bridge-entity discovery and evidence compression.

| **HotpotQA Case Study 1: Bridge-Entity Multi-hop Retrieval** |
|---|
| **Query.** " *'One Less Set of Footsteps' is a song written and performed by a singer born in which year?* " |
| **Hop 0 (init).** Tags for query: [One Less Set of Footsteps, songwriter, performer, birthdate]
▷ **node 5891** : *"One Less Set of Footsteps" is a song written and performed by Jim Croce. It was released in 1973 as the first single from his album "Life and Times".* (bridge entity identified: **Jim Croce**)
▷ node 5892 : "One Less Set of Footsteps" reached a peak of #37 on the "Billboard" Hot 100, spending ten weeks on the chart.
......
▷ node 15424: Mariah Carey, born on March 27, 1969 or 1970, is an American singer, songwriter, record producer, and actress. |
| **Query_new** = Integrated ( Query, top-k nodes) |
| **Hop 1 (refine & expand).** Tags for query: [Jim Croce, birth year]
▷ **node 5897** : *James Joseph "Jim" Croce was born on January 10, 1943, and died on September 20, 1973.* (golden fact retrieved: **Jan 10, 1943**)
node 5895 : James Joseph "Jim" Croce was an American folk and rock singer active between 1966 and 1973. He released five studio albums and singles during this period.
......
node 5883 : "It Doesn't Have to Be That Way" was originally released early in 1973 as the B-side of the "One Less Set of Footsteps" single and was reissued in December of the same year as the third and final single from the album "Life and Times". |
| **Reasoning (compressed).**
▷ *"One Less Set of Footsteps" is written and performed by Jim Croce.*
▷ *Jim Croce was born on January 10, 1943.* |
| **Answer. 1943**. |

PLUGMEM's abstraction-to-specificity routing and budgeted candidate control expand the reachable search space and deepen evidence chaining, without being constrained to short range neighbor expansion from an arbitrary starting node.

### B.1.3. WEBARENA

This WebArena case study demonstrates that the advantage of PLUGMEM lies not in executing a particular workflow correctly, but in how guidance is constructed, adapted, and consumed during decision-making. As shown in Table **??**, PLUGMEM enables the agent to solve the task without relying on static workflows or verbose episodic recall, instead producing compact, step-adaptive guidance that directly reflects the agent's current informational bottleneck.

Compared to **task-agnostic** retrieval-based methods, the core limitation exposed by this task is not retrieval coverage but retrieval granularity. Flat retrieval systems can surface past trajectories or similar task descriptions, but these memories are typically either too specific (tied to particular pages or layouts) or too verbose to be directly actionable. As a result, the agent must internally interpret and reconcile large amounts of loosely relevant information, leading to context explosion and brittle decision making. In contrast, PLUGMEM retrieves procedural knowledge units—such as category navigation, constraint-based filtering, and ranking under partial observability—and further compresses them through its reasoning module into concise, decision-aligned instructions. The benefit is not simply fewer tokens, but a sharper alignment between retrieved memory and the agent's immediate control decision.

On the other end of the spectrum, **task-specific** web agents such as AWM and SteP hard-code or accumulate a single canonical workflow per domain. While effective when tasks closely match the assumed structure, such workflows lack the ability to adapt their internal logic as the task unfolds. This case highlights that web navigation tasks are not monolithic: the agent alternates between qualitatively different reasoning regimes (e.g., locating the correct semantic scope, enforcing numeric constraints, and resolving ties under incomplete information). A static workflow cannot anticipate these shifts. Although AWM updates its workflow after task completion, it does not revise guidance within a task or condition it on intermediate observations. PLUGMEM, by contrast, recomputes guidance at each step, allowing the agent's strategy to evolve as the environment reveals new structure.

More broadly, this example illustrates that PLUGMEM 's advantage comes from treating knowledge as the unit of memory access, rather than raw trajectories or fixed workflows. Task-agnostic methods fail because they retrieve too much unstructured experience; task-specific methods fail because they retrieve too little adaptability. PLUGMEM occupies a middle ground: it abstracts experience into reusable procedural knowledge while retaining the ability to contextualize and

*Table 11.* Example 2 of retrieval trace showing bridge-entity discovery and evidence compression.

---

**HotpotQA Case Study 2: Bridge-Entity Multi-hop Retrieval**

---

**Query.** *"Bill Nelson flew as a Payload Specialist on a Space Shuttle launched for the first time in what year?"*

---

**Hop 0 (init).** Tags for query: [Bill Nelson, Payload Specialist, Space Shuttle, first launch]
▷ **node 1375** (v=0.501): *In January 1986, Clarence William Nelson II became the first sitting member of the United States House of Representatives to fly in space, serving as a Payload Specialist on the Space Shuttle Columbia.* (bridge entity identified: **Space Shuttle Columbia**)
▷ node 1371: In 1983, Byron Kurt Lichtenberg and Ulf Merbold became the first Payload Specialists to fly on the shuttle.
......
▷ node 1383: Dirk Dries David Damiaan, Viscount Frimout, flew aboard NASA Space Shuttle mission STS-45 as a payload specialist, making him the first Belgian in space.

---

**Query_new** = `Integrated` (Query, top-k nodes)

---

**Hop 1 (refine & expand).** Tags for query: [Space Shuttle Columbia, first launch]
▷ **node 1365**: *Space Shuttle "Columbia" (Orbiter Vehicle Designation: OV-102) launched for the first time on mission STS-1 on April 12, 1981, marking the first flight of the Space Shuttle program.* (golden fact retrieved: **1981**)
▷ node 1380: STS-61-C's seven-person crew included . . . and Representative Bill Nelson (D-FL), the second sitting politician to fly in space.
......
▷ node 1384: A payload specialist (PS) is an individual selected and trained . . . for a NASA Space Shuttle mission.

---

**Reasoning (compressed).**
▷ *Bill Nelson (Clarence William Nelson II) served as a Payload Specialist on the Space Shuttle Columbia.*
▷ *Space Shuttle Columbia first launched on April 12, 1981 (STS-1).*

---

**Answer. 1981**.

---

re-specialize that knowledge through reasoning. This combination allows PLUGMEM to generalize across WebArena tasks while still producing highly targeted, step-specific guidance, which neither task-agnostic nor task-specific designs can reliably achieve.

## B.2. The Role of Knowledge-Centric Structuring in Memory Effectiveness

Our structuring module serves as the core component that enables *memory-to-knowledge abstraction*, transforming raw experience into explicit, decision-relevant knowledge units. To better understand its impact, we use case studies to examine how this design differs from the most closely related line of prior work, i.e., **summarization-style memory abstraction**.

Existing methods in this category also perform a form of abstraction over past experience. However, instead of organizing memory into explicit knowledge units, they typically produce higher-level summaries of interactions, such as notes (Xu et al., 2025), compressed descriptions (Huang et al., 2026), or iteratively refined abstractions of the main content (Sarthi et al., 2024; Chhikara et al., 2025). These representations may capture the overall topic or salient points of past experience, but they do not explicitly align with the underlying structure of knowledge required for decision-making.

To concretely illustrate this distinction, we compare PLUGMEM with two representative baselines. First, we consider **A-Mem** (Xu et al., 2025), which incrementally summarizes experience into higher-level notes, serving as a canonical example of abstraction-based memory that operates through progressive summarization. This allows for a direct comparison between knowledge-centric memory and high-level abstracted summaries across all three benchmarks, as shown in Table 12, 13, and 14. Second, we include **RAPTOR** (Sarthi et al., 2024), which represents a complementary abstraction paradigm based on hierarchical summarization. Due to its design, RAPTOR is primarily applicable to semantic memory settings and does not naturally support procedural memory tasks such as web agent scenarios. In particular, applying RAPTOR to such settings would require substantial prompt redesign and task-specific adaptation, and without careful tuning, its performance would not be representative of its intended use. To ensure a fair comparison, we therefore evaluate RAPTOR only on LongMemEval (Table 15) and HotpotQA (Table 16), where its design assumptions are aligned with the task requirements.

Across these case studies, we observe a consistent pattern: **the performance difference does not stem from whether abstraction is performed, but from how memory is structured and retrieved**. While summarization-based methods compress experience into more compact forms, their representations remain loosely organized and less aligned with task-

relevant knowledge units. In contrast, our knowledge-centric structuring enables more precise retrieval and more effective integration of memory during decision-making. The detailed examples are presented below from Table 12 to Table 16.

*Table 12.* PlugMem vs. A-Mem: Comparison Example from LongMemEval.

---

**Gold passage.**
User: *"…I actually helped out at the church's annual holiday food drive on December 10th…"*
Agent: *"That's great to hear! It's wonderful that you had a positive experience volunteering at the church's holiday food drive…"*

---

**A-Mem's notes (based on Zettelkasten principles).**
**Keywords:** {volunteer opportunities, …}
**Context:** *The content discusses ways to volunteer and make an impact in addressing food insecurity, particularly through involvement with food banks, pantries, meal services, food recovery, and community-based projects, targeting individuals interested in giving back to their communities through structured and meaningful activities.*
**Tags:** {Community Service, …}
(missing key memory: participation in a faith-related activity on December 10th)

---

**PlugMem's knowledge extraction result.**
**Propositional knowledge 1:**
*User volunteered at the church's annual holiday food drive on December 10th, sorting donations and packing boxes for families in need. The user found the experience rewarding.*
**Tags:** {User, church, holiday food drive, December 10th, volunteering, faith-related activity, …}
**Propositional knowledge 2:**
*Food banks and pantries need dedicated volunteers for various tasks such as sorting and packing donations…*
**Tags:** {Food banks, volunteering, donations, …}
(key memory preserved: church activity on December 10th)

---

**Takeaway.**
A-Mem uses a summarization-based memory construction strategy. In this example, the summary captures the main topic of the passage—community service and food insecurity—but fails to preserve the specific fact that the user participated in a faith-related activity on December 10th. As a result, the memory becomes poorly aligned with the later query, which focuses on one particular aspect of the original interaction.
PlugMem instead performs **knowledge extraction** rather than coarse summarization. This allows it to retain fine-grained factual units that may not be central to the main topic of the passage but can still be critical for future retrieval and question answering.

---

# C. Related Work

## C.1. Cognitive Science Based Agent Memory

Cognitive science has long studied the mechanisms of human memory, providing valuable insights into how memory can be leveraged in agents. The Atkinson–Shiffrin memory model defines long-term memory as information that can be retained and recalled over extended periods of time (Atkinson & Shiffrin, 1968). Long-term memory is further categorized into three major types (Tulving, 1972; Squire, 2004): semantic memory, which stores concepts and factual knowledge; procedural memory, which is responsible for knowing how to perform actions and skills; and episodic memory, which represents raw records of personal experiences and events. For intelligent agents, most interactions with the environment are episodic memory, such as raw contexts and working trajectories. From these episodic experiences, agents can extract more reusable representations. For example, user preferences can be distilled as semantic memory (Li et al., 2025), while web navigation skills can be abstracted as procedural memory (Wang et al., 2024b). In this sense, semantic and procedural memories are not isolated components, but rather structured abstractions derived from episodic memory. Knowledge can be viewed as a further abstraction over memory. Specifically, propositional knowledge corresponds to highly structured and verifiable abstractions of semantic memory, while prescriptive knowledge represents generalized and reusable forms of procedural memory that describe how to accomplish goal-oriented tasks (Mokyr, 2012).

Inspired by these cognitive theories, many studies have explored memory mechanisms for intelligent agents. Episodic and semantic memory have been introduced to handle long-context information, such as long documents in document understanding agents (Lee et al., 2024) and long-term user–agent interactions in conversational agents (Li et al., 2025). Additionally, some approaches construct knowledge graphs containing both episodic and semantic subgraphs to better organize and manage agent memory (Anokhin et al., 2024; Rasmussen et al., 2025). Procedural memory has also been incorporated into decision-making agents, particularly web agents, where accumulated experiences are leveraged to improve

*Table 13.* PlugMem vs. A-Mem: Comparison Example from HotpotQA.

---

**Gold passage.**
*Southwell Minster is a minster and cathedral in Southwell, Nottinghamshire, England. It is situated six miles from Newark-on-Trent and thirteen miles from Mansfield. It is the seat of the Bishop of Southwell and Nottingham and the Diocese of Southwell and Nottingham.*

---

**A-Mem's notes (based on Zettelkasten principles).**
*The text provides a brief overview of Southwell Minster, highlighting its status as a cathedral and the seat of the Bishop within the Diocese of Southwell and Nottingham, aimed at an audience interested in religious architecture or regional history.*
(overly generic; misses precise factual details such as location and distance)

---

**PlugMem's knowledge extraction result.**
▷ *Southwell Minster is a minster and cathedral located in Southwell, Nottinghamshire, England. It is situated six miles from Newark-on-Trent and thirteen miles from Mansfield.*
▷ *Southwell Minster serves as the seat of the Bishop of Southwell and Nottingham and the Diocese of Southwell and Nottingham.*
(preserves factual details faithfully, including location and distance information)

---

**Observation.**
A-Mem's note is overly generic and fails to preserve several key factual details from the original passage, especially the precise location and distance information. It also introduces interpretive language, such as the intended audience and topical framing, that is not explicitly stated in the source.

In contrast, PlugMem preserves the source content in a more faithful and fine-grained form through proposition-level knowledge extraction, retaining details that may be useful for future retrieval and downstream question answering.

---

future strategies (Zhao et al., 2024). This line of research moves toward self-evolving agents that continuously refine their decision-making policies based on incoming memory (Wang et al., 2024b; Fang et al., 2025; Ouyang et al., 2025).

However, most existing methods are designed for specific tasks and are unable to effectively support all memory types simultaneously. In contrast, PLUGMEM standardizes heterogeneous episodic memories across diverse tasks and stores extracted knowledge in a propositional–prescriptive dual knowledge graph, thereby overcoming the limitations of task generalizability present in prior approaches.

## C.2. Memory Module Designs

Across agentic systems, external memory is most commonly realized through retrieval. Agents store past interactions, observations, or documents outside the model context and retrieve relevant information at inference time to condition decision making. This retrieval first paradigm forms the basic prototype of external memory, and nearly all subsequent memory designs can be viewed as extensions built on top of it.

Early task agnostic memory systems adopt flat retrieval over unstructured episodic memory. Vanilla retrieval and vanilla RAG store raw interaction histories or documents as text and retrieve relevant segments based on similarity to the current query (Lewis et al., 2021; Wang et al., 2024a). These methods provide a simple and broadly applicable memory interface, but the retrieved content remains tightly bound to specific episodes. As a result, retrieved memories often contain redundant details and exhibit low information density when reused across different decision contexts.

To improve retrieval effectiveness, later work introduces structure on top of episodic memory. Graph based and hierarchical retrieval systems organize stored content to support multi hop retrieval, aggregation, or global reasoning (Edge et al., 2025; Sarthi et al., 2024). Related approaches further refine retrieval by altering the internal organization of memory graphs, such as ranking paths or linking propositions (Gutiérrez et al., 2025; Wang & Han, 2025). While these methods differ in structure and traversal strategy, they largely preserve the same retrieval oriented abstraction. Memory remains organized around episodic traces, entities, or text spans, and structural mechanisms primarily serve to improve access to stored experiences rather than to transform their content.

In parallel, task-specific memory systems explore more aggressive forms of memory processing. Temporal knowledge graphs for conversational agents, workflow memories for web navigation, and reasoning oriented memory banks explicitly extract higher level representations from experience (Gutiérrez et al., 2025; Wang et al., 2024b; Ouyang et al., 2025; Zhang et al., 2025). These systems demonstrate that abstracting experience into reusable representations can substantially improve performance within a fixed task setting. However, the abstraction process is typically guided by task-specific assumptions,

*Table 14.* PlugMem vs. A-Mem: Comparison Example from WebArena.

| |
|---|
| **Objective.** *"List products from the kids' bedding category by descending price."* |

**A-Mem's notes (based on Zettelkasten principles, abridged).**
▷ **Memory 1:** *Show me products under $78 in "children dental care" category.* The retrieved trajectory emphasizes category navigation and price-range filtering, but does not mention sorting products by price.
▷ **Memory 2:** *What is the price range for products from Amazon Basic?* The retrieved trajectory mainly concerns price-range inspection for a specific brand, which is largely irrelevant to the current goal.

**PlugMem's knowledge extraction result**
▷ Navigate to the relevant subcategory under *Home & Kitchen*, specifically *Kids' Bedding*.
▷ Use filtering options only if needed to narrow the search space.
▷ Use the *Sort By* dropdown to organize products by *Price*.
▷ Then inspect the ranked list and proceed with the desired product interaction.

**Observation.**
The correct strategy for this task is to first access the *Kids' Bedding* subcategory and then sort the product list by price in descending order. PLUGMEM captures this action pattern and provides guidance aligned with the current objective. In contrast, A-Mem retrieves trajectories that are only loosely related at the surface level: one focuses on price-range filtering in another category, and the other concerns price inspection for a specific brand. As a result, the retrieved memories are not well adapted to the current task.

which constrains reuse of the memory module across heterogeneous agents and environments.

Viewed together, prior work reveals a key distinction in how memory is treated. Some systems focus on improving retrieval over episodic experience, while others implicitly or explicitly transform experience into more abstract forms. Only the latter enables memory to generalize across episodes and tasks, since episodic memory is inherently tied to a particular agent and interaction trajectory, whereas semantic and procedural knowledge abstract away task-specific execution details. This observation motivates memory to knowledge transformation as a critical operation for reusable agent memory.

PLUGMEM builds on this evolution by making memory-to-knowledge transformation an explicit and central component of the memory module. Instead of treating episodic traces as the primary unit of retrieval, PLUGMEM organizes memory around propositional and prescriptive knowledge units corresponding to semantic and procedural memory. These knowledge units serve as the basis for memory organization, retrieval, and reasoning, while episodic memory is retained as verifiable source evidence. By elevating knowledge to the unit of memory manipulation, PLUGMEM enables memory to be reused across agents and tasks without task-specific redesign, while remaining compatible with the retrieval based external memory paradigm.

### C.3. Memory Benchmarks and Evaluation Scope

A broad range of benchmarks have been used to study memory related behaviors in language models and agents, spanning diverse task settings and evaluation objectives. One prominent line of work focuses on long context utilization, evaluating whether models can access and reason over large inputs within a single inference window, such as long document understanding, narrative comprehension, or needle in a haystack style retrieval (Liu et al., 2023; Bai et al., 2024; Zhang et al., 2024). These benchmarks assess context usage and attention behavior, but do not model memory accumulation, update, or reuse across time. A separate line of work focuses on long-horizon conversational agents and user-centric memory, evaluating whether systems can accumulate, maintain, and reuse information such as preferences, personal attributes, and dialogue context over extended interactions (Wu et al., 2024; Maharana et al., 2024). These benchmarks explicitly stress episodic accumulation and robustness to redundancy and noise, and are commonly used to study memory abstraction in interactive settings. Another class of benchmarks evaluates retrieval-augmented question answering over static knowledge sources, including factoid and multi-hop QA tasks, where memory is treated as a fixed external repository and the primary challenge lies in retrieval and evidence composition rather than memory evolution (Petroni et al., 2021; Kwiatkowski et al., 2019; Yang et al., 2018; Trivedi et al., 2022). While effective for studying semantic retrieval and composition, these benchmarks assume a static memory base and do not capture how memory is constructed or refined through interaction. In agentic settings, additional benchmarks evaluate end-to-end task performance in interactive environments, including web navigation, tool use, and embodied reasoning, where memory interacts with planning, perception, and action execution (Zhou et al., 2024; Trivedi et al., 2024; Furuta et al., 2024; Shridhar et al., 2021; Liu et al., 2025). In such benchmarks, memory is often entangled with other agent capabilities, making it difficult to isolate memory-specific contributions.

*Table 15.* PlugMem vs. RAPTOR: Comparison Example from LongMemEval.

---

**Gold passage.**

User Say: *"I need to update my address with my health insurance provider. Can you walk me through the process or give me a phone number to call? By the way, I just recently changed my last name..."*

Agent Say: *"Congratulations on your recent name change! I'd be happy to help you update your address with your health insurance provider..."*

---

**RAPTOR's recursive structure.**

**First-layer abstraction:**

*...You shared that you recently changed your last name from Johnson to Winters and are still getting accustomed to it...*

**Second-layer abstraction:**

*...Option 1: Online Update — Log into your health insurance account via the website or mobile app. Navigate to your profile and update your address and name (e.g., from Johnson to Winters)...*

**Third-layer abstraction:**

*...Updating Records After Name Change: gather certified documents, contact credit bureaus individually, submit written requests, and verify updates...*

(the key fact is diluted in higher-layer abstraction and mixed with irrelevant procedural details)

---

**PlugMem's knowledge extraction result.**

**Semantic memory:**

*User needs to update their address and name (changed from Johnson to Winters) with their health insurance provider.*

**Tags:** {change name, health insurance provider, address update, Johnson, Winters, ...}

(the key factual relation is preserved explicitly: Johnson → Winters)

---

**Observation.**

RAPTOR relies on recursive abstractive processing to construct higher-level memory representations. In this example, although the lower-layer abstraction still preserves the key fact that the user changed their last name from *Johnson* to *Winters*, this information becomes increasingly diluted in higher-layer abstractions and is eventually mixed with broader procedural content that is not directly useful for the query. As a result, the gold information becomes difficult to retrieve.

PlugMem instead directly extracts the key fact from the original interaction and stores it as an explicit knowledge unit. More generally, PlugMem's structuring module better preserves query-critical information than RAPTOR's recursive abstraction, because it emphasizes clear knowledge extraction rather than progressively coarser abstraction.

---

While these benchmarks differ widely in surface form, they each probe only a limited slice of memory behavior. Collectively, prior work reveals a fragmented evaluation landscape, where no single benchmark provides comprehensive coverage of how memory is accumulated, abstracted, organized, and reused in agent decision-making.

Within this landscape, memory benchmarks can be distinguished by how memory is constructed, organized, and reused. Some benchmarks emphasize long term accumulation of episodic experience, requiring agents to extract stable information from extended interaction histories. Others focus on structured access to semantic knowledge, stressing multi-hop retrieval and organization over memory. A third class emphasizes procedural reuse, where agents must generalize skills or workflows acquired from prior experience to new task instances. No single benchmark captures all of these aspects simultaneously, motivating the need for a complementary evaluation suite.

Based on this perspective, we intentionally select three benchmarks that probe distinct and complementary memory behaviors. LongMemEval targets long-horizon conversational settings where memory is accumulated over time and relevant information must be abstracted from noisy episodic interactions (Wu et al., 2024). HotpotQA evaluates semantic memory organization and multi-hop retrieval over a large static knowledge source, serving as a canonical benchmark for structured memory access and composition (Yang et al., 2018). WebArena evaluates procedural memory in interactive environments, where agents must reuse and adapt previously acquired strategies across multiple task instances derived from shared intent templates (Zhou et al., 2024).

Beyond their complementary coverage of episodic accumulation, semantic organization, and procedural reuse, the selected benchmarks are representative within their respective evaluation settings. LongMemEval, HotpotQA, and WebArena have each been widely adopted in prior work, with a broad range of memory and agent baselines already evaluated on these benchmarks. This established usage allows existing systems to be reliably reproduced and compared under consistent protocols, reducing evaluation bias introduced by task-specific tuning. As a result, evaluating memory modules on these benchmarks enables fairer and more stable comparison across heterogeneous memory designs, while supporting conclusions

*Table 16.* PlugMem vs. RAPTOR: Comparison Example from HotpotQA.

**Gold passage.**
**Title: Allan McNish**
*Allan McNish is a British former racing driver, commentator, and journalist from Scotland. He won the FIA World Endurance Championship in 2013 and has also been a co-commentator and pundit for BBC Formula One coverage on TV, radio, and online.*
**Title: 2013 6 Hours of Silverstone**
*The 2013 6 Hours of Silverstone was the opening round of the 2013 FIA World Endurance Championship season. Allan McNish, Tom Kristensen, and Loïc Duval won the race for Audi.*

**RAPTOR's recursive structure.**
**First layer (raw content).**
*Allan McNish is a former racing driver and commentator. He won the FIA World Endurance Championship in 2013 and has also been a co-commentator and pundit for BBC Formula One coverage.*
**Second layer (summary).**
*The 2013 6 Hours of Silverstone opened the 2013 FIA World Endurance Championship season. Allan McNish, alongside Tom Kristensen and Loïc Duval, won the race for Audi, marking another highlight in his endurance racing career.*
**Third layer (higher-level summary / clustering).**
*The documents concern motorsport and racing history, especially endurance competition, connecting the 2013 6 Hours of Silverstone with Allan McNish's broader career and FIA World Endurance Championship results.*
(higher-level abstraction preserves the topic but weakens the cross-node relation needed for multi-hop reasoning)

**PlugMem's knowledge extraction result.**
▷ *Allan McNish has been a co-commentator and pundit for BBC Formula One coverage on TV, radio, and online.*
▷ *Allan McNish, Tom Kristensen, and Loïc Duval won the 2013 6 Hours of Silverstone for Audi.*
(explicitly preserves the entity-event relation and the commentator affiliation needed for multi-hop reasoning)

**Observation.**
RAPTOR preserves a recursive hierarchy from concrete evidence to higher-level abstraction. However, this same compression process can weaken the cross-node relational links required for multi-hop reasoning. In this example, the higher-level summaries retain the general motorsport topic, but they do not preserve the two key facts in a form that is easy to connect: that Allan McNish won the 2013 6 Hours of Silverstone, and that he served as a co-commentator for BBC Formula One.

PlugMem instead extracts these facts as explicit knowledge units, preserving the entity-event and entity-role relations needed for multi-hop retrieval. As a result, it provides stronger support for questions whose answer depends on linking facts across passages.

---

that generalize beyond any single task or interaction pattern.

# D. Implementation Details

**Overview.** This appendix provides implementation-oriented details that complement the high-level description in Section 3.1. As summarized in Table 17, the memory structuring module includes two main phases: Phase 1 "Standardize" and Phase 2 "Structure." In Phase 1, to handle different types of episodic memory, PLUGMEM standardize the raw agent trajectory $\tau$ into uniformed episodic memory $M_{epi}$. Specifically, for each turns of observation and action $\{o_t, a_t\}$, we prompt LLM to derive the state $s_t$, the subgoal $g_t$ and the reward $r_t$. In Phase 2, PLUGMEM employs induction processes to distill this episodic memory $M_{epi}$ into Semantic Memory and Procedural Memory. These memory are the inserted into the memory graph, with a series of universal API that manage them.

## D.1. Episodic Standardization

To standardize episodic memory at time $t$, we represent each turns of the raw agent trajectory as an observation-action pair $\{o_t, a_t\}$ and formalize it through a three-stage process:

We first derive the agent state $s_t$ using the previous state $s_{t-1}$, the previous action $a_{t-1}$, the current observation $o_t$, and the initial goal $G$. Incorporating $G$ ensures the state remains grounded in the agent's initial objective. The derivation is achieved by querying an LLM with the prompt in Listing 1. The resulting state serves as a concise context summary, effectively reducing overall context length for distilling procedural memory when pursuing long-context agentic tasks (Wu et al., 2024; Zhou et al., 2024).

Next, we infer a subgoal $g_t$ from $s_t$, $o_t$, $a_t$ and $G$ to analyze how the agent decomposes the global task into incremental steps. Using the prompt in Listing 2, this step serves two purposes: it enriches the information available for knowledge

*Table 17.* **Layered Memory System Construction via Standardized Formalization.** PLUGMEM adopts a layered memory graph: raw interactions ($\tau$) are first formalized into episodic memory (Phase 1). This formalized history then serves as the input for inducing semantic and procedural knowledge blocks (Phase 2). All memory types are managed through a universal API by standard data operations.

| Memory Type | Construction Logic (Input → Output) | Graph Structure | Universal API & Cognitive Heuristics |
|---|---|---|---|
| | *Phase 1: Standardize* | | **1. Create (Ingestion & Induction)** |
| **Episodic** | **Formalization** *Input:* Raw Agent Trajectory ($\tau$) *Process:* $Formalize(\tau) \rightarrow M_{epi}$ *Role:* Converts unstructured logs into standardized Source Nodes. | **Source Node (Event Window)** *Edge:* Knowledge $\xleftarrow{proves}$ Source | • `memory.create(trajectory)` • Formalizes raw $\tau$ into standardized Episodic, and then triggers induction to abstract Facts/Skills from new Episodic entities. **2. Retrieve (Association)** |
| | *Phase 2: Structure* | | • `memory.retrieve(goal, state)` • Maps Goal and State to Intent/Concept indices and retrieve Knowledge Blocks, then traces back to Episodic Sources for context. |
| **Semantic** | **Fact Induction** *Input:* Episodic Memory ($M_{epi}$) *Process:* $Induce_{fact}(M_{epi}) \rightarrow M_{sem}$ *Role:* Distills static truths from specific event traces. | **Proposition Node (Fact Block)** **Concept Node (Entity/Term)** *Edge:* Concept $\xleftarrow{mentions}$ Proposition | **3. Update (Reflection & Refinement)** • `memory.update(feedback)` • Detect and mitigate contradictions (Semantic) or optimize workflow efficiency (Procedural). |
| **Procedural** | **Skill Induction** *Input:* Episodic Memory ($M_{epi}$) *Process:* $Induce_{skill}(M_{epi}) \rightarrow M_{proc}$ *Role:* Synthesizes reusable workflows from successful history. | **Prescription Node (Workflow Block)** **Intent Node (User Goal/Task)** *Edge:* Intent $\xleftarrow{solves}$ Prescription | **4. Delete (Forgetting & Pruning)** • `memory.delete(criteria)` • Decays low-utility Nodes (Semantic/Procedural) while compressing old Episodic Nodes to save space. |

extraction and enables trajectory segmentation. Specifically, we can structure the raw trajectory into workflow by chunking episodic memory based on the cosine similarity of adjacent subgoal embeddings.

Finally, we determine the reward $r_t$ based on $s_t$, $a_t$, $g_t$ and $o_{t+1}$ using the prompt in Listing 4. This reward quantifies the effectiveness of the action in achieving its intended goal, facilitating the extraction of quality-aware procedural memory.

Listing 1. Template for deriving the current state $s_t$ given previous state $s_{t-1}$, previous action $a_{t-1}$, and current observation $o_t$.

```
============================================================
Prompt Get_State
============================================================

You will receive four pieces of information:
Goal: The agent's current objective or task.
Previous State (at time t): A natural language summary describing the agent's context,
    history, and partial progress so far.
Action (at time t): The action the agent decided to take next, expressed in natural
    language.
Observation (at time t+1): The outcome or feedback resulting from that action.

Your task is to derive the new updated state---a coherent natural language summary that
    integrates all relevant information from the previous state, the action, and the
    new observation.

Steps to Follow:
Interpret the Inputs:
Examine the goal, the previous state, the action, and the observation to understand
    what has changed in the agent's situation and the detailed information about
    location and time.

Reason about the Update:
Describe the logical process by which the new state should differ from the previous one
    . Identify what progress has been made, what new information was gained, and how
    the context or focus may have shifted.

Generate the Updated State:
Write a clear and concise natural-language description summarizing the new state of the
    agent at time t+1. The new state should:
-- Include all the detailed information in the action and observation, especially
    information about location and time.
-- Integrate the outcome of the latest action and observation.
```

```
Output Format:
### Reasoning
(Explain step by step how the new state should be updated based on the inputs.)

### State
(Provide the final updated state summary here.)

---
Input:
Goal: {goal}
Previous State (at time t): {state}
Action (at time t): {action}
Observation (at time t+1): {observation}
```

Listing 2. Template for deriving the subgoal $g_t$ of turn $t$ given initiate goal $G$, current state $s_t$, observation $o_t$ and action $a_t$.

```
============================================================
Prompt Get_Subgoal
============================================================

At time t, the agent takes an action based on its state, observation, and overall goal.
     Your task is to infer the subgoal-the immediate or intermediate objective-that
    best explains why the agent chose this action.

Use the following information as context: Overall Goal: {goal} Current State (summary
    of past context): {state} Current Observation: {observation} Action at time t: {
    action}

Step 1: Reasoning Analyze how the current state and observation relate to the overall
    goal. Explain how the given action helps the agent make progress toward that goal-
    possibly by achieving a smaller intermediate objective. Be explicit and causal:
    describe why this action makes sense given the context.

Step 2: Subgoal Inference After reasoning, infer the agent's likely subgoal-a short
    natural-language statement that describes the immediate purpose behind the action.

Output Format:

### Reasoning
[Your reasoning process-a few sentences explaining how the action relates to the goal
    and context]

### Subgoal
[A short sentence describing the inferred subgoal]
```

Listing 3. Prompt template for deriving return from procedural memories.

```
============================================================
Prompt Get_Return
============================================================

You will receive:\newline
Goal Description: A text explaining what the agent was trying to achieve.\newline
Process Description: A text describing the agent's actions, decisions, and progress
    toward that goal.\newline
\newline
Your task:\newline
Analyze the agent's process and determine how much of the goal was completed,
    considering the following aspects:\newline
\newline
```

```
Grading Criteria (Score 1--10):\newline
10: The agent fully accomplishes the goal with no significant omissions; actions are
    fully aligned with the goal.\newline
8--9: The agent completes most of the goal with only minor gaps; strong alignment but
    not perfect.\newline
6--7: Partial completion; the agent covers many key elements but leaves notable parts
    unfinished or poorly executed.\newline
4--5: Limited progress; the agent attempts the goal but completes less than half or
    does so in an ineffective way.\newline
2--3: Very little completion; actions barely connect to the goal or achieve only
    minimal results.\newline
1: No meaningful progress; actions do not contribute to achieving the goal at all.\
    newline
\newline
Important Instructions:\newline
Base the score only on completion level and alignment with the stated goal.\newline
Do not provide explanations or commentary unless requested.\newline
Output must follow the format below exactly.\newline
\newline
Output Format:\newline
\texttt{\#\#\# Score}\newline
[number from 1 to 10]\newline
\newline
Input:\newline
---\newline
Goal:\newline
\{subgoal\}\newline
Process:\newline
\{procedural\_memory\}
```

Listing 4. Template for deriving the reward $r_t$ of action $a_t$ given current state $s_t$, goal $g_t$ and action $a_t$, as well as next observation $o_{t+1}$.

```
============================================================
Prompt Get_Reward
============================================================

You will be given:
Goal: the agent's overall objective.
State (at time t): what the agent knew and had done before taking the action.
Action (at time t): the single action the agent chose.
Observation (at time t + 1): the immediate outcome produced by that action.
Your task is to infer the reward - that is, a evaluation in natural language on how the
    agent's action contributes (positively or negatively) to achieving the overall
    goal, based on the resulting observation.
Follow these steps carefully:
1. Reasoning Process:
Explain how the action relates to the Goal given the State, and whether the Observation
    matches the expected helpful or unhelpful outcome.
Consider whether the action advances progress, causes setbacks, reveals new useful
    information, or wastes effort.
Summarize your reasoning about the causal contribution of the action to the goal.
2. Final Reward:
Use descriptive language to write a concise natural-language evaluation of the agent's
    action.
The reward should express how much and in what way the action helped or hindered
    achieving the goal.
Input:
Goal: {goal}
State (at time t): {state}
Action (at time t): {action}
Observation (at time t + 1): {observation}
Output format:
### Reasoning
[Your reasoning process here]
```

```
### Reward
[Natural-language reward statement that evaluate the agent's action]
some prompt
```

## D.2. Extract Knowledge

**Semantic** Given a standardized episodic unit $e_t$, we extract semantic memories as proposition–tag pairs. We query an LLM using the prompt in Listing 5, which returns up to $N_{\max} = 10$ items in a fixed schema. Each item contains a `Statement` field (a proposition in the main paper) and a `Tags` field (a list of associated concepts). We deterministically parse the LLM output into $(p, \mathcal{T})$ pairs, where $p$ is the proposition text and $\mathcal{T}$ is the corresponding concept set.

We materialize these outputs into the semantic graph $G^S$ by creating a low-level proposition node for each $p$ and high-level concept nodes for each $c \in \mathcal{T}$, caching dense embeddings and metadata for all nodes. We add two edge types: *i) membership* edges linking propositions to their concept tags ($p \to c$), and *ii) provenance* edges linking each proposition back to its source episodic unit in the episodic graph $G^E$ ($p \to e_t$). When multiple propositions originate from the same episodic unit, we additionally connect them with *sibling* edges to preserve local co-occurrence structure.

Listing 5. Prompt template for extracting semantic knowledge from episodic trajectories.

```
============================================================
Prompt Get_Semantic
============================================================

You are an expert at extracting precise, factual information from documents.
Your output must prioritize specificity, avoid ambiguity, eliminate redundancy, and
    strictly follow all formatting rules.

**CORE INSTRUCTIONS:**

1. Fact/Statement Extraction & Deduplication:
* Identify distinct, factual statements from the document.
* Resolving Vague References: If the subject of a statement is a pronoun or a vague
    description (e.g., 'the film', 'the band', 'the company', 'he', 'she', 'they'), you
     MUST rewrite the statement based on understanding of whole document so that the
    subject is a fully specified and concrete entity name taken from the document. You
    are NOT allowed to keep vague subjects in the final fact/statement.
* Example of Resolving Vague References:
BAD: 'The film was directed by xxx.', 'This movie is produced by xxx'
GOOD: 'Vaada Poda Nanbargal was directed by Manikai.', 'Vaada Poda Nanbargal is
    produced by xxx'
* Concrete Phrasing: Every statement MUST be phrased using explicit, identifiable names
     or titles. You are FORBIDDEN from using ANY vague references including but not
    limited to: 'the tour', 'the film', 'the movie', 'the band', 'it', 'he', 'she', or
    'they'.
* Example of Concrete Phrasing:
BAD: 'The tour earned over $50 million.'
GOOD: 'NSYNC's Second II None Tour earned over $50 million.'
* Statement Length Policy (IMPORTANT): Each fact/statement does NOT have to be a single
     short sentence. A statement MAY be a compact multi-sentence block that groups
    tightly related information, but it must contain AT MOST 4 sentences total. These
    sentences should come from the original document material, potentially lightly
    edited ONLY to resolve vague references.
* Avoid Redundancy: You MUST merge similar or overlapping facts into single,
    comprehensive statements. Do NOT create multiple statements that repeat the same
    core information with minor variations.

2. Tag Generation:
* For each fact/statement, generate a list of tags.
* The number of tags per fact is flexible and should reflect the information density of
     the statement.
* Tags should cover key spans such as: entity names, years, numbers, nationalities,
```

```
              languages, genres, roles, object types, descriptive words, etc.
    * The MAJORITY of tags SHOULD be SHORT TEXT SPANS copied VERBATIM from the statement (
        exact substrings). These tags are directly grounded in the surface form of the text
        .
    * You MAY occasionally create additional tags that are not literal substrings of the
        current statement, in the following cases:
    * You combine adjacent words into a single phrase (e.g., 'romantic comedy film').
    * You import a surface span (exact phrase) from ANOTHER part of the document to make
        the meaning of the current statement explicit (for example, when resolving pronouns
         or phrases like 'the film', 'the band', etc.).

    * When the subject of a statement is a pronoun or a vague description (e.g., 'the movie
        ', 'the band', 'the hotel'), you MUST add at least one tag that names the
        underlying entity explicitly, using the exact surface form that appeared elsewhere
        in the document. This cross-statement tag may come from a previous or later
        sentence as long as it is clearly the same entity.
    * If a tag is a verb, you MUST use its base (lemma) form (e.g., 'play', not 'played', '
        playing', or 'to play').
    * No Schema/Type Labels: You are FORBIDDEN from using meta-labels or ontology-like
        category names such as: 'Name', 'Person', 'Year', 'Date', 'City', 'Country', '
        Location', 'Genre', 'Language', 'FilmTitle', etc. Tags are NOT type labels; they
        are content-bearing phrases.
    * Tags should be relatively short and as fine-grained as needed. For example,
        adjectives and modifiers like 'Indian', 'Tamil-language', 'romantic', 'comedy'
        SHOULD usually be separate tags if they appear in the statement.

    OUTPUT CONSTRAINTS:
    * Extract up to 10 facts, but prioritize QUALITY over quantity. If there are fewer than
         10 truly distinct facts, output fewer.
    * Each fact must provide unique information not covered by other facts.
    * ABSOLUTELY NO generic references -- every statement must explicitly name the specific
         entity.

    **DOCUMENT:**
    {observation}

    OUTPUT FORMAT:

    ### Facts

    1. **Statement:** [statement]
    **Tags:** [tag0, tag1, tag2, tag3, ...]
    2. **Statement:** [statement]
    **Tags:** [tag0, tag1, tag2, tag3, ...]
    ...
```

**Procedural**  Given a standardized episodic sequence $M_{\text{epi}} = [e_t]_{t=1}^{T}$ with subgoal annotations $g_t$, we first segment the full trajectory into coherent workflow chunks by detecting shifts in adjacent subgoals. We compute embeddings of each subgoal and start a new segment if the two adjacent subgoals have similarity below a set threshold. Each segment $M_{\text{epi}}^{(i)}$ is then linearized into a compact process trace (e.g., a stepwise state–action–reward description) and fed to an LLM using the prompt in Table 6 to produce a structured *(intent, prescription)* pair $(u^{(i)}, \pi^{(i)})$, where $u^{(i)}$ abstracts the segment objective and $\pi^{(i)}$ describes an environment-agnostic procedure that can be reused in future tasks.

To support quality-aware reuse, we assign each prescription a scalar return score $\rho^{(i)} \in [1, 10]$ via a separate LLM evaluator (Table 3), conditioned on the intent and the segment trace. We then materialize $(u^{(i)}, \pi^{(i)}, \rho^{(i)})$ into the procedural graph $G_P$ by creating an intent node and a prescription node, caching embeddings and metadata for both. To reduce intent proliferation, an incoming intent embedding $\phi(u)$ is matched against existing intent nodes; if the best cosine similarity exceeds threshold $\theta_{\text{equal}}$, we merge the two intent strings via an LLM rewrite (Table 7) and refresh the node embedding,

otherwise we create a new intent node. The hierarchical relation is represented as a directed adjacency from intent to its prescriptions, i.e., $u \xrightarrow{\text{SOLVES}} \pi$ (implemented as intent-node child lists, hence unidirectional). Finally, each prescription node is linked back to its originating episodic evidence through provenance edges $\pi \to e_t$ for all $e_t \in M_{\text{epi}}^{(i)}$, enabling downstream verification and optional recovery of concrete interaction context when a retrieved procedure requires elaboration.

Listing 6. Prompt template for extracting procedural experience from episodic trajectories.

```
============================================================
Prompt Get_Procedural
============================================================

You will be given an utterance of an agent. Your task is to analyze the utterance and
    derive
-- a main goal that the agent is pursuing.
-- an experiential insight --- a concise reflection that summarizes the agent's
    behaviour.

Follow this process when generating your response:

Reasoning: Analyze the utterance and write down the generalizable information and
    patterns that would be useful as memory for future tasks.

Output goal and experiential insight: Produce one sentence describing the general goal
    of the utterance, using abstract language. Produce one paragraph in natural
    language that clearly expresses the experiential insight and the reflection that
    summarizes the agent's behaviour.

Output format:

### Reasoning
[Your reasoning process]

### Goal
[A sentence that concludes the goal]

### Experiential Insight
[The paragraph that expresses the experiential insight]

Input: Trajectory: {trajectory}

============================================================
Prompt Get_Return
============================================================

You will receive:
Goal Description: A text explaining what the agent was trying to achieve.
Process Description: A text describing the agent's actions, decisions, and progress
    toward that goal.

Your task:
Analyze the agent's process and determine how much of the goal was completed,
    considering the following aspects:

Grading Criteria (Score 1--10):
10: The agent fully accomplishes the goal with no significant omissions; actions are
    fully aligned with the goal.
8--9: The agent completes most of the goal with only minor gaps; strong alignment but
    not perfect.
6--7: Partial completion; the agent covers many key elements but leaves notable parts
    unfinished or poorly executed.
4--5: Limited progress; the agent attempts the goal but completes less than half or
    does so in an ineffective way.
2--3: Very little completion; actions barely connect to the goal or achieve only
```

```
    minimal results.
1: No meaningful progress; actions do not contribute to achieving the goal at all.

    Important Instructions:
    Base the score only on completion level and alignment with the stated goal.
    Do not provide explanations or commentary unless requested.
    Output must follow the format below exactly.

    Output Format:
    ### Score
    [number from 1 to 10]

    Input:
    ---
    Goal:
    {subgoal}
    Process:
    {procedural_memory}
```

---

**Listing 7. Prompt template for merging two similar intents.**

```
============================================================
Prompt Get_New_Subgoal
============================================================

Each goal may contain overlapping or complementary information.
Your task is to carefully combine them into a single, coherent, and well-structured
    goal that preserves all important details from both.
Avoid redundancy and ensure the merged goal sounds natural and consistent in tone.

Input:
Earlier goal: {goal_1}
Later goal: {goal_2}

Output:
Merged goal: [Write the unified goal here]
```

---

### D.3. Retrieval with Interleaved Abstraction and Specificity

This appendix provides implementation-oriented details that complement the high-level description in Section 3.2. PLUG-MEM performs interleaved multi-hop retrieval over the semantic graph $G^S$ and procedural graph $G^P$ by alternating between *i)* abstraction-level routing through concept/intent nodes and *ii)* specificity-level expansion over proposition/prescription nodes. Given a query $Q$, the retriever is orchestrated by three LLM prompts: GETMODE (Listing 8) selects the memory type(s) to emphasize; GETPLAN (Table 9) proposes the next-hop abstract signals (concept tags for $G^S$ or subgoals/intents for $G^P$); and MULTIHOPCTRL (Listing 10) serves as a retrieval controller that decides whether the current evidence is sufficient to stop, or which low-level nodes to prioritize in the next hop.

At hop $t$, retrieval starts from a low-level candidate pool $C_t$. We first expand candidates through two complementary channels. *(Embedding channel)* We embed the current query text $Q_t$ and retrieve a set of high-scoring low-level nodes by dense similarity. *(Linking channel)* In parallel, we run GETPLAN to produce an abstract plan (concept tags for $G^S$ or intents/subgoals for $G^P$), match it to the corresponding high-level nodes, and activate their adjacent low-level neighbors via membership edges. The two candidate sets are merged (union), then reranked and pruned back to a fixed budget (top-$K$) using a value function (primarily relevance, optionally combined with metadata such as importance/recency). Throughout the process, high-level nodes are used only as intermediate routing signals and are not retained in $C_t$.

After obtaining the budgeted candidate set, we invoke MULTIHOPCTRL to assess sufficiency. Concretely, MULTIHOPCTRL returns a strict JSON object with an `enough` flag and an optional list of selected low-level node IDs (`top_node_ids`), constrained to be a subset of the available candidates; if `enough=true`, it returns an empty list. If `enough=true`, retrieval terminates early and the current candidates are returned (or mapped to provenance-linked episodic nodes when

episodic memory is requested). Otherwise, we keep only the selected nodes (capped in our experiments) as the "focus set" and integrate them with the previous query to form the next-hop query $Q_{t+1}$. This integration can be performed by an LLM-based rewriter, but we find that simple concatenation of $Q_t$ and the selected facts performs comparably well and is used as the default for efficiency.

The retrieval loop repeats until MULTIHOPCTRL signals sufficiency or a hop limit is reached. The overall control flow is summarized in Listing 18.

---

**Listing 8. Prompt template for inferring potential memory type to be used from query.**

```
============================================================
Prompt Get_Mode
============================================================

You are given a task description that the agent is pursuing and the observation from
    the task
Please analyze the task description and observation to determine the type of memory
    required to complete it effectively. There are three possible memory types:
Episodic Memory: This is needed if the task requires you to answer questions based on
    events. For example, answering user's question depending on historical conversation
    .
Semantic Memory: This is needed if the task requires you to recall objective
    information. For example, answer the question based on objective knowledge or
    information.
Procedural Memory: This is needed if the task is completing a subgoal under an
    interactive environment that agent need to perform a workflow. For example,
    completing an instruction in web navigation tasks.
First analyze that task and observation and decide which only one memory type needed.
    When there is a conflict, prioritize the information in the Task Description when
    making decisions.

Output Format:

### Reasoning

[Your analyze of which memory is needed depending on task and observation]

### Memory Type

## [Your final decision, episodic_memory or semantic_memory or procedural_memory]

Input:
Task Description: {task_type}
Observation: {observation}
```

---

**Listing 9. Prompt template for inferring abstraction concepts for next step of retrieval.**

```
============================================================
Prompt Get_Plan
============================================================

You are an expert at analyzing an agent's goal and current observation and generating
    retrieval tags for a goal-directed question-answering system.
Your setting:

* Goal: The agent's overall objective to accomplish.
* Current Subgoal: The subgoal the agent is currently pursuing. (can be None)
* Current State: A description of the agent's current internal state. (can be None)
* Input (Current Observation): The agent's most recent observation (typically a
    question or task instruction).
* Task: Extract a prioritized set of high-quality tags that are most likely to retrieve
     information that directly helps accomplish the Goal.
```

```
Instructions:

1. Goal-directed Tag Selection (CRITICAL):

* Read the Goal and the Current Observation carefully.
* Only generate tags that are HIGHLY LIKELY to retrieve evidence needed to solve/
    complete the Goal.
* Prefer tags that identify:
* The target entity/entities the Goal is asking about (people, organizations, places,
    works, events).
* Bridge entities implied by the observation that are likely required for multi-hop
    retrieval.
* Explicit constraints: dates, years, roles, titles, unique descriptors, numbers.

* Avoid low-signal or generic tags that are unlikely to retrieve helpful evidence (e.g
    ., "known for", "famous", "character" unless the Goal specifically depends on them)
    .

2. Concrete, Grounded Tags:

* The MAJORITY of tags MUST be short text spans copied VERBATIM from the Goal or the
    Current Observation (exact substrings).
* You MAY add a SMALL number of non-literal tags only if they are short, strongly
    implied, and clearly necessary for retrieval (e.g., a canonical name expansion or a
     standard alias).
* If a tag is a verb, you MUST use its base (lemma) form (e.g., "direct", not "directed
    " or "directing").

3. Prioritization & Quantity:

* the number of tags in total should be proper. (not "as many as possible").
* Sort tags by expected retrieval usefulness (most useful first).
* Ensure tags are content-bearing and relatively short.

4. CRITICAL -- Forbidden tags:

* Do NOT generate the tag "user".
* Do NOT use meta-labels or type names such as "Name", "Person", "Year", "Date", "City
    ", "Country", "Location", "Genre", "FilmTitle", etc.

Output Format:

### Reasoning

[You process of analyzing the information and completing the task]

### Tags

**Tags:** ["tag0", "tag1", "tag2", "tag3", ...]
(for example: "Central Area", "focal point", "famous for", etc)

### Next Subgoal

## [A single best next subgoal that the agent should pursue now.]

Input:
Goal: {goal}
Current Subgoal: {subgoal}
Current State: {state}
Current Observation: {observation}
```

Listing 10. Prompt template for controlling mutli-hop retrieval process.

*Table 18.* Multi-hop retrieval control flow (pseudo-code style).

```
Input:   query Q; graphs (G^E, G^S, G^P) with provenance links to
G^E; budget K; hop limit T_max
Output:  retrieved context R

F ← TaskAdapter(Q)                                      { F ⊆ {E, S, P} }
C ← InitLowLevel(Q, {G^S, G^P})            { score all low-level nodes }
C ← TopK(C, K)

for t = 1 to T_max do
   if Stop(Q, C) then break
   q^a ← Abstract(Q, C)          { concepts for G^S; intents for G^P }
   C ← C ∪ ExpandLowLevel(q^a, {G^S, G^P})
   C ← RerankPrune(C, K)                    { e.g., relevance, etc.  }
end for

if E ∈ F then
   R ← ToEpisodic(C, G^E)                         { map via provenance }
else
   R ← C
end if
return R
```

```
================================================================
Prompt Multi-hop_Retrieval
================================================================

You are a retrieval controller for multi-hop question answering.
Return **STRICT JSON only**:
{
"enough": true/false,
"top_node_ids": [int, int, ...]
}

**Constraints:**
-- top_node_ids length <= {n_facts_new_query}
-- top_node_ids must be a subset of available ids
-- if enough=true => top_node_ids=[]

**Question:**
{question}

**Available node ids:**
{available_ids}

**Retrieved facts:**
{semantic_memory_str}
```

### D.4. Reasoning

We design tailored reasoning prompts for each memory type based on their unique structural characteristics.
In alignment with the objective of the reasoning module is to aggregate and condense retrieved memory mentioned in Section 3.3, we query the LLM to extract information relevant to the question, utilizing the prompt structure detailed in Listing 12.

Given that episodic memory is often long and unstructured, we observed that information extraction is inefficient as most of the LLM is not deliberately trained on the task. Instead, we find that, like prompt in Listing 11, directly asks the LLM to reason over every memory and answer the question successfully leveraging the reasoning ability of LLM while containing most of the useful information in the reasoning process. The entire output including the reasoning process is then preserved as the extracted information.

Procedural memory is primarily utilized during multi-turn decision-making tasks. To support that, we prompt the LLM to integrate diverse experiences consisting of both successful and failed memory from similar tasks. As shown in Listing 13, this process generates coherent and actionable guidance to fuel the agent's next action.

**Listing 11. Prompt template for reasoning and compressing for retrieved episodic memories.**

```
============================================================
Prompt Reason_Episodic
============================================================

I will give you information of history chats between you and a user. Please answer the
    question based on the information. Answer the question step by step: first extract
    all the relevant information, and then reason over the memory to get the answer.

Information:

{information}

Current Date: {time}
Question: {question}
Answer (step by step):
```

**Listing 12. Prompt template for reasoning and compressing for retrieved semantic memories.**

```
============================================================
Prompt Reason_Semantic
============================================================

I will give you several retrieved facts. Extract all the useful information relevant to
     the question.
In the output reasoning information, use the original wording from the retrieved facts
    as much as possible, and do not replace it with synonyms or near-synonyms.
If no useful information found, just return "null".

## Output format:

### Reasoning

(process of extract information)

### Information

## (The useful information you extract)

Input:
Facts: {semantic_memory}
Current Date: {time}
Question: {observation}
```

**Listing 13. Prompt template for reasoning and compressing for retrieved procedural memories.**

```
============================================================
Prompt Reason_Procedural
============================================================

Following information will be provided:
```

```
Question: The question the user is asking.
Information: Several pieces of information that may be relevant to the question.

Your task:

1. Carefully read the user's question.
2. Analyze each piece of retrieved information and determine how relevant and useful it
    is for answering the question.
3. Based on your analysis, integrate all the useful information into a single coherent
    piece of content that helps the agent to answer the question. When you think
    information is insufficient or contradictory, generate the most possible
    information. The integrated content should be concise, accurate, and relevant to
    the user's question.

## Output format:

### Reasoning

(Your reasoning for analysis of given question and information.)

### Final Information

## (The synthesized information that should be provided to the agent.)

Input:
Question: {observation}
Information: {procedural_memory}
```

## D.5. Full Operations on Memory Graph

**Setup.** We conduct multiple rounds of controlled experiments on the HotpotQA-induced semantic subgraph. We choose HotpotQA for two practical reasons: *i)* its evidence is largely factual and extractive, which makes semantic memories comparatively stable and less sensitive to stylistic variance; and *ii)* the resulting subgraph is lightweight enough to support fast iteration over different merge thresholds and update policies.

Given a semantic node, our update routine first performs *candidate discovery* by collecting all semantic nodes that share at least one concept/tag with it (i.e., neighbors under the tag-induced projection). We then compute embedding similarity between the current node and each candidate, rank candidates by similarity, and select the top-$m$ candidates whose similarity exceeds a pre-defined threshold $\tau$ (we use $m=1$ by default, i.e., only the most similar candidate above threshold is considered). If a pair $(s_i, s_j)$ is selected, we trigger a merge operation by calling an LLM to *i)* synthesize a new semantic memory that better summarizes the combined information, and *ii)* decide whether to deactivate the original nodes (delete neither, delete both, etc.) according to explicit rules encoded in the prompt; the full prompt is shown in Listing 14.

Listing 14. Prompt template for checking and merging semantic nodes in memory update.

```
============================================================
Prompt Merge_Semantic
============================================================

You are given two memory items about a related topic. One came earlier (Information 1)
    and the other came later (Information 2).

Your tasks:
(1) Merge them into ONE improved, clear, concise statement. Do not invent new facts.
(2) Decide whether to deactivate (soft delete) the original two nodes after merging.

Inputs:
Information 1 (Earlier Information): {memory_earlier}
Information 2 (Later Information): {memory_later}

Deactivation decision rules (choose exactly ONE case):
```

```
Case A: "UPDATE_SAME_FACT"

* Condition: Information 1 and 2 are essentially describing the same fact/event, and
    Information 2 mainly updates/corrects/refines details of Information 1.
* Action: deactivate BOTH originals (earlier and later) because the merged node fully
    supersedes them.

Case B: "SAME_TOPIC_MERGE_WELL"

* Condition: Information 1 and 2 are strongly related under the same topic, and the
    merged statement reads naturally as a unified summary (not an awkward splice).
* Action: deactivate BOTH originals (earlier and later).

Case C: "WEAK_RELATED_STITCH_RISK"

* Condition: Information 1 and 2 are only weakly related; merging feels like stitching
    two segments; and deactivating either original would likely harm future retrieval.
* Action: deactivate NEITHER original.

Hard constraints:

* Output MUST be valid JSON (no Markdown, no extra text).
* relationship MUST be one of the three labels above.
* If relationship is Case A or B => deactivate_earlier=true AND deactivate_later=true.
* If relationship is Case C => deactivate_earlier=false AND deactivate_later=false.
* If the two memories conflict, prefer Information 2 as the more up-to-date.
* Output the simple reasoning of why you made the decision.

Output MUST be valid JSON with exactly these keys:

* merged_statement (string)
* relationship ("UPDATE_SAME_FACT" | "SAME_TOPIC_MERGE_WELL" | "
    WEAK_RELATED_STITCH_RISK")
* deactivate_earlier (boolean)
* deactivate_later (boolean)
* simple_reasoning (string)

Return ONLY the JSON object.
```

**Thresholds and evaluation.** We evaluate two merge thresholds, $\tau \in \{0.6, 0.7\}$, on the same HotpotQA subset. We report downstream QA performance (EM/F1) as well as graph statistics before and after update. The results show that performance remains within normal fluctuation, while the graph becomes more compact with more controllable candidate fan-out. Concretely, with $\tau{=}0.6$, 477/3413 semantic nodes trigger a merge; with $\tau{=}0.7$, 171/3413 nodes trigger a merge, reflecting the expected trade-off between aggressiveness and conservativeness.

We compare the downstream QA performance on the same HotpotQA subset before and after semantic graph update under two merge thresholds. Without update, the system achieves an EM/F1 of $61.00/74.39$. With update at $\tau{=}0.6$, the EM/F1 becomes $63.00/73.97$, and with update at $\tau{=}0.7$, the EM/F1 becomes $62.00/74.65$. Overall, these differences are within normal run-to-run fluctuation, suggesting that semantic graph update/merge does not materially degrade end-to-end performance on this subset.

**Graph quality: compactness and candidate controllability.** We characterize the structural effect of update/merge along two axes.

- **Compactness.** We measure compactness using the number of active semantic nodes $N_s$, the number of used tags $N_t$, and the number of semantic–tag bipartite edges $E_{\text{bip}}$ (attachments). With the stricter threshold $\tau{=}0.7$, the updated graph reduces active semantic nodes from 3413 to 3242 ($-5.0\%$), used tags from 12501 to 11812 ($-5.5\%$), and bipartite edges from 23230 to 20604 ($-11.3\%$), indicating lower redundancy in semantic memories and their tag attachments. We also report the (non-deduplicated) upper bound of tag-induced semantic co-occurrence pairs,

$\sum_t \binom{\deg(t)}{2}$, which decreases from 78279 to 57581 ($-26.5\%$). This suggests that update/merge effectively prunes redundant co-attachments that would otherwise create dense but uninformative connectivity.

- **Candidate controllability.** Our retrieval/update pipeline uses tag-induced candidate generation: for a semantic node, we take the union of semantic IDs attached to its tags as the candidate set. To quantify how controllable the candidate fan-out is, we sample active semantic nodes and compute the size of their candidate sets (unique neighbors reachable via tags, excluding itself). With $\tau=0.7$, the candidate set size becomes substantially smaller and thus more controllable: the sampled mean decreases from 38.36 to 31.28 ($-18.5\%$), and the sampled median decreases from 18.0 to 14.0 ($-22.2\%$). Intuitively, by merging near-duplicate semantic items and deactivating redundant originals, the graph reduces unnecessary expansions through shared tags, improving computational stability without degrading downstream EM/F1.

Overall, these results indicate that semantic update/merge can improve graph compactness and reduce redundancy while preserving retrieval support, as evidenced by stable HotpotQA accuracy under both thresholds.

## E. Information-Theoretic Evaluation of Memory Efficiency

We present a framework to quantify the utility of generated memory. We decompose memory efficiency into two dimensions: accuracy (Pointwise) and certainty (Distributional), and provide a unified interpretation of their connections.

### E.0.1. PRELIMINARIES AND NOTATION

We define the operational environment and the agent's decision-making process.

- **State Space** ($\mathcal{S}$): The set of all possible environmental states or contexts. Let $s \in \mathcal{S}$ denote the current state.

- **Action Space** ($\mathcal{A}$): The set of all possible actions. Let $a \in \mathcal{A}$ denote an action taken by the agent.

- **Optimal Action** ($a^*$): Let $a^* \in \mathcal{A}$ denote the ground-truth optimal action (or the actual action chosen by a demonstrator/oracle) given the current state $s$.

- **Memory Space** ($\mathcal{M}$): The output space of the memory generation module. Let $m \in \mathcal{M}$ denote a generated memory sequence. Here, $m$ is not necessarily a subsequence of the raw corpus, but can be a constructed representation (e.g., abstraction, inference, or summary) optimized to augment the agent's decision-making in the current state $s$.

- **Memory Generator** ($\mathcal{G}_\phi$): A parameterized function (e.g., a search engine, or an LLM) that maps the current state $s$ and a raw memory base $\mathcal{K}$ to a memory artifact. We define the process as $m = \mathcal{G}_\phi(s, \mathcal{K})$. This function encapsulates operations like retrieval and reasoning to explicate implicit logic.

- **Memory Length** ($|m|$): Let $|m|$ denote the length of the memory sequence $m$ in number of tokens.

### E.1. Pointwise Metric: Accuracy and Density

### E.1.1. QUANTIFYING INFORMATION GAIN VIA POINTWISE MUTUAL INFORMATION (PMI)

We model the agent's decision-making as a conditional probability distribution over actions and distinguish between the agent's policy with and without access to the memory module.

- **Baseline Policy (Prior):** The probability of selecting the optimal action $a^*$ given only the current state $s$, without memory augmentation:

$$P_{\text{base}}(a^* \mid s)$$

- **Memory-Augmented Policy (Posterior):** The probability of selecting the optimal action $a^*$ given the current state $s$ and the generated memory $m$:

$$P_{\text{mem}}(a^* \mid s, m)$$

To quantify the specific contribution of the generated memory $m$ toward selecting the correct action $a^*$, we employ **Pointwise Mutual Information (PMI)**.

We define the **Decision Information Gain** as:

$$\text{PMI}(a^*; m \mid s) = \log_2 \frac{P_{\text{mem}}(a^* \mid s, m)}{P_{\text{base}}(a^* \mid s)}.$$

**Interpretation.**

- If $\text{PMI} > 0$: The memory $m$ provided positive information, increasing the probability of selecting the correct action $a^*$.
- If $\text{PMI} = 0$: The memory $m$ was irrelevant to the decision.
- If $\text{PMI} < 0$: The memory $m$ was misleading, decreasing the probability of selecting $a^*$.

### E.1.2. MEMORY INFORMATION DENSITY (POINTWISE GAIN PER TOKEN)

We normalize the total information gain by the cost of processing the memory (its token length). The **Memory Information Density** is defined as:

$$\rho(a^*, m) = \frac{\text{PMI}(a^*; m \mid s)}{|m|}.$$

Substituting the PMI definition:

$$\rho(a^*, m) = \frac{1}{|m|} \cdot \log_2 \left( \frac{P_{\text{mem}}(a^* \mid s, m)}{P_{\text{base}}(a^* \mid s)} \right).$$

**Unit.** Bits per Token (bits/token).

### E.1.3. SUMMARY OF THE METRIC

This framework evaluates the memory module not only on *effectiveness* (did it help?) but also on *conciseness* (did it help efficiently?).

- **Objective:** Maximize $\rho(a^*, m)$.
- **Optimization Goal:** Generate memory content that maximizes the likelihood of $a^*$ while minimizing the token count $|m|$.

### E.1.4. AGGREGATE EVALUATION: GLOBAL MEMORY EFFICIENCY

To evaluate the memory module's performance over a dataset

$$\mathcal{D} = \{(s_i, a_i^*, m_i)\}_{i=1}^{N},$$

consisting of $N$ decision instances, we compute the **Global Memory Information Density**, denoted as $\rho_{\text{global}}$.

Unlike the arithmetic mean of individual instance efficiencies (which can be numerically unstable for short memory sequences), $\rho_{\text{global}}$ represents the *amortized information gain* per token consumed across the entire evaluation set. It answers the question: *"For every token of memory generated by the system, how many bits of decision-relevant information are gained on average?"*

We define $\rho_{\text{global}}$ as the ratio of the total cumulative Pointwise Mutual Information to the total cumulative memory length:

$$\rho_{\text{global}} = \frac{\sum_{i=1}^{N} \text{PMI}(a_i^*; m_i \mid s_i)}{\sum_{i=1}^{N} |m_i|}.$$

Expanding the PMI term, the calculable form is:

$$\rho_{\text{global}} = \frac{\sum_{i=1}^{N} \log_2 \left( \frac{P_{\text{mem}}(a_i^* \mid s_i, m_i)}{P_{\text{base}}(a_i^* \mid s_i)} \right)}{\sum_{i=1}^{N} |m_i|}.$$

**Properties of this Metric**

1. **Token-Level Weighting:** By summing the lengths in the denominator, this metric implicitly weighs the contribution of longer memory sequences more heavily, ensuring that expensive generations must justify their cost with proportionally higher information gain.

2. **Robustness:** It is robust to the "small denominator problem," where an instance with a very short memory context (e.g., $|m_i| = 1$) might otherwise produce an artificially high efficiency score that skews the dataset average.

3. **System-Wide Interpretation:** A value of $\rho_{\text{global}} = 0.5$ indicates that, on aggregate, the system requires two tokens of memory context to gain one bit of information about the optimal action.

E.1.5. CONTROL AND FILTERING: DEFINING THE EVALUATION SCOPE

To ensure the metric captures the *marginal utility* of the memory module rather than the underlying difficulty of the tasks, we introduce a filtering mechanism. This isolates instances where the agent stands to benefit from external information.

**The Redundancy Filter (High Prior Confidence)**    We exclude instances where the baseline policy is already confident in the optimal action, as memory generation in these cases is functionally redundant.

We define the **Active Evaluation Subset**, denoted as $\mathcal{D}_{\text{active}} \subseteq \mathcal{D}$, as:

$$\mathcal{D}_{\text{active}} = \{(s_i, a_i^*, m_i) \in \mathcal{D} \mid P_{\text{base}}(a_i^* \mid s_i) < \tau_{\text{conf}}\},$$

where $\tau_{\text{conf}}$ is a pre-defined confidence threshold (e.g., $\tau_{\text{conf}} = 0.8$ or $0.9$).

**Rationale.**    If $P_{\text{base}} \geq \tau_{\text{conf}}$, the agent "knows" the answer. Any generated memory $m$, even if relevant, yields negligible Information Gain (PMI $\to 0$). Including these lowers $\rho_{\text{global}}$ unfairly.

**The Empty Memory Generation Case**    We must account for cases where the memory module decides not to generate any information or returns an empty sequence ($|m| = 0$).

- **Handling:** If $|m_i| = 0$, the instance is excluded from the density calculation because the denominator is zero.

- **Note:** While excluded from the *density* metric (efficiency), these samples should still be tracked separately for *recall rate* (effectiveness).

**Refined Global Metric**    Combining the "Ratio of Sums" approach with the "Redundancy Filter," the final operational metric is calculated only over the active subset:

$$\rho_{\text{final}} = \frac{\sum_{i \in \mathcal{D}_{\text{active}}} \text{PMI}(a_i^*; m_i \mid s_i)}{\sum_{i \in \mathcal{D}_{\text{active}}} |m_i|}.$$

This refined formulation ensures we measure the *efficiency of necessary memories*, providing a cleaner signal of the module's contribution to complex reasoning.

**E.2. Geometric Analysis: The Utility–Cost Landscape**

To understand the mechanism behind efficiency gains, we analyze the memory module through a geometric lens. We construct a Utility–Cost coordinate system where:

- **X-Axis (Cost):** Memory length, $L = |m|$.

- **Y-Axis (Utility):** Information gain, $I(L) = \text{PMI}(a^*; m \mid s)$.

Standard intuition might suggest monotonic logarithmic growth of utility with respect to context length. However, in real-world agentic scenarios involving noise, we propose that the curve follows a **unimodal (peaking) distribution**.

E.2.1. THE PEAKING PHENOMENON AND NOISE TOXICITY

We model the effective information gain $I(L)$ not merely as signal accumulation, but as a superposition of signal extraction and attention dilution:

$$I(L) = S(L) - \eta(L),$$

where $S(L)$ is the logarithmic signal-accumulation function (monotonically increasing), and $\eta(L)$ represents the **noise toxicity** or cognitive-load penalty.

We identify three qualitative regions on this curve:

1. **Under-fitting Region** ($dI/dL > 0$, $d^2I/dL^2 > 0$): High marginal utility; critical semantic information is being generated.

2. **The Sweet Spot** ($dI/dL \approx 0$): The peak $I_{\max}$ where signal is maximized relative to noise.

3. **Toxicity Region** ($dI/dL < 0$): The *negative marginal utility* zone. Extending memory length $L$ introduces more irrelevant tokens (HTML tags, tangents) than valid signal, diluting the agent's attention mechanism. As a result, the probability of the correct action $P(a^*)$ *decreases* despite having "more context."

E.2.2. GEOMETRIC INTERPRETATION OF EFFICIENCY ($\rho$)

In this coordinate system, the core metric, i.e., **Global Memory Density** $\rho$, has a precise geometric meaning:

$$\rho(L) = \frac{I(L)}{L},$$

which is the slope of the *secant line* from the origin $(0, 0)$ to the operating point $(L, I(L))$.

This reveals a trade-off between two optimality criteria:

- **Point A: Maximum Performance Point.** The peak of the curve where $\frac{dI}{dL} = 0$. This is the performance ceiling independent of cost.

- **Point B: Maximum Efficiency Point.** The point where the secant line from the origin becomes a tangent to the curve. Geometrically, this maximizes the slope $\rho = I(L)/L$.

**Insight: Efficiency–Performance Gap.** Typically, Point B lies to the left of Point A. Pursuing maximum performance (moving from B toward A) requires disproportionately more tokens and yields diminishing, or even negative, returns.

**The "Epiphany" Threshold (Point C).** It is also theoretically instructive to identify the *Inflection Point* (where $I''(L) = 0$), located to the left of Point B. Economically, this represents the point of **Maximum Marginal Utility** ($\max dI/dL$). In the context of memory generation, Point C corresponds to the acquisition of the "Critical Semantic Mass", i.e., the specific token or phrase that provides the initial breakthrough in context (e.g., retrieving the correct entity name). However, stopping at Point C is generally suboptimal (under-fitting), as the agent has acquired the key signal but lacks the necessary context (accumulated between C and B) to robustly execute the decision.

E.2.3. VECTOR DECOMPOSITION OF GAINS

To quantify the improvement of our proposed memory module relative to other baselines, we perform a vector decomposition in the $L$–$I$ plane.

Let the baseline vector be

$$\mathbf{v}_{\text{baseline}} = (L_{\text{baseline}}, I_{\text{baseline}})$$

and our improved method be

$$\mathbf{v}_{\text{ours}} = (L_{\text{opt}}, I_{\text{opt}}).$$

The transformation vector is:

$$\Delta\mathbf{v} = \mathbf{v}_{\text{ours}} - \mathbf{v}_{\text{baseline}} = (\Delta L, \Delta I).$$

We classify the improvement direction as:

- **Horizontal Shift** ($\Delta L < 0$, $\Delta I \approx 0$): *Pure Compression*, where gains arise solely from cost reduction.

- **Vertical Shift** ($\Delta L \approx 0$, $\Delta I > 0$): *Pure Enhancement*, which indicates better reasoning with the same budget.

- **North-West Shift** ($\Delta L < 0$, $\Delta I > 0$): **Hybrid Gain**. This indicates the baseline operates in the Toxicity Region. By refining the memory, we simultaneously reduce cost *and* increase accuracy by removing noise that harmed decision-making.

### E.2.4. PARETO DOMINANCE

We posit that the proposed module does not merely traverse the same trade-off curve as the naive baseline. Instead, it induces a **Pareto frontier shift**.

Let the naive baseline curve be $\mathcal{C}_{\text{base}}$ and our method's curve be $\mathcal{C}_{\text{ours}}$. We claim that over the relevant domain:

$$\forall L, \quad I_{\text{ours}}(L) \geq I_{\text{base}}(L).$$

This implies our method raises the theoretical ceiling of the memory system, enabling the agent to handle more complex tasks (higher required $I$) that were previously infeasible due to the noise constraints of $\mathcal{C}_{\text{base}}$.

**Remark: Baseline Invariance and Coordinate Shift.** It is worth noting that since the naive baseline confidence $\sum \log P_{\text{base}}(a^* \mid s)$ is constant for a fixed agent, the Utility axis $I(L)$ is functionally a vertical translation of the raw posterior log-likelihood $\sum \log P_{\text{mem}}$. While this constant offset shifts the absolute position of the origin $(0, 0)$, thereby scaling the absolute value of the density slope $\rho$, it preserves the *topological features* of the landscape. Consequently, the relative comparisons between methods (e.g., the existence of a "toxicity drop" or the location of the "sweet spot") remain invariant to the baseline performance.

### E.3. Distributional Metric: Certainty and Calibration

While PMI quantifies the accuracy of the memory with respect to the ground truth $a^*$, it fails to capture the global impact of memory on the agent's uncertainty. A memory sequence might slightly increase $P(a^*)$ (positive PMI) while leaving the agent confused among many other suboptimal actions.

To measure the memory's ability to *prune the search space* and *sharpen the decision boundary*, we extend the framework to evaluate the **Distributional Information Density**.

### E.3.1. QUANTIFYING ACTION SPACE COMPRESSION VIA UNCERTAINTY REDUCTION

We employ Shannon Entropy to quantify the uncertainty (or "cognitive load") inherent in the agent's policy.

- **Prior Uncertainty** ($\mathcal{H}_{\text{base}}$): The uncertainty of the agent given only the state $s$.

$$\mathcal{H}_{\text{base}}(s) = -\sum_{a \in \mathcal{A}} P_{\text{base}}(a \mid s) \log_2 P_{\text{base}}(a \mid s)$$

- **Posterior Uncertainty** ($\mathcal{H}_{\text{mem}}$): The uncertainty after processing the memory $m$.

$$\mathcal{H}_{\text{mem}}(s, m) = -\sum_{a \in \mathcal{A}} P_{\text{mem}}(a \mid s, m) \log_2 P_{\text{mem}}(a \mid s, m)$$

The **Action Space Compression** (or Uncertainty Reduction), denoted as $\Delta \mathcal{H}$, represents the amount of information (in bits) the memory contributes toward resolving ambiguity:

$$\Delta \mathcal{H}(m \mid s) = \mathcal{H}_{\text{base}}(s) - \mathcal{H}_{\text{mem}}(s, m).$$

**Interpretation.**

- $\Delta \mathcal{H} > 0$: **Sharpening.** The memory reduced the effective size of the search space (pruning invalid options).

- $\Delta \mathcal{H} < 0$: **Confusion.** The memory introduced conflicting information, flattening the distribution and increasing uncertainty.

E.3.2. DISTRIBUTIONAL INFORMATION DENSITY (DISTRIBUTIONAL GAIN PER TOKEN)

Analogous to the pointwise metric, we define the **Distributional Information Density**, $\rho_{\text{dist}}$, as the rate of uncertainty reduction per unit of processing cost:

$$\rho_{\text{dist}}(m) = \frac{\Delta\mathcal{H}(m \mid s)}{|m|} = \frac{\mathcal{H}_{\text{base}}(s) - \mathcal{H}_{\text{mem}}(s, m)}{|m|}.$$

**Unit.** Bits of Uncertainty Removed per Token (bits/token).

**Safety Analysis: The Confidence-Validity Quadrants** To rigorously evaluate memory, we project each instance into a 2D plane defined by:

- **X-Axis (Certainty):** Action Space Compression, $\Delta\mathcal{H}(m \mid s)$.

- **Y-Axis (Accuracy):** Decision Information Gain, $\text{PMI}(a^*; m \mid s)$.

This projection categorizes memory interaction into one of four regimes:

1. **Quadrant I: Efficient Reasoning** $(\Delta\mathcal{H} > 0, \text{PMI} > 0)$
   *"Sharper and Correct."*
   The memory confirmed the correct action and ruled out distractors. The agent moved from uncertainty to correct certainty. This is the ideal operational state.

2. **Quadrant II: Corrective Calibration** $(\Delta\mathcal{H} < 0, \text{PMI} > 0)$
   *"Breaking False Confidence."*
   Here, the agent likely started with high confidence in a *wrong* action (Low Prior Entropy). The memory introduced necessary doubt, flattening the distribution but raising the probability of the true optimal action $a^*$.
   **Significance:** This represents a "Rescue" mechanism where the memory fixes the agent's internal misconceptions.

3. **Quadrant IV: The Hallucination Trap** $(\Delta\mathcal{H} > 0, \text{PMI} < 0)$
   *"Confident but Wrong."*
   The memory reduced uncertainty but pointed away from the ground truth. The agent became "dogmatically wrong."
   **Risk:** This is "Toxic Certainty," the most dangerous failure mode in retrieval-augmented generation (RAG).

4. **Quadrant III: Destructive Noise** $(\Delta\mathcal{H} < 0, \text{PMI} < 0)$
   *"Confused and Misled."*
   The memory not only failed to point to the correct answer but also increased overall confusion (entropy), effectively acting as distraction.

E.3.3. VALIDITY-ADJUSTED DISTRIBUTIONAL INFORMATION DENSITY

To synthesize the quadrant analysis into a single scalar metric that rewards efficient reasoning while penalizing hallucinations, we propose the **Validity-Adjusted Distributional Information Density** ($\rho_\Phi$). This metric integrates the *magnitude* of the distributional shift with the *directionality* of the accuracy gain.

**Metric Definition** We define $\rho_\Phi$ as the product of the validity sign and the normalized distributional work:

$$\rho_\Phi(m) = \underbrace{\text{sgn}(\text{PMI}(a^*; m \mid s))}_{\text{Direction Validity}} \cdot \frac{|\Delta\mathcal{H}(m \mid s)|}{|m|}$$

where:

- $\text{sgn}(\cdot)$ is the sign function ($+1$ for improvement, $-1$ for detriment).

- $|\Delta\mathcal{H}(m \mid s)|$ is the absolute magnitude of the uncertainty change (bits).

- $|m|$ is the memory length (tokens).

**Properties and Interpretation** This formulation provides a unified evaluation across all four cognitive regimes:

1. **Reward for Efficiency (Quadrant I):**
   When the agent becomes more certain about the correct answer ($\Delta\mathcal{H} > 0, \mathrm{PMI} > 0$), the metric is **positive**. Higher density indicates faster convergence to the truth.

2. **Reward for Rectification (Quadrant II):**
   When the memory corrects a confident error ($\Delta\mathcal{H} < 0, \mathrm{PMI} > 0$), the metric remains **positive**. Although entropy increases (the agent becomes less dogmatic), the term $|\Delta\mathcal{H}|$ captures the significant "cognitive work" performed to break the false confidence, and $\mathrm{sgn}(\mathrm{PMI})$ validates this shift as beneficial.

3. **Penalty for Hallucination (Quadrant IV):**
   When the agent becomes confident in a wrong answer ($\Delta\mathcal{H} > 0, \mathrm{PMI} < 0$), the metric becomes **negative**. The more "convincing" the hallucination (higher $\Delta\mathcal{H}$), the severe the penalty. This acts as a soft safety constraint.

4. **Penalty for Destructive Noise (Quadrant III):**
   When the memory confuses the agent and lowers accuracy ($\Delta\mathcal{H} < 0, \mathrm{PMI} < 0$), the metric is **negative**. Here, $|\Delta\mathcal{H}|$ represents the magnitude of the *confusion* introduced. Since the memory failed to support the correct action, this "negative work" is penalized, reflecting the cost of processing distracting information.

**Summary.** The metric $\rho_\Phi$ answers: *"How many bits of valid distributional reshaping (entropy shift) does the memory module provide per token of cost?"*

## E.4. Theoretical Unification: The Oracle-Divergence Principle

Thus far, we have analyzed memory efficiency through specific metrics: *Accuracy* (PMI) and *Certainty* (Entropy). In this section, we situate these metrics within a broader information-theoretic framework. We propose that the fundamental objective of the memory module is to minimize the information-geometric distance between the agent's policy and the ideal policy.

We term this the **Oracle-Divergence Principle**.

### E.4.1. THE GENERAL OBJECTIVE: DIVERGENCE REDUCTION

Let $\mathcal{Q}$ denote the **Oracle Distribution** (or the "God View"), representing the ideal policy for the current state $s$. The goal of memory generation is to transform the agent's prior belief $P_{\text{base}}$ into a posterior $P_{\text{mem}}$ that is statistically closer to $\mathcal{Q}$.

We quantify this improvement using the reduction in **Kullback-Leibler (KL) Divergence**. The generalized **Oracle Information Gain**, $\Delta_{\text{div}}$, is defined as:

$$\Delta_{\text{div}}(m) = D_{\text{KL}}(\mathcal{Q} \parallel P_{\text{base}}) - D_{\text{KL}}(\mathcal{Q} \parallel P_{\text{mem}}).$$

Since the Oracle distribution $\mathcal{Q}$ and the baseline prior $P_{\text{base}}$ are fixed for a given instance, maximizing this gain is equivalent to minimizing the divergence of the posterior:

$$\arg\max_m \Delta_{\text{div}}(m) \equiv \arg\min_m D_{\text{KL}}(\mathcal{Q} \parallel P_{\text{mem}}).$$

Ideally, if the memory is perfect, $P_{\text{mem}}$ converges to $\mathcal{Q}$, and the divergence becomes zero.

### E.4.2. PRACTICAL INSTANTIATION: PMI AS A SPECIAL CASE

While the Oracle $\mathcal{Q}$ can theoretically model soft labels or multimodal distributions, in the vast majority of discrete agentic tasks (e.g., tool selection, multi-step reasoning), the objective is canonically defined by a single unique ground truth. Under this standard deterministic setting, the Oracle distribution **collapses** from a general probability vector into a Dirac delta (One-hot) distribution:

$$\mathcal{Q}(a) = \begin{cases} 1 & \text{if } a = a^* \\ 0 & \text{otherwise} \end{cases}$$

Under this specific **One-hot Assumption**, the KL-Divergence term simplifies significantly:

$$D_{\text{KL}}(\mathcal{Q} \parallel P) = \sum_{a \in \mathcal{A}} \mathcal{Q}(a) \log_2 \frac{\mathcal{Q}(a)}{P(a)} = 1 \cdot \log_2 \frac{1}{P(a^*)} = -\log_2 P(a^*).$$

Substituting this back into the generalized gain equation yields:

$$\begin{aligned}
\Delta_{\text{div}}(m) &= [-\log_2 P_{\text{base}}(a^*)] - [-\log_2 P_{\text{mem}}(a^* \mid m)] \\
&= \log_2 P_{\text{mem}}(a^* \mid m) - \log_2 P_{\text{base}}(a^*) \\
&= \text{PMI}(a^*; m).
\end{aligned}$$

**Implication: Tractability and Sufficiency.** This derivation reveals that **Pointwise Mutual Information (PMI)** is not merely a heuristic, but the **algebraic collapse** of the generalized Divergence Reduction under the deterministic assumption. This establishes PMI as the optimal engineering metric because it is:

- **Computationally Tractable:** It requires tracking only the probability of the ground truth $P(a^*)$, avoiding the computational cost of modeling the full distributional distance.

- **Theoretically Sufficient:** In the one-hot regime, maximizing PMI is mathematically isomorphic to minimizing the KL-divergence.

Thus, PMI serves as a proxy that effectively bridges abstract information geometry with practical, low-cost evaluation.

### E.4.3. THE ENTROPIC COROLLARY

This generalized perspective also explains the role of Entropy Reduction ($\Delta \mathcal{H}$).

Since the Oracle distribution $\mathcal{Q}$ (One-hot) has an entropy of zero ($H(\mathcal{Q}) = 0$), any policy $P_{\text{mem}}$ that successfully minimizes the divergence $D_{\text{KL}}(\mathcal{Q} \parallel P_{\text{mem}})$ must necessarily lower its own entropy.

$$P_{\text{mem}} \to \mathcal{Q} \implies H(P_{\text{mem}}) \to 0.$$

Therefore, *Accuracy* (PMI) and *Certainty* (Entropy) are not independent objectives. They are coupled features of the same optimization process:

- **PMI** measures the *alignment* of the probability mass with the peak of $\mathcal{Q}$.

- **Entropy** measures the *compactness* of the distribution, which is a prerequisite for resembling $\mathcal{Q}$.

### E.4.4. SUMMARY.

In this unified view, our framework offers a dual-layered approximation of the theoretical trajectory $\Delta_{\text{div}}$:

1. **The Accuracy Gain Efficiency Proxy ($\rho$):** By assuming the standard One-hot Oracle, the Pointwise Mutual Information $\rho$ serves as the **primary computational metric**. It provides an exact, low-cost measurement of the agent's transport toward the optimal action $a^*$, making it sufficient for large-scale performance benchmarking.

2. **The Validated Certainty Gain Proxy ($\rho_\Phi$):** The Distributional Density $\rho_\Phi$ acts as a **diagnostic complement**. It approximates the entropy-minimization requirement of the Oracle-Divergence, explicitly safeguarding against "off-target" confidence (where the agent minimizes entropy toward a wrong distribution $\mathcal{Q}' \neq \mathcal{Q}$).

Thus, future memory research and prompting work can employ $\rho$ to measure *how much* the additional tokens (e.g., the memory tokens) help decision-making, and $\rho_\Phi$ to diagnose *how certainly* they do so.

# F. Additional Experiment Details

## F.1. LongMemEval

**Setup.** To evaluate the agent's ability to answer questions based on historical user-agent conversations, we utilize LongMemEval (Wu et al., 2024). Specifically, we employ the $LongMemEval_S$ subset, which features a conversation context of 115K tokens per test case, aligning with established standards in agent memory research (Li et al., 2025; Rasmussen et al., 2025). For the PLUGMEM framework, we adopt Qwen2.5-32B-Instruct for the inference of structuring and retrieval modules, and GPT-4o for the reasoning module. All baseline methods is driven by GPT-4o. NV-Embed-v2 is used for generating embeddings and GPT-4o is used for all methods as the base agent that answer the question depending on memory generated. During the retrieval phase, we set k=10, extracting the top 10 most relevant memory nodes for downstream reasoning.

**Baselines.** We consider three categories of baselines: *i) Vanilla*, including no historical conversation in its prompt that answer depending on the backbone model's parametric knowledge, or simply answering "I don't know".*ii) Task-agnostic*, which are not specifically designed for agent-user conversation QA benchmarks. Examples include a standard dense RAG pipeline that performs turn-level embedding-based retrieval; and agentic memory approaches such as A-Mem (Xu et al., 2025), which take notes from episodic memory and maintain/update them via a graph-based organization mechanism. *iii) Task-specific*, tailored for historical conversation based QA, including recursive abstraction based method RAPTRO (Sarthi et al., 2024), knowledge graph based method Zep (Rasmussen et al., 2025), and cognitive science oriented structural retrieval method LiCoMemory (Huang et al., 2026). Specifically, for Zep, we adopt the result reported in their paper. For A-Mem, RAPTOR and LiCoMemory, we randomly shuffle the test cases in the original `longmemeval_s_cleaned.json` file from LongMemEval and evaluate them on the same 100 prefix cases.

**Metrics.** We report Accuracy (Acc. in Table 3), which is evaluated based on LLM-as-a-Judge method of Long-MemEval(Wu et al., 2024), using GPT-4o as evaluator. Additionally, we also report global density relying on information-theoretic measures introduced in Section 4.1. Specifically, we instantiate the likelihood terms in Eq. (1) using per-instance answer overlap: $P_{\text{mem}}$ and $P_{\text{base}}$ are set to 1.0 if the evaluator judges the answer as correct, and 0.0 otherwise. To avoid mathematical errors when computing $\log(P_{\text{mem}}/P_{\text{base}})$ in cases where either $P_{\text{mem}}$ or $P_{\text{base}}$ equals 0.0, we apply additive smoothing and use $\log\left(\frac{P_{\text{mem}}+\epsilon}{P_{\text{base}}+\epsilon}\right)$ when computing PMI, where $\epsilon$ is **1%** of the success rate of agent without memory on LongMemEval.

**Main Results.** The experiment results are summarized in Table 3. PLUGMEM outperforms all baselines on LongMemEval. Moreover, PLUGMEM attains the highest *global information-gain density* among all baselines. As shown in Table 19, PLUGMEM achieves competitive performance compared to several task-specific methods. Notably, PLUGMEM delivers the top performance on the multi-session subset. This subset is particularly challenging as it requires both accurate retrieval and precise memory extraction; the agent must retrieve multiple "gold" memories and distinguish between them to maintain an accurate count. This reveals that the architecture of PLUGMEM successfully bridges the gap between massive data retrieval and precise cognitive extraction, proving that integrating external memory modules can significantly enhance an agent's long-term consistency in dynamic, multi-turn environments.

*Table 19.* Subset Performance on LongMemEval. (S-S-U: single-session-user, S-S-A: single-session-assistant, S-S-P: single-session-preference, K-U: knowledge-update, T-R: temporal reasoning, M-S: multi-session)

| Method | S-S-U | S-S-A | S-S-P | K-U | T-R | M-S | **Avg.** |
|---|---|---|---|---|---|---|---|
| Zep | 92.9 | 80.4 | 56.7 | 83.3 | 62.4 | 57.9 | 71.2 |
| LiCoMemory | 92.9 | 90.9 | 50.0 | 81.2 | 65.4 | 63.0 | 73.0 |
| PLUGMEM | **94.3** | **98.2** | **86.7** | **83.3** | **76.7** | **75.2** | **82.8** |

**Statistical Variance.** We compare A-MEM, RAPTOR, LICOMEMORY, and PLUGMEM on the same evaluation set, reporting Accuracy (Acc.). Each method is run three times with different random seeds; Table 20 reports mean ± standard deviation across the three runs. Our method PLUGMEM achieves the highest Acc..

**Ablations.** We further conduct ablation studies on the three components of PLUGMEM. For `no-structuring`, we replace structured indexing with a simple chunking of the original user–agent conversations at the turn level. For

*Table 20.* Mean $\pm$ std over 3 runs per method (best in bold). Higher is better.

| Method | A-Mem | RAPTOR | LiCoMemory | PlugMem |
|---|---|---|---|---|
| Acc. | $60.0_{\pm 1.63}$ | $49.33_{\pm 2.87}$ | $71.0_{\pm 2.16}$ | $\mathbf{82.33}_{\pm 1.88}$ |

`no-retrieving`, we directly concatenate all semantic memories extracted by the structuring module and feed them into the reasoning module without retrieval.As shown in Table 6, removing all the module leads to consistent degradation in both task performance and global information-gain density, highlighting their importance for effective memory utilization. Specifically, removing the retrieval module results in the worst performance, indicating that effective memory utilization must be grounded in retrieval. Removing the structuring module degrades performance to a level comparable to vanilla retrieval. While removing the reasoning module only causes a slight drop in accuracy, it leads to a substantial increase in input memory tokens.

**Cost Analisys.** Table 21 reports the retrieval latency and token consumption per sample for different methods on LongMemEval. Since Zep is only partially open-sourced, we randomly evaluate several samples using Graphit, the open-source graph module on which Zep is built, to estimate its token usage. All the experiments are conducted on four NVIDIA A800 GPU.

Importantly, while the token consumption of PLUGMEM is within similar order of magnitude as competing approaches, the actual deployment cost differs substantially. PLUGMEM relies exclusively on open-source models for inference, which can be executed offline or at significantly lower per-token cost. In contrast, several competing methods depend on closed-source models such as GPT-4o for inference, whose per-token pricing is considerably higher. As a result, when accounting for model pricing rather than token count alone, PLUGMEM is expected to be substantially more cost-efficient in practice, despite comparable token usage.

*Table 21.* Cost Statistics on LongMemEval (k tokens per sample). Q32 denotes the average time for one call to Qwen-32B, and 4o denotes the time for one call to GPT-4o. We parameterize external model-call times instead of measuring them on our own machine, as they may vary across hardware, configurations, and API settings, providing a fairer and more reproducible baseline for comparison.

| Method | Latency (s) | NVE | $Q32_{in/out}$ | $4o_{in/out}$ |
|---|---|---|---|---|
| Task-Agnostic | | | | |
| Vanilla RAG | - | 107 | - | - |
| A-Mem | 1.52s | 332 | - | 786/177 |
| Task-Specific | | | | |
| Zep (Graphit) | 2.58s | 194 | - | 2545/1189 |
| LiCoMemory | - | 75 | - | 585/217 |
| Ours | | | | |
| PLUGMEM | $0.66s + 2 \times Q32 + 1 \times 4o$ | 197 | 1604/418 | 9/0.4 |

## F.2. HotpotQA

**Setup** HotpotQA (Yang et al., 2018) is a multi-hop question answering benchmark that is widely used to evaluate multi-step retrieval and reasoning in RAG systems and agentic memory frameworks. Following the evaluation protocol of HippoRAG2 (Gutiérrez et al., 2025), we use their preprocessed subset containing 1,000 examples. For all methods, we adopt Qwen2.5-32B-Instruct (Qwen et al., 2025) as the backbone LLM and NV-Embed-v2 (Lee et al., 2025) as the embedding model, with decoding parameters max-tokens = 2048, temperature = 0.0, top-p = 1.0 During retrieval, we set top-k = 10, returning the 10 most relevant memory nodes for downstream reasoning.

**Episodic memory standardization.** PLUGMEM standardizes agent trajectories into RL-inspired episodic tuples $\langle o_t, a_t, s_t, r_t, i_t \rangle$ (observation, action, state, reward, intent/subgoal). For HotpotQA, the corpus is indexed **passively** rather than generated by an acting agent. We therefore treat each unit of corpus text as a single-step "trajectory" (i.e., $T = 1$): each episodic item contains one observation $o_1$ (the text unit being indexed), while action-related fields are not instantiated by execution. Concretely, we keep the unified tuple interface by setting $a_1$, $s_1$, $r_1$, and $i_1$ to an empty (or `N/A`) placeholder. The semantic extraction and retrieval primarily operate on the observation content.

**HotpotQA multi-hop retrieval control.** At each hop $t$, the retriever gathers a pool of candidates via the two-channel update: *i)* link-based expansion via abstract nodes, where the retriever infers a small set of abstract concepts from current query $Q_t$, matches them to high-level concept nodes in the memory graph, and expands to adjacent low-level proposition nodes via membership edges; *ii)* embedding-based retrieval from query $Q_t$ directly.

We then invoke an LLM controller to assess whether the currently retrieved candidates are sufficient to answer the question. If the controller returns `enough=true`, we terminate early and pass the current fact set to the downstream QA model. Otherwise, the controller selects a small subset of the most promising candidates to drive the next hop; in our experiments we cap this subset at top-2 candidates. To form the next-hop query, we integrate the selected candidates with the previous query. This integration can be performed by an LLM-based re-writer; however, we find that a simple concatenation of the query and selected candidates performs comparably well in practice, and we therefore adopt concatenation as the default for efficiency.

**Baselines** We consider three categories of baselines: *i) Vanilla*, including no-context inference that relies solely on the backbone model's parametric knowledge, and an oracle (gold-context) setting where the model is provided with the gold supporting context, serving as an approximate upper bound. *ii) Task-agnostic*, which are not specifically designed for knowledge-intensive QA benchmarks. Examples include a standard dense RAG pipeline that treats the input as a one-dimensional text stream, segments it into chunks using a fixed chunk size and overlap window, and performs embedding-based retrieval; and agentic memory approaches such as A-Mem (Xu et al., 2025), which structure the text stream into notes and maintain/update them via a graph-based organization mechanism. *iii) Task-specific*, tailored for knowledge-intensive QA and multi-hop retrieval, including hierarchical/tree-structured retrieval frameworks RAPTOR (Sarthi et al., 2024) and graph-oriented designs GraphRAG (Edge et al., 2025), HippoRAG2 (Gutiérrez et al., 2025), and PropRAG (Wang & Han, 2025).

**Metrics.** We report token-level Exact Match and F1 score (EM and F1 in Table 4) between the model-generated answer and the reference. Beyond standard end-task metrics such as EM and F1, we also adopt the information-theoretic measures introduced in Section 4.1 and Appendix E to quantify the *information gain* by the memory module. For one sample from benchmark, Pointwise Mutual Information (PMI) is computed as follows:

$$\text{PMI}(a^*; m \mid s) = \log_2 \frac{P_{\text{mem}}(a^* \mid s, m)}{P_{\text{base}}(a^* \mid s)} \tag{4}$$

Over a dataset, we report a global, amortized density via a ratio-of-sums:

$$\rho_{\text{global}} = \frac{\sum_i \text{PMI}(a_i^*; m_i \mid s_i)}{\sum_i |m_i|} \tag{5}$$

Concretely, we instantiate the likelihood terms in Eq. (4) using per-instance answer overlap: $P_{\text{mem}}$ is computed from the F1 score between the prediction and the reference answer; Similarly, for $P_{\text{base}}$, the prediction is answer from base agent without memory module. Following the ratio-of-sums aggregation, we report global density by normalizing the summed PMI across instances by the summed number of memory tokens.

It's worth noting that, $P_{\text{base}}$ or $P_{\text{mem}}$ can be zero for some instances, which would make Eq. (1) ill-defined (division by zero and/or $\log 0$). To stabilize the computation, we apply additive smoothing and use $\log\left(\frac{P_{\text{mem}}+\epsilon}{P_{\text{base}}+\epsilon}\right)$ when computing PMI. We find that choosing $\epsilon$ too small can lead to excessively large PMI values when $P_{\text{base}} = 0$ but $P_{\text{mem}} > 0$, which disproportionately affects the aggregate density. Therefore, in our implementation we set $\epsilon$ to **1%** of the base agent's F1 score (no memory), and use the same $\epsilon$ for both the numerator and denominator.

**Statistical Variance.** We compare RAPTOR, HIPPORAG2, A-MEM, PROPRAG, and PLUGMEM on the same evaluation set, reporting Exact Match (EM) and F1 score. Each method is run three times with different random seeds; Table 22 reports mean $\pm$ standard deviation across the three runs. Our method PLUGMEM achieves the highest EM and F1.

**Ablations.** We further conduct ablation studies over the three components of PLUGMEM. For `no-structuring`, we replace structured indexing with a chunk-based pipeline (fixed chunk size and overlap) for corpus preprocessing and

*Table 22.* Mean $\pm$ std over 3 runs per method (best in bold). Higher is better.

| Method | RAPTOR | HippoRAG2 | A-Mem | PropRAG | PlugMem |
|--------|--------|-----------|-------|---------|---------|
| EM | $54.23_{\pm2.14}$ | $59.53_{\pm0.15}$ | $44.07_{\pm0.31}$ | $57.43_{\pm0.40}$ | $\mathbf{61.27}_{\pm0.40}$ |
| F1 | $67.25_{\pm2.15}$ | $72.88_{\pm0.19}$ | $53.34_{\pm0.36}$ | $71.62_{\pm0.48}$ | $\mathbf{74.19}_{\pm0.26}$ |

evaluation; for `no-retrieving`, we populate the retrieval candidate set using randomly sampled structured memory items. As shown in Table 7, removing any single component consistently degrades both task performance and the global information-gain density. In particular, removing retrieval causes the most substantial performance drop, while removing the reasoning module markedly increases the number of memory tokens injected into context, leading to lower information gain per token.

**Cost Analysis.** In addition, we analyze the token cost of PLUGMEM on a HotpotQA subset, logging token usage during both memory construction (indexing) and evaluation-time retrieval and reasoning, and compare it against competing baselines.

As shown in Table 23, the token usage of PLUGMEM is comparable to that of other strong baselines and remains within the same order of magnitude. It is noteworthy that PLUGMEM performs a more fine-grained and structured memory construction process, enabling unified memory editing, retrieval, and reasoning in downstream stages. In particular, PLUGMEM jointly constructs three complementary memory types, i.e., episodic, semantic, and procedural, within a single pipeline, whereas most baselines extract only one or two types in isolation.

While HotpotQA predominantly evaluates semantic knowledge, the one-time cost of joint $(E/S/P)$ memory construction enables reuse across tasks and time, allowing the agent to support heterogeneous memory operations without rerunning separate extraction pipelines. Consequently, this upfront investment naturally amortizes in multi-task, continual, or long-horizon settings, potentially leading to lower total token cost in realistic deployments. All the experiments are conducted on NVIDIA A6000 Server.

*Table 23.* Cost Statistics on HotpotQA. Latency denotes retrieval latency (s), and $Q32_{in/out}$ reports input/output tokens (k).

| Method | Latency (s) | $Q32_{in/out}$ |
|--------|-------------|----------------|
| *Task-Specific* | | |
| RAPTOR | 2.0 | 331 / 40 |
| HippoRAG2 | 2.3 | 1350 / 490 |
| PropRAG | 1.8 | 1651 / 643 |
| *Task-Agnostic* | | |
| Vanilla RAG | 0.3 | 110 / 20 |
| A-Mem | 1.2 | 1778 / 261 |
| *Ours* | | |
| PLUGMEM | $1.3s + 8 \times Q32$ | 1919 / 272 |

### F.3. WebArena

**Setup.** WebArena (Zhou et al., 2024) is a realistic and reproducible web navigation benchmark consisting of 812 tasks spanning five site domains(e.g. shopping, GitLab, etc.), each task instantiated from a curated collection of 241 intent templates. In this paper, we focus on the Shopping, GitLab, and Multi-site subsets, which collectively cover (i) domain-specific, procedure-heavy interactions (Shopping, GitLab) and (ii) compositional, cross-site workflows (Multi-site). To directly test memory evolution and reuse, we adopt an online/offline split aligned with WebArena's task construction: WebArena intents are written as templates with multiple instantiations, where tasks from the same template share high-level semantics but may require different concrete execution traces. Specifically, for each template (e.g., 5 instantiations), we place one instantiation into the online set and the remaining instantiations into the offline set. To further examine cross-template knowledge adaptation, we leave template with only one task id in the offline set. The exact data split is stored in our code repository.

We first evaluate the agent on the online set, then augment the memory module with a small number of high quality

human demonstrations, and finally evaluate on the offline set using the resulting augmented memory module to measure generalization from newly acquired procedural and semantic knowledge. The agent is allowed to insert and retrieve memories as it is being evaluated on the online set, but only retrieval is allowed on the offline set. For PLUGMEM, the base agent is set to be AgentOccam (Yang et al., 2025a) with GPT-4o, a strong baseline that operates within the action space defined by WebArena and without relying on specialized memory mechanisms. For all methods, we adopt GPT-4o and Qwen2.5-32B-Instruct (Qwen et al., 2025) as the backbone LLMs and NV-Embed-v2 (Lee et al., 2025) as the embedding model, with decoding parameters max-tokens = 2048, temperature = 0.0, top-p = 1.0. For all LLM calls within PLUGMEM, we use Qwen2.5-32B-Instruct first and only delegate quries to GPT-4o if the output perplexity from Qwen2.5 is higher than a set threshold. For PLUGMEM, we record the entire action trajectory up to the maximum number of steps allowed for the web agent (20 steps in all experiments). During retrieval, we set top-k = 2, returning the 2 most relevant memory nodes for downstream reasoning.

**PLUGMEM integration with AgentOccam.** PLUGMEM is integrated with base web navigation agent for our experiments. For a given objective, PLUGMEM first retrieve memory and feed the response from the reasoning module to AgentOccam. After AgentOccam has taken the action (e.g. click [1234]), PLUGMEM generates a short description of the action into natural language form (e.g. click at My account), this creates a new episodic memory along with the new observation after the action is taken. At the end of each task, the episodic memory sequence gets inserted into PLUGMEM using the process described in Methodology and Appendix D. We slightly modified the prompt for AgentOccam by adding an instruction to follow retrieved memory entries from PLUGMEM. Please see Listing 15 for the added instructions.

**PLUGMEM incorporates human demonstration.** PLUGMEM is able to incorporate human demonstration trajectories into memory graph for warm start. In our experiment, we record human demonstration for failed tasks on the online set, then we inject these recordings for offline evaluation. This is done through 'replaying' the demo trajectory and rebuilding episodic memory sequence for insertion. For the offline set evaluation, we disable memory graph inserts to test the quality of the memory graph. Normally, for a particular task domain, we record and insert human demos in the same task domain. For the Multi-site task domain, in addition to demo on Multi-site tasks, we additionally include trajectories from other task domains to further strengthen the memory graph. In this way, we test the cross-domain retrieval and reasoning capability of PLUGMEM. Over the course of development, we collected 23 Shopping demos, 18 GitLab demos, 10 Reddit demos, 10 Map demos, and 5 Multi-site demos. These human demonstrations are available in our code repository.

**Baselines.** We consider three categories of baselines: *i) Vanilla*, base agents that does not rely on retrieval augmentation, serving as a measure of performance of the backbone LLM. We use AgentOccam (Yang et al., 2025a) as the base agent. *ii) Task-agnostic*, which are baselines that adopts retrieval augmentations not specifically designed for web navigation agents. We include a vanilla dense retrieval pipeline that stores and retrieve past interaction trajectories. We also evaluate A-Mem (Xu et al., 2025), an agentic memory system that summarizes past interactions into notes and organizes them in a graph-like structure to allow retrieval of related trajectories. For vanilla retrieval, we set top-k=1. When evaluating A-Mem, we set top-k=3 to best utilize A-Mem's ability to retrieve relevant notes. For both methods, we store the first 10 steps per task at max. *iii) Task-specific*, open-sourced agentic systems that utilizes specialized memory insertion and retrieval mechanisms. We include AWM (Wang et al., 2024b), which summarizes past interactions into workflows. To examine and compare how high quality human demonstrations could be utilized cross all methods, we add human demonstration trajectories between online and offline evaluations for all task-agnostic and task-specific baselines.

**Metrics.** WebArena evaluates task completion using functional correctness validator, i.e., programmatic checks that determine whether the execution achieves the intended goal. We use WebArena's evaluators and report the success rates on both the online and offline set. We compute PMI per task sample. We apply additive smoothing and use $\log\left(\frac{P_{\text{mem}}+\epsilon}{P_{\text{base}}+\epsilon}\right)$ when computing PMI. We set $\epsilon$ to be $1\%$ of AgentOccam's average success rate over WebArena.

**Ablations.** We further conduct ablation studies over the three components of PLUGMEM. For `no-structuring`, replace the memory graph construction pipeline with vanilla retrieval process, while keeping the reasoning module; for `no-retrieving`, we populate the retrieval candidate set using randomly sampled structured memory items. Additionally, for `no-human demo`, we omit the intermediate step where we insert high quality human demonstrations between online and offline evaluations. According to Table 8, the removal of any components would degrade agent success rates or global information density. We find that removing the retrieval component has the biggest negative impact on all metrics measured.

**Cost Analysis.** We analyze the token cost of PLUGMEM and baselines on WebArena, logging LLM token usage and

retrieval and reasoning latency. All experiments conducted on a remote server with four NVIDIA A6000 GPU cards.

As shown in Table 24, PLUGMEM exhibits a higher average token usage than task-agnostic and task-specific baselines on WebArena. This difference reflects the additional computation required for memory-to-knowledge abstraction, rather than redundant processing. We note that the latency induced by memory node retrieval is shorter than baseline methods.

In WebArena, reusable knowledge is predominantly procedural, and each task involves multiple interaction steps that must be recorded, segmented, and abstracted. Accordingly, PLUGMEM allocates additional tokens to standardizing long episodic trajectories and organizing them into subgoal-aligned procedural memory units. Compared to methods that store only raw trajectories or single static workflows, this process naturally involves more computation.

Notably, the additional tokens consumed by PLUGMEM are mostly generated using open-source models, whereas several baselines rely on a small number of calls to closed-source models, meaning that their per-token cost is substantially higher in practice. In contrast, PLUGMEM 's token usage remains within a practical range while enabling richer and reusable procedural representations.

Moreover, these costs are primarily incurred during the online phase and can be amortized across multiple task instances that share similar user intents. Once constructed, the procedural memory graph can be reused without re-running the full abstraction pipeline, leading to decreasing effective cost as task horizon and diversity increase.

*Table 24.* Cost Statistics on WebArena (k tokens per sample).

| Method | Latency (s) | $Q32_{in/out}$ | $4o_{in/out}$ |
|---|---|---|---|
| Task-Agnostic | | | |
| Vanilla RAG | - | - | - |
| A-Mem | 0.91s | - | 7431/1758 |
| Task-Specific | | | |
| AWM | - | - | 4918/1202 |
| Ours | | | |
| PLUGMEM | $0.71s + 2 \times Q32 + 1 \times 4o$ | 10587/1377 | 3128/371 |

Listing 15. Additional instruction for AgentOccam to incorporate retrieved memory.

```
==============================================================
Prompt AgentOccam Retrieved Memory Instruction
==============================================================

The retrieved memory contains relevant information from past experiences that may help
    you complete the current task. Use this memory to:
- Learn from similar past tasks and their successful strategies
- Avoid repeating previous mistakes
- Apply proven approaches that worked in similar situations
- Understand patterns and relationships that can guide your actions
When retrieved memory is provided, carefully consider how it relates to your current
    task and incorporate relevant insights into your decision-making process.

Some task may require access to Wikipedia, the only way you may access Wikipedia is
    through the url "http://<BASE_URL>:8888/wikipedia_en_all_maxi_2022-05/A/User:
    The_other_Kiwix_guy/Landing"
BASE_URL: localhost
```

# G. Analysis of the Structuring Module

The structuring module is the core component that realizes the *memory-to-knowledge abstraction* in our system. As it operates at the very upstream stage of the pipeline, its effectiveness fundamentally determines the quality of all downstream processes, including retrieval and reasoning. Given that our overall memory system consistently outperforms existing baselines, it is important to further examine whether this module itself is reliable, accurate, and robust in practice.

In this section, we focus on analyzing the behavior of the structuring module, with particular emphasis on the **knowledge extraction** step. Our goal is to understand whether the module can accurately extract decision-relevant knowledge from raw experience, and whether potential imperfections in this process affect downstream performance.

Our analysis leads to two key findings. First, we observe that modern LLMs already achieve strong performance in knowledge extraction, resulting in a very low rate of critical factual errors that could affect final decisions. Second, to account for potential imperfections, we explicitly design a set of **fallback mechanisms** that mitigate extraction errors. These mechanisms provide substantial recovery capability, ensuring that even when extraction is incomplete or imperfect, the system remains robust and maintains strong performance.

### G.1. Design of the Structuring Module

We design the structuring module based on three core steps:

*i)* **Standardization.** Raw interaction trajectories are first converted into standardized episodic units (state, action, subgoal, reward), enabling consistent processing across heterogeneous tasks.

*ii)* **Knowledge Extraction.** Semantic and procedural knowledge are extracted into structured forms:

- Semantic knowledge is represented as propositions with associated concepts, forming a semantic graph.

- Procedural knowledge is represented as intent–prescription pairs, forming a procedural graph.

*iii)* **Graph Construction and Linking.** The extracted knowledge is organized into a unified memory graph:

- Relational edges connect concept–proposition and intent–prescription pairs.

- Provenance edges link each knowledge unit back to its source trajectory, enabling grounding and verification.

To ensure robustness against potential upstream errors, we further introduce three layers of fallback mechanisms:

- **Comprehensive retrieval:** Instead of relying on a single extracted memory, the system retrieves a ranked set of candidates.

- **Aggregation over experiences:** Knowledge is induced from multiple retrieved candidates, reinforcing consistent signals while suppressing noise.

- **Episodic fallback:** When inconsistencies arise, the system traces back to raw episodic memory (trajectory logs) to resolve conflicts.

These mechanisms ensure that potential imperfections in knowledge extraction do not propagate and degrade downstream performance.

### G.2. Evaluation Protocol for Structuring Quality

To systematically evaluate the quality of the structuring module, we introduce a categorization framework that classifies extracted knowledge into three types: *i)* **decision-sufficient**, *ii)* **helpful but imperfect**, and *iii)* **misleading**.

This categorization provides a principled way to distinguish between successful and failed knowledge construction. Notably, the **"helpful but imperfect"** category is defined under a strict criterion: even minor omissions of task-relevant details, despite otherwise correct extraction, are classified as imperfect.

For LongMemEval and HotpotQA, we use three strong LLM evaluators (GPT-5.1, DeepSeek-V3, Sonnet-3.7) to assess all instances, complemented by manual verification on 30 samples and all failure cases. For WebArena, we conduct full manual evaluation due to the limited reliability of current VLM-based evaluation in complex web environments.

### G.3. Empirical Analysis Across Benchmarks

**(1) Structuring is near-perfect on knowledge-centric tasks.**

On HotpotQA, we observe that **misleading cases (hallucinations or factual errors) are below 2%**, while **imperfect cases are below 5%**, consistently across both LLM-based evaluation and manual verification. This is because the knowledge units in this benchmark are fine-grained factual statements, which modern LLMs can extract reliably. Errors only occur in rare cases involving long contexts and multi-step reasoning.

Importantly, we find **zero cases where extraction errors lead to incorrect final answers**. Instead, failures are primarily due to the intrinsic difficulty of the benchmark, which is intentionally designed to make retrieval highly challenging. (In fact, we observe that our system already approaches **near-oracle performance given the retrieved context**, indicating that retrieval is already performing strongly.) This indicates that, in such settings, the structuring module is highly reliable and not a bottleneck.

**(2) Imperfect extractions are largely recoverable.**

Under our strict evaluation protocol, LongMemEval exhibits **less than 1% misleading cases** and **less than 14% imperfect cases**. We want to emphasize that this number largely reflects strict completeness requirements rather than critical errors. In LongMemEval, there exists a significant mismatch between the format of the stored memory and the form of the query. For example, a dialogue may contain a statement such as "I went to church on a specific day in December," while the corresponding question may ask "what faith-related activities did the user engage in during December." This requires an additional level of abstraction that is not explicitly present in the raw interaction. In standard knowledge extraction, such transformations are not always directly inferred (e.g., mapping "going to church" to "faith-related activity"). Moreover, our definition of incompleteness is intentionally strict: even if most relevant information is correctly extracted, the omission of a single potentially useful element leads to classification as imperfect. This holds even when the missing information can be recovered later through retrieval aggregation. As a result, the reported imperfect cases largely reflect this strict evaluation criterion rather than substantial deficiencies in extraction quality.

We observe that approximately **65% of imperfect cases are recovered through retrieval aggregation fallback**, meaning that missing information is compensated by other retrieved knowledge. Furthermore, **all factual errors are corrected by fallback mechanisms**. This demonstrates that the system is robust to incomplete extraction.

**(3) Robustness under procedural tasks.**

On WebArena, **misleading cases are below 15%**, mainly due to two well-known challenges in web agent research: *i)* accurately understanding and summarizing complex web environments, and *ii)* reliably estimating whether a trajectory has successfully completed a task (i.e., reward modeling). Both are active research problems in current foundation models. Therefore, these errors primarily reflect limitations of the underlying models, rather than issues with the structuring module itself.

Even in these cases, the system remains robust. Through retrieval aggregation, the agent can compare multiple experiences and avoid being misled by individual errors. For example, in the GitLab subset, tasks with misleading memory still achieve approximately **46% success rate**, compared to around 39% without memory.

Further analysis shows that the primary bottleneck lies in the base agent's interaction incapability with the environment rather than the structuring module. In many failure cases, the structuring module functions correctly, but the agent struggles to correctly interpret web pages and identify task-relevant information, or fails to execute actions reliably in the environment. These issues are tied to the agent's perception, decision-making, and action execution capabilities, as well as the underlying foundation model's understanding of the task, which leads to their failure to generate useful experience.

Overall, these results demonstrate that the structuring module is both **accurate and robust in practice**. It produces high-quality knowledge representations with low error rates, and, importantly, its design ensures that potential imperfections are effectively mitigated through fallback mechanisms. As a result, the module does not hinder system performance and serves as a reliable foundation for downstream memory utilization.

## H. LLM Routing Mechanism and PLUGMEM Usage Cost Analysis

In designing memory systems for LLM agents, it is important to recognize the inherent trade-off between **computational cost** and **performance**. In general, increasing computational resources, such as using larger models, scaling up inference time, or introducing multi-agent or search-based mechanisms, can lead to improved performance. However, such gains often come at a disproportionately high cost, making them inefficient and less practical for real-world deployment.

In PLUGMEM, we explicitly aim to move beyond this conventional trade-off. Rather than improving performance by simply increasing computation, our goal is to **shift the Pareto frontier**, achieving better performance at comparable or lower cost. That is, we seek a design where the same level of computational budget yields stronger results, and where efficiency is treated as a first-class objective rather than a secondary consideration.

To this end, we introduce a simple yet effective mechanism: **LLM routing**. Before processing a given memory-related query, the system *adaptively* selects the most appropriate language model based on the estimated difficulty and requirements of the task. For simpler operations, such as basic information extraction, smaller models are sufficient and more efficient. In contrast, more challenging tasks, such as complex reasoning over long contexts or processing rich web observations, are assigned to stronger models with higher capacity. This adaptive allocation allows the system to match computational resources to task complexity, avoiding unnecessary overhead. In practice, this adaptive routing can be implemented with lightweight heuristics. For example, short and structurally simple queries are typically handled by smaller models, while longer inputs such as web environment or context-rich observations are routed to more capable models. Additionally, queries can be grouped by type: by sampling a small subset of similar tasks and evaluating their difficulty with a stronger model, the system can infer an appropriate routing strategy for the entire group. Such designs enable both simple and more advanced versions of routing, while keeping the overhead of the adaptation process itself minimal.

Importantly, for fair comparison with prior work, all baselines are evaluated using a **uniform strong-model setting**, where the same high-capacity model is used throughout the entire pipeline regardless of task difficulty. In contrast, PLUGMEM with routing selectively invokes smaller models when appropriate. From a cost perspective, especially in self-hosted settings, calls to smaller open-source models (e.g., 32B-scale models) incur negligible cost compared to large proprietary models, making this design significantly more efficient.

We summarize the token usage cost and latency across methods in Tables 21, 23, and 24 (see the Additional Experimental Details section F). From these results, we highlight three key observations: *i)* Under a uniform model setting, PLUGMEM achieves superior performance while maintaining token consumption at a similar scale to competitive baselines, often not being the most expensive despite delivering better results and richer memory representations; *ii)* With adaptive routing enabled, PLUGMEM consistently uses the fewest closed-model tokens; *iii)* On simpler benchmarks, routing reduces closed-model usage by **two to three orders of magnitude**. *iv)* During the memory graph retrieval stage, the system latency remains **comparable to other memory modules**.

Overall, these results demonstrate that our design not only improves performance, but does so in a **cost-efficient manner**. The integration of LLM routing within the memory module enables substantial reductions in computational cost while preserving, and often enhancing, effectiveness.

## I. Task Adaptation via Integrating Task-Specific Heuristics on Top of PLUGMEM

### I.1. LongMemEval

**Reflective Memory Management.** RMM (Tan et al., 2025) introduces a novel reflective mechanism for long-term memory management. In general, RMM employs Prospective Reflection for knowledge extraction and memory updating, and Retrospective Reflection to refine memory retrieval in an online reinforcement learning manner.

Inspired by these ideas, we perform task adaptation of PLUGMEM to LongMemEval by incorporating the Prospective Reflection mechanism from RMM for semantic memory extraction and semantic node updating. Specifically, for memory extraction, we adopt the memory extraction prompt proposed in RMM to distill semantic memory for PLUGMEM. Compared to the original semantic memory extraction prompt used in PLUGMEM (Listing 5), the task-adapted version places greater emphasis on extracting user-specific personal information and includes in-context examples to better align with the LongMemEval setting. For memory updating, we follow the reflective updating strategy introduced in RMM to update semantic memory within the memory graph. Specifically, unlike the original PLUGMEM, which directly inserts newly extracted memory, the task-adapted version first retrieves the Top-K most similar existing semantic memories from the

memory graph by computing cosine similarity between semantic memory embeddings and ranking them accordingly. The LLM is then queried to decide whether the new memory should be added as a new node or merged with an existing similar node. The prompt to extract memory and update memory is shown in Listing 16 and Listing 17.

**Experiment.** We conduct experiments on the multi-session subset of LongMemEval, with results shown in Table 25. The multi-session subset is among the most challenging portions of LongMemEval, as it requires the agent to answer questions that primarily involve counting the occurrences of a series of similar events across multiple sessions. This setting not only places significant demands on the retrieval module of the memory module to retrieve multiple gold memories simultaneously, but also challenges the structuring and reasoning module to generate precise semantic memories and perform accurate counting over them. The task-adapted version outperforms the original method for two main reasons: first, the task-adapted memory extraction prompt produces more precise semantic memories; second, the reflective updating strategy prevents redundant and repetitive semantic memories from being inserted into the memory graph, thereby enabling more efficient retrieval.

*Table 25.* Task Adaptation on the multi-session subset of LongMemEval.

| Method | Acc. (Multi-session) | PMI | #Tok$_{Avg.}$ | Info. Density |
|---|---|---|---|---|
| PLUGMEM | 75.2 | 6.72 | 224.1 | 2.9e-2 |
| PLUGMEM $_{adapt}$ | 78.2 | 7.02 | 228.2 | 3.1e-2 |

**Listing 16.** Prompt template to extract semantic memory for task adaptation.

```
============================================================
Prompt Extract_Memory
============================================================

Task Description: Given a session of dialogue between User and Agent, extract the
    personal summaries of User. Ensure the output adheres to the following rules:
Output results in OUTPUT format. The top-level tittle is "### Memory". The value should
     be a list of dictionaries, where each dictionary has the key "Summary":
- summary: A concise personal summary, which captures relevant information about User
    experiences, preferences, and background, across multiple turns.
If no personal summary can be extracted, return NO_TRAIT.
Example:
INPUT:
Turn 0:
- User: Did you check out that new gym in town?
- Agent: Yeah, I did. I am not sure I like the vibe there, though.
Turn 1:
- User: What was wrong with it?
- Agent: The folks there seemed to care more about how they looked than working out. It
     was a little too trendy for me. I am pretty plain.
Turn 2:
- User: Ah, got it. Well, maybe one of the older gyms will work out better for you-or I
     guess you could get that treadmill you were talking about before. Are you leaning
    one way or the other yet?
- Agent: I am leaning towards the treadmill. I think it will work better for my
    lifestyle. I just do not know which type to get. There are so many choices out
    there. Do you use a treadmill at your gym? Do you have a suggestion for a home one?
Turn 3:
- User: I usually just lift weights there, to be honest. But I think I have heard good
    things about the NordicTrack?
- Agent: Yeah, I have heard good things about that, too. I like the idea of a multi-
    exercise piece of equipment. As long as the weather is not too bad, then I prefer
    to go for a run. But since it rains quite a bit here, I like the idea of an inside
    option. How is the weather in New England?
OUTPUT:
### Memory:
1. **Summary:** User asked about a new gym in town and suggested older gyms or a
    treadmill as alternatives.
```

```
2. **Summary:** User usually lifts weights at the gym rather than using a treadmill.
3. **Summary:** User has heard good things about the NordicTrack treadmill.
Task: Follow the OUTPUT format demonstrated in the example above and extract the
    personal summaries for User from the following dialogue session.
Input: {episodic_memory}
Output:
```

Listing 17. Prompt template to update semantic memory for task adaptation.

```
=============================================================
Prompt Update_Memory
=============================================================

Task Description: Given a list of history personal summaries for a specific user and a
    new and similar personal summary from the same user, update the personal history
    summaries following the instructions below:
Input format: Both the history personal summaries and the new personal summary are
    provided in JSON format, with the top-level keys of "history_summaries" and "
    new_summary".
Possible update actions:
- Add: If the new personal summary is not relevant to any history personal summary, add
     it.
Format: Add()
- Merge: If the new personal summary is relevant to a history personal summary, merge
    them as an updated summary.
Format: Merge(index, merged_summary)
Note: index is the position of the relevant history summary in the list. merged_summary
     is the merged summary of the new summary and the relevant history summary. Two
    summaries are considered relevant if they discuss the same aspect of the user
    personal information or experiences.
Do not include additional explanations or examples in the output-only return the
    required action functions.
Example:
INPUT:
History Personal Summaries:
- {{"history_summaries": ["SPEAKER_1 works out although he does not particularlyenjoy
    it."]}}
New Personal Summary:
- {{"new_summary": "SPEAKER_1 exercises every Monday and Thursday."}}
OUTPUT ACTION:
Merge(0, SPEAKER_1 exercises every Monday and Thursday, although he does not
    particularly enjoy it.)
Task: Follow the example format above to update the personal history for the given case
    .
INPUT:
History Personal Summaries: {history_summaries}
New Personal Summary: {new_summary}
OUTPUT ACTION:
```

### I.2. HotpotQA

**Inspiration from HippoRAG2.** Our retrieval design is partly inspired by HippoRAG2, which demonstrates that introducing an explicit graph structure and a controlled multi-hop traversal can outperform purely similarity-based retrieval on multi-hop QA. In particular, HippoRAG2 separates *recognition* (filtering/selecting relevant structured units) from *expansion* (graph-based propagation to retrieve supporting evidence), reducing spurious hops and improving evidence quality. PLUGMEM adopts this high-level principle but generalizes it to typed agent memory. We construct semantic and procedural memory graphs and use abstract nodes (concepts/intents) as routing signals to activate specific proposition/prescription nodes, enabling abstraction-to-specificity traversal. Moreover, analogous to HippoRAG2's recognition step, PlugMem employs an LLM-based controller to *i)* select memory types (GETMODE), *ii)* propose next-hop abstract signals (GETPLAN), and *iii)* decide early stopping or focus candidates under a fixed budget (MULTIHOPCTRL). Finally, we integrate a two-channel

candidate update: link-based expansion through abstract nodes and embedding-based retrieval from a refined query, followed by reranking/pruning and optional reasoning-based compression.

**Test-time Scaling.** Beyond the default configuration, PLUGMEM admits two practical test-time "knobs" that consistently improve performance. First, we can increase the hop limit $T_{\max}$, effectively scaling the amount of retrieval computation at inference time. This allows the retriever to chain additional abstraction–specificity steps and is analogous to test-time scaling in reasoning-centric methods. We experiment with a subset on HotpotQA, when increasing the multi-hop limit $T_{\max}$ from 2 to 4, we observe a performance gain from 66.00/74.73 to 69.00/78.11 (measured in Exact Match/F1-Score).

Secondly, when forming the next-hop query $Q_{t+1}$, we optionally replace naive concatenation of $(Q_t,$ selected facts) with an LLM-based query synthesizer that rewrites a focused and self-contained query. Empirically, synthesis-based refinement reduces lexical drift and redundancy, and helps the embedding channel retrieve more targeted evidence in later hops. We treat both mechanisms as controllable components within the retrieval controller, enabling a smooth accuracy–cost trade-off depending on the task budget.

### I.3. WebArena

Prior work designed specifically for WebArena often incorporates manually engineered *domain-level instructions* (or "tips") into the agent's input context. A common pattern in task-specific adaptations, such as AWM (Wang et al., 2024b), SteP (Sodhi et al., 2024), and more recently ColorBrowserAgent (Wang et al., 2026), is the inclusion of explicit guidance that biases the agent toward particular interaction patterns within a site domain (e.g., navigation conventions or preferred UI elements). In contrast, in our main experiments we deliberately avoid injecting such handcrafted instructions or explicitly defined action policies. Instead, we rely solely on PLUGMEM 's ability to automatically store, retrieve, and re-apply both agent-generated and human demonstration trajectories as structured memory, allowing behavior to emerge from experience rather than prompt engineering.

**Further task adaptation** We additionally conduct a targeted case study to examine whether PLUGMEM can benefit from manual task adaptation when such domain-specific constraints are known. We identify a subset of WebArena tasks that remain challenging for PLUGMEM and introduce a small set of manually designed instructions into the reasoning module. Specifically, we observe that PLUGMEM occasionally instructs the agent to navigate via category dropdown menus when the correct strategy in WebArena is to use the search bar. By incorporating the additional guidance shown in Prompt 18, we resolve several previously failing cases (see Example 26) by explicitly conditioning the agent's action selection. WebArena uses strict, programmatic success criteria but concise queries, where otherwise reasonable strategies may be scored as incorrect. Although PLUGMEM successfully retrieves relevant procedural knowledge for tasks such as "searching for teeth grinding mouth guards," it does not, by default, infer that certain plausible actions are disallowed by the benchmark's implicit rules. This case study demonstrates that PLUGMEM natively supports fine-grained manual conditioning when necessary, while remaining effective without task-specific prompting in the general case.

Listing 18. Additional instructions for the agent and reasonig module for further task-adaptation

```
============================================================
Prompt WebArena Task-specific Instructions
============================================================

1. When asked about a kind of product (not a category), search for the product using
   the search bar instead of using the category dropdown menu.
2. When asked about a category of product, access the dropdown menu through hover
   actions.
```

*Table 26.* Example of PLUGMEM with manual instructions.

**WebArena Manual Instruction Effect**

**Query.** *" What is the price range of teeth grinding mouth guard in the One Stop Market? "*

**PLUGMEM with Manual Instruction.**
▷ **Reasoning**
The user's question is about determining the price range of wireless earphones in the One Stop Market. According to the guidelines, when asked about a specific kind of product (e.g., "wireless earphone"), the agent must use the search bar to locate the product instead of navigating through dropdown menus. The current observation shows that the search bar is present but appears disabled. However, based on the relevant experience provided, attempting to use the search bar is still recommended, as its functionality may not be accurately represented by its appearance.
▷ **Final Information**
1. Use the search bar to type "wireless earphone" and press Enter to locate relevant products.
2. Once the search results are displayed, review the prices of the listed wireless earphones to determine the price range.
3. If the search bar does not function, report the issue and consider alternative methods to locate the product, such as contacting support or exploring other navigation options.

**AgentOccam Action 1: type [1913] [wireless earphone] [1]**. (agent uses the search bar)

...

**PLUGMEM without Manual Instruction.**
▷ **Reasoning**
The user's question is about finding the price range of teeth grinding mouth guards in the One Stop Market. The observation does not provide any direct information about teeth grinding mouth guards or their prices. However, the relevant experience suggests that the agent should use the hierarchical menu structure to navigate to the appropriate category (likely under "Beauty & Personal Care" or "Health & Household") and then refine the search to locate the specific product type. If a price filter is available, it can be applied to determine the price range.
▷ **Final Information**
To answer the question, the agent should:
1. Hover over the hierarchical menu structure, starting with categories like "Beauty & Personal Care" or "Health & Household."
2. Look for subcategories related to oral care or dental care.
3. Navigate to the relevant subcategory (e.g., "Oral Care" or "Mouth Guards").
4. Review the products listed in the subcategory to identify teeth grinding mouth guards and note their prices.
5. If a price filter is available, apply it to determine the price range more efficiently.
6. Summarize the price range of the identified products.

**AgentOccam Action 1:**
**hover [1534]**. (agent operates the dropdown menu (WRONG!))

...

