# OpenReview forum: "PlugMem: A Task-Agnostic Plugin Memory Module for LLM Agents"
_ICML.cc/2026/Conference — ICML 2026 regular_

### Official Review · Reviewer_RFWh · 2026-02-18

**Soundness:** 3
**Presentation:** 3
**Significance:** 3
**Originality:** 2
**Overall Recommendation:** 4
**Confidence:** 3

**Summary:**

This work proposes PlugMem, a task-agnostic module to enhance the generalization of memory for reasoning performance for LLM agents. The contribution is a framework that trasnform raw trajectories into higher-level abstractions with a graph format to improve the generalizability of agent memories.

**Compliance With Llm Reviewing Policy:**

Affirmed.

**Final Justification:**

The additional case study seems rational to me, I have updated score accordingly.

**Key Questions For Authors:**

How the graphs are built in the work, can the authors say more about it?

**Limitations:**

It is suggested the authors include some limitations about this proposed method.

**Strengths And Weaknesses:**

Strengths
1. The writing is clear and eay-to-follow.
2. The motivation is clear and strong.
3. The experimental performance is significant in different settings, with fair settings.
4. The method that adopts the structuring-retrieval-reasoning pipeline, with the trasnformation of raw-level info to high-level more abstracted info to improve transferability, seems sound.

Weaknesses
1. One confusion for me is the experiment setting: What datasets the method reply on to build the graph (e.g., $G^P$), is it using the same dataset as the evaluation dataset, for instance, HotpotQA for multihopQA or uses a uniform external dataset to extract the graph and more abstract knowledge? The authors claim task-agnostic memory, if using the same dataset as the one to be evaluated, it may not be task-agnostic. In this case, the performance improvement may be arising from the graph transformation (high-order relational data) with more-abstract knowledge. Hence I feel **task-agnostic** might be a bit overclaim.
2. The idea of using high-level abstraction of experiences (or memory) is not new, many previous work explore this idea to enhance reasoning, such as RAPTOR in ICLR2024, Memoria in ICML2024, MemTree in ICLR2025, ExpeL in AAAI2024 etc. The nolvety lies in the incorporation of multiple graphs.
3. The contribution in introduction section says about "released code", but I cannot find it.

---

> ### Author Rebuttal · Authors · 2026-03-27
>
> We thank you for the positive assessment of our work, particularly regarding clarity, motivation, and empirical performance. Based on the scores and comments, we understand that the primary concern may lie in the novelty and positioning of our contribution. We therefore focus on clarifying this point, and also address several related misunderstandings that may have contributed to this concern.
>
> ### On experimental setting and task-agnostic memory
> We believe the concern regarding “task-agnostic” memory stems from a different interpretation of memory-centric benchmarks.
>
> In benchmarks such as LongMemEval, memory is **not an external dataset**, but an intrinsic part of the task definition. The agent is required to utilize past interactions (e.g., user history or trajectories) to answer future queries. Therefore, organizing memory from such interaction data follows the **standard evaluation protocol** rather than introducing data leakage. The evaluation setup is the same as in previous work that you mentioned (e.g., RAPTOR).
>
> For environments with explicit generalization protocols (e.g., WebArena), we strictly separate **memory construction** and **evaluation**. Memory is built from prior interaction trajectories, while evaluation is conducted on disjoint tasks (e.g., online/offline settings), ensuring no data leakage. We also include multi-site settings, where memory constructed in one environment must transfer to others, demonstrating cross-environment knowledge generalization.
>
> More broadly, “task-agnostic” refers to the **transferability of the memory mechanism**, rather than the independence of memory data from the task. Concretely, the same memory module must operate across fundamentally different types of memory:
>
> - **Semantic memory**, where the system accumulates and reuses factual information (e.g., conversational history in LongMemEval or wikipedia pages in HotpotQA);
> - **Procedural memory**, where the system extracts and reuses action strategies from trajectories (e.g., WebArena).
>
> These impose qualitatively different requirements on memory representation and usage. Designing a single module that works effectively across both is therefore a non-trivial problem.
>
> ### On novelty and positioning
>
> We clarify that the core contribution is not simply the use of abstraction or multiple graphs, but addressing:
>
> > Can a memory module be task-agnostic, i.e., transferable across heterogeneous memory regimes without task-specific redesign?
>
> To the best of our knowledge, **prior work is typically evaluated either on factual/knowledge-intensive benchmarks or on procedural/environment-based tasks, but not both within a unified framework.** In contrast, PlugMem is **the very first task-agnostic, plugin memory module that can be attached to arbitrary agents without modification regardless of what task they perform**. We evaluate the same module unchanged across heterogeneous benchmarks (LongMemEval, HotpotQA, and WebArena) spanning distinct memory regimes (semantic vs. procedural), yet PlugMem consistently achieves strong performance and efficiency gains. We will revise the paper to make this positioning more explicit.
>
> ### On how the memory graphs are constructed
> We thank the reviewer for this question. The graph construction process is described in Section 3.1 (Structuring Module), where we provide the design rationale (Table 2), a step-by-step illustration of the structuring process (Figure 2), and a prototypical example of the resulting memory graph (Figure 3, left). Additional implementation details are provided in Appendix C.2.
>
> The graph construction follows a three-stage process:
>
> 1. **Standardization**: Raw interaction trajectories are converted into standardized episodic units (state, action, subgoal, reward) for unified processing.
> 2. **Knowledge extraction**:
>    - Semantic knowledge is extracted as propositions with associated concepts, forming a semantic graph.
>    - Procedural knowledge is extracted as intent–prescription pairs, forming a procedural graph.
> 3. **Graph construction and linking**:
>    - Relational edges connect concept–proposition and intent–prescription pairs.
>    - Provenance edges link propositions and prescriptions to their source trajectories, enabling grounding and verification.
>
> We will further highlight these components, add pseudocode for clarity, and improve cross-referencing to make the construction pipeline easier to follow.
>
> ### On code availability
> Code is included in the supplementary material under the double-blind policy; we will make this clearer.
>
> ### On limitations
> We will include a dedicated discussion of limitations in the final version.
>
> We hope these clarifications address your concerns, particularly regarding originality, and help better reflect the contribution of our work. We would appreciate your reconsideration of the overall assessment based on our responses, and are happy to provide further clarification if needed.

---

> > ### Author Rebuttal · Reviewer_RFWh · 2026-04-02
> >
> > The author's response has partially address my concern. But I do believe that PlugMem should at least compare with one baseline that also utilize high-level abstracted memory to demonstrate the effectiveness of this approach, and it would be better to use a case-study of using the structuring module can indeed help where PlugMem succeeds and the other method fails. Currently the direct comparison with another more-relevant approach seem missing in the draft. I am glad to reconsider my evaluation with the new results.

---

> > > ### Author Response · Authors · 2026-04-08
> > >
> > > We thank you for actively engaging with our work! Based on your latest comment, we understand that your remaining question is whether our method has been sufficiently compared against **relevant high-level abstracted memory approaches**, and whether there is **clear evidence (e.g., case studies)** demonstrating the benefit of our structuring module over such methods. Therefore, we report analysis to address it below.
> > >
> > > ### Clarifying Comparison with Abstracted Memory Baselines
> > > We note that our experimental setup already includes **A-Mem**, which is a representative abstraction-based memory method that incrementally summarizes experience into higher-level notes. This provides a direct comparison with a "high-level abstracted memory" approach across all three benchmarks. We will better highlight its role in our evaluation in the revision.
> > >
> > > In addition, following your mention of RAPTOR in the first round, we further include and analyze RAPTOR in the rebuttal, which represents a complementary abstraction paradigm based on **hierarchical summarization**. Due to its design, RAPTOR is applicable only to semantic memory settings, and we therefore evaluate it on LongMemEval and HotpotQA, because WebArena majorly assesses procedural memory organization.
> > >
> > > These two baselines cover both:
> > > - Flat summarization-based abstraction (A-Mem)
> > > - Hierarchical abstraction (RAPTOR)
> > >
> > > allowing us to assess how PlugMem compares to existing abstraction-based memory designs.
> > >
> > > ### Key Finding
> > > Across comparisons, we observe a consistent pattern:
> > >
> > > > The performance difference is not due to abstraction alone, but to how memory is structured and retrieved.
> > >
> > > ### Case Study Evidence
> > > To further support this, we provide concrete case studies illustrating where PlugMem succeeds while abstraction-based baselines fail.
> > >
> > > - **PlugMem vs. A-Mem (LongMemEval)**:
> > > For a question asking *"How many days did I spend participating in faith-related activities in December?"*, A-Mem summarizes the interaction into a *general* note about volunteering and community service, but omits the specific date (December 10th) and its relevance to faith-related activity. In contrast, PlugMem extracts explicit propositional knowledge that preserves the event and its temporal detail, enabling correct reasoning over the query.
> > > - **PlugMem vs. A-Mem (HotpotQA)**:
> > > Given a passage about the Southern State Parkway, A-Mem produces a *brief* summary of the main description, but omits the route designation detail (its relation to NY 908M). PlugMem instead extracts multiple propositions, including this detail, which is essential for answering compositional, multi-hop questions.
> > > - **PlugMem vs. A-Mem (WebArena)**:
> > > In a task requiring sorting products by price within a specific category, A-Mem retrieves past trajectories that are *only topically related but fail to identify the correct action sequence* (e.g., navigating to the category and applying "Sort by Price"). PlugMem retrieves procedural knowledge that explicitly captures these reusable action steps, leading to successful execution.
> > > - **PlugMem vs. RAPTOR (LongMemEval)**:
> > > For a query asking *"What was my last name before I changed it?", *RAPTOR's recursive abstraction produces multiple layers of summaries where the key fact ("changed last name from Johnson to Winters") becomes diluted in higher-level summaries. As their retrieval tends to rely on higher-level abstractions, this specific fact is not reliably retrieved. In contrast, PlugMem extracts this information as a standalone propositional knowledge unit, ensuring it remains directly accessible for answering the query.
> > > - **PlugMem vs. RAPTOR (HotpotQA)**:
> > > For a multi-hop question linking a race winner to their role as a co-commentator, RAPTOR organizes information into hierarchical summaries where event-level details (e.g., winners of the 2013 6 Hours of Silverstone) and role-based facts (e.g., Allan McNish as a BBC commentator) are grouped into separate clusters. This weakens the explicit connection needed for multi-hop reasoning. In contrast, PlugMem extracts both facts as explicit propositions and preserves their entity-level linkage, enabling the retriever to jointly access the relevant information and support correct reasoning.
> > >
> > > These examples consistently show that summarization-based abstraction (A-Mem) and hierarchical abstraction (RAPTOR) may lose or dilute information that is not central to the main topic but is crucial for downstream queries, while PlugMem preserves such information through knowledge extraction and structured representation.
> > >
> > > We will include these case studies and analyses in the revision to better illustrate the differences between PlugMem and prior abstraction-based approaches. We thank you again for this suggestion, which has helped us strengthen the empirical evidence and clarify our contributions.
> > >
> > > We believe these additions address your remaining question. We would like to kindly request your re-evaluation of our work accordingly. Thank you!

---

### Official Review · Reviewer_g3ba · 2026-03-12

**Soundness:** 2
**Presentation:** 3
**Significance:** 3
**Originality:** 2
**Overall Recommendation:** 3
**Confidence:** 3

**Summary:**

This paper introduces PLUGMEM, a task-agnostic plugin memory module for LLM agents that organizes agent experience around propositional and prescriptive knowledge units rather than raw episodic traces. The module consists of three components: a structuring module that standardizes heterogeneous episodic memories and induces propositional and prescriptive knowledge; a retrieval module that selects task-relevant knowledge from structured memory graphs via abstraction-aware retrieval; and a reasoning module that further adapts retrieved knowledge into actionable guidance for the base agent.

**Compliance With Llm Reviewing Policy:**

Affirmed.

**Final Justification:**

Thanks to the authors for their response. The claim that "prior work has not demonstrated strong performance across both settings, as existing approaches are typically evaluated in only one" represents a very weak contribution. This can only be regarded as empirical coverage, rather than methodological novelty. The fact that they are tested on only one setting cannot be used to prove that they are not plug-and-play. Furthermore, the authors have not provided experimental validation to support this core claim of plug-and-play capability. Therefore, I maintain my decision of weak reject.

**Key Questions For Authors:**

see weaknesses

**Limitations:**

yes

**Strengths And Weaknesses:**

# Strengths
- comparing memory systems that differ in both effectiveness and efficiency. The derivation connecting PMI to KL-divergence under the one-hot assumption (Appendix D.4) is theoretically sound and elegantly presented.
- The authors release code and experimental results, and provide detailed prompts and implementation details in the appendices. This level of transparency is commendable and should be encouraged.

# Weaknesses
- The structuring module relies heavily on LLM-based extraction of propositions, concepts, intents, and prescriptions. However, the paper provides no analysis of extraction quality，no discussion of how errors propagate through the system.
- There is an issue of notation inconsistency: the standardized tuple presented in Figure 2 does not align with the one described in line 196.
- While the paper outlines the update and delete mechanisms, I have not fully grasped the specifics of their implementation process.
- I have concerns regarding the novelty of this work. There is already a significant body of work on multi-memory architectures (episodic, semantic, and procedural memory), and PropRAG has also introduced a paradigm using propositions as the fundamental retrieval unit. In the comparison of related work (Table 1), the authors have overlooked similar studies; thus, this work appears to be more of an incremental innovation.
- In Table 3, why does PLUGMeM use the Q32+4o approach while the baseline uses only 4o?

---

> ### Author Rebuttal · Authors · 2026-03-27
>
> We appreciate your thoughtful feedback and for recognizing the strength of our theoretical analysis and experimental transparency. We note that the work is rated positively in terms of soundness, presentation, and significance. We address all concerns below, with a focus on clarifying originality, which appears to be the main remaining issue underlying the overall assessment.
>
> ### Novelty and relation to prior work
> We thank you for raising the question regarding novelty. A central challenge in agent memory is how to design an effective general memory module that can be **task-agnostic and transferable across heterogeneous settings**. Such an effective task-agnostic memory module has significant benefits, as it can be used to enhance any agent regardless of the specific tasks to be performed. PlugMem is the very first task-agnostic memory module, thus not an incremental innovation.
>
> While there exists a line of work on multi-memory systems (e.g., episodic, semantic, and procedural memory), **these approaches have not demonstrated that a single memory module can perform effectively across fundamentally different memory regimes**. Prior work is typically evaluated either on factual/knowledge-intensive tasks or on procedural/skill-based environments, but not both.
>
> In contrast, our contribution is a **task-agnostic memory-to-knowledge abstraction framework**, which **enables a single module to operate across heterogeneous environments without modification**. We demonstrate this by applying the same module across both knowledge-intensive benchmarks (e.g., HotpotQA, LongMemEval) and procedural environments (e.g., WebArena).
>
> To the best of our knowledge, **prior work has not demonstrated strong performance across both settings**, as existing approaches are typically evaluated in only one.
>
> We also thank you for pointing out additional related work. Some mentioned approaches (e.g., PropRAG) are already included in our empirical evaluation (Table 5) as baselines. Table 1 highlights representative categories rather than exhaustively listing all methods; we will expand it for more comprehensive coverage.
>
> ### Extraction quality and error propagation
> We agree that extraction quality is important. Our framework mitigates error propagation through:
>
> - **Comprehensive retrieval:** A ranked list of candidates is retrieved instead of relying on a single extraction.
> - **Aggregation over experiences:** Knowledge is induced from multiple candidates, reinforcing consistent patterns and down-weighting noise.
> - **Episodic fallback:** When inconsistency risk arises, the system traces back to episodic memory (trajectory logs) to resolve conflicts.
>
> Due to space constraints, we will include case studies in the final version showing recovery from imperfect extraction.
>
> ### Model usage in Table 3 (Qwen-32B + GPT-4o)
> PlugMem uses a lightweight model (Qwen-32B) with GPT-4o via routing for efficiency, not performance gain. This is because some submodules (e.g., formatting or lightweight transformations) do not require strong reasoning.
>
> Importantly, **Qwen-32B is weaker than GPT-4o on both factual and reasoning benchmarks**. Thus, our approach does not introduce stronger models, but replaces part of the computation with a weaker one to reduce cost. In contrast, **all baselines use GPT-4o throughout, a strictly stronger setup**. Despite this, PlugMem achieves superior performance, indicating improvements come from the framework rather than model capacity. We will clarify this.
>
> ### Update and delete mechanisms
> We apologize for the lack of clarity. These mechanisms are described in Appendix C.5. While evaluated on existing benchmarks (e.g., HotpotQA), the main results often fail to reflect the advantages of update and delete operations, as their contributions are largely overshadowed by other functionalities of the memory module. Therefore, we include additional experiments and tailored metrics in Appendix C.5 to better isolate and evaluate their effectiveness.
>
> Briefly, when new knowledge is induced, the system retrieves similar nodes in the memory graph and either augments or refines existing knowledge. Each node is associated with a utility score; outdated or less useful nodes are marked as inactive and are preferentially removed when memory constraints arise. We will clarify this process in the main text.
>
> ### Notation inconsistency
> We apologize for the confusion. The transition is captured in our step-level formulation: each tuple e_t = (o_t, s_t, a_t, r_t, g_t), where s_t is derived from (s_{t−1}, a_{t−1}, o_t) (Line 201). The s' in Figure 2 corresponds to the next state s_{t+1}. We will revise the notation for consistency.
>
> We hope these clarifications address your concerns, particularly regarding originality. If so, we would appreciate your reconsideration of the overall assessment. We are happy to provide further clarification if needed.

---

> > ### Author Rebuttal · Reviewer_g3ba · 2026-04-02
> >
> > I appreciate the authors' response. Regarding the performance of the structured module, experiments should be conducted to analyze the distribution of its successful and failed constructions; therefore, I tend to keep my original score.

---

> > > ### Author Response · Authors · 2026-04-08
> > >
> > > We thank you for engaging with our discussion and for articulating the remaining question regarding the distribution of successful and failed constructions of the structuring module. To fully address your question, we report analyses below.
> > >
> > > ### Evaluation Protocol
> > > We categorize extracted knowledge into three types: (i) decision-sufficient, (ii) helpful but imperfect, and (iii) misleading. This categorization is designed to reasonably define "successful and failed constructions," where the **"helpful but imperfect" label is intentionally strict**, as even minor omission of task-relevant details is classified into this category. For LongMemEval and HotpotQA, we use three strong LLM evaluators (GPT-5.1, DeepSeek-V3, Sonnet-3.7) across all instances, with manual verification on 30 samples and all failed cases. For WebArena, we conduct full manual evaluation, because current VLMs are not sufficiently reliable for such challenging evaluation.
> > >
> > > ### (1) Structuring is near-perfect on knowledge-centric tasks
> > > We find that on HotpotQA, **misleading cases (i.e., hallucinations or factual errors) are <2%**, while **imperfect cases are <5%**, under both LLM-based evaluation and manual verification. This is because the knowledge units on HotpotQA are fine-grained factual statements (e.g., entity-level relations or atomic propositions), making extraction relatively straightforward for modern LLMs. Issues only arise in rare cases where the source spans are long and require multi-step reasoning, which may cause hallucination. As a result, the vast majority of extracted knowledge is correct.
> > >
> > > Importantly, **we observe 0 cases where extraction errors lead to incorrect final answers**, as verified through manual inspection. Instead, downstream failures are primarily due to retrieval challenges inherent to the benchmark design. This indicates that, in this setting, the structuring module is highly reliable and does not constitute a performance bottleneck.
> > >
> > > ### (2) Even strictly labeled imperfect extractions are strongly recoverable
> > > Under our **strict evaluation protocol**, where the "helpful but imperfect" label is applied conservatively: **even if most extracted knowledge is correct, missing a single query-relevant element leads to classification as imperfect**, we observe that on LongMemEval, **misleading cases are <1%**, and **<14% of cases are labeled as imperfect**.
> > >
> > > Importantly, this 14% largely reflects **strict completeness requirements**. In practice, we find that **~65% of imperfect cases are recovered through the retrieval aggregation fallback**, meaning that the missing information does not largely affect the final outcome. Furthermore, **all factual errors are corrected by the fallback mechanism**.
> > >
> > > Overall, this analysis shows that while strict evaluation surfaces incompleteness in extraction, the system design ensures that such cases are **effectively recoverable and do not degrade decision performance**.
> > >
> > > ### (3) Robustness under challenging procedural tasks
> > > We find that on WebArena, **misleading cases are <15%**, which mainly arise from two sources: (i) errors in summarizing trajectories, and (ii) incorrect judgments of the success state of a run. Both are well-known challenges in web agent domain and reflect **foundation models' limitations** rather than our system design failure.
> > >
> > > Importantly, even when misleading memory occurs, the system remains recoverable. Through retrieval aggregation fallback, the agent can compare multiple retrieved experiences and avoid being misled by any single incorrect memory. For example, in the GitLab subset, we observe that **even for tasks with misleading memory, the system achieves ~46% success rate**, with its improvements over the no-memory baseline (~39%) supported by the fallback mechanism.
> > >
> > > Finally, our inspection indicates that on WebArena, the bottleneck lies in the **base agent’s ability to execute tasks successfully**, rather than in the structuring module. Specifically, for most failed cases, the structuring operates correctly, but the performance is bounded by the base agent’s incapability to successfully finish the task and provide useful experience to guide later runs.
> > >
> > > These results consistently show that the structuring module is reliable in practice, and importantly, **robust to imperfect extraction through effective recovery mechanisms**. We will include the result tables and case studies in the revision to substantiate these findings.
> > >
> > > We appreciate your follow-up to allow us to improve our paper by including the full analysis of the distribution of success and failure cases. We believe that we have now fully addressed your remaining question. We respectfully ask for a re-evaluation of our work accordingly.
> > >
> > > ---
> > > ### Result Table
> > > | | Decision-Sufficient | Helpful but Imperfect | Misleading | Imperfect & Misleading Recovery Rate |
> > > |-|-:|-:|-:|-:|
> > > |HotpotQA|94.5%|4.4%|1.1%|100%|
> > > |LongMemEval|86%|13.6%|0.6%|64.7%|
> > > |WebArena|85.87%| - |14.13%|23.4%|

---

### Official Review · Reviewer_qunr · 2026-03-12

**Soundness:** 2
**Presentation:** 3
**Significance:** 3
**Originality:** 3
**Overall Recommendation:** 3
**Confidence:** 5

**Summary:**

The paper proposes PLUGMEM, a plug-and-play external memory module for LLM agents, motivated by Tulving's cognitive taxonomy of episodic, semantic, and procedural memory. The system comprises three components: (i) a structuring module that transforms raw interaction traces into standardized episodic tuples and extracts propositional/prescriptive knowledge organized as a multi-layer memory graph; (ii) a retrieval module implementing abstraction-specificity interleaved multi-hop retrieval; and (iii) a reasoning module that compresses retrieved memory into task-adaptive guidance. Alongside the system, the authors propose a novel information-theoretic evaluation framework based on pointwise mutual information (PMI), measuring decision-relevant information gain per memory token. PLUGMEM is evaluated on three heterogeneous benchmarks: LongMemEval, HotpotQA, and WebArena, showing consistent improvements over both task-agnostic and task-specific baselines.

**Compliance With Llm Reviewing Policy:**

Affirmed.

**Key Questions For Authors:**

Q1. Table 4 shows "No Reasoning" outperforms full PLUGMEM on LongMemEval accuracy (78.0 vs 77.2). Can the authors provide a case-level analysis explaining when the reasoning module hurts accuracy? Is this consistent across runs?
Q2. How are P_base(a* | s) and P_mem(a* | s, m) concretely computed for WebArena, where actions are free-form UI operations? What approximation is used, and how sensitive is the ρ metric to this approximation?
Q3. The total number of LLM calls per task step (across structuring, retrieval, and reasoning) appears substantial. What is the average inference cost (in API calls or FLOPs) per agent decision compared to baselines such as Vanilla Retrieval or A-Mem?
Q4. In WebArena Multi-site tasks, PLUGMEM achieves only 20.0/21.6 SR, compared to the 20.0/15.8 vanilla baseline. The online performance is identical. Does this suggest that PLUGMEM's procedural memory does not generalize across website domains, potentially contradicting the cross-task generalization claim?
Q5.  Can the authors clarify whether the baseline results are taken from the original preprints or independently reproduced? If reproduced, were the same hyperparameters and evaluation protocols used? If taken directly from preprints, how do the authors ensure comparability given that these works have not been peer-reviewed and their experimental setups may differ?

**Limitations:**

The Impact Statement briefly touches on privacy risks and bias inheritance, but falls short in two key areas. First, no methodological limitations are discussed in the main text , notably, the reasoning module's accuracy regression (Table 4: 78.0 → 77.2) is left unacknowledged, and the system's cascading dependence on LLM-based extraction quality is never analyzed as a potential failure mode. Second, the privacy and bias concerns are raised in a single sentence each without any concrete mitigations proposed. Given that PLUGMEM is explicitly designed to accumulate and abstract personal interaction histories, a more substantive treatment of selective forgetting and bias propagation through the memory graph is warranted. The authors are encouraged to add a brief dedicated Limitations paragraph to the main paper.

**Strengths And Weaknesses:**

* Strength

1. The core tension between task-specific effectiveness and task-agnostic transferability in agent memory is clearly articulated.Evaluating a single unchanged module across three qualitatively distinct benchmarks (conversational QA, multi-hop retrieval, interactive web navigation) is a  commendable empirical design choice.
2. The decomposition into propositional (semantic) and prescriptive (procedural) knowledge aligns with well-established cognitive theories. Unlike prior systems that invoke cognitive science loosely, the authors provide an explicit structural mapping (Table 2) between memory type, representational unit, and graph architecture, which is a principled design choice rather than pure analogy. Ablations across all three benchmarks with three component variants (No Structuring, No Retrieval, No Reasoning) provide meaningful insight into component roles.
3. The conclusion that retrieval determines operability, structuring determines retrievability, and reasoning determines efficiency is well-supported. The online/offline split on WebArena to test cold-start mitigation and procedural knowledge reuse is an interesting and somewhat novel evaluation design for this domain.

* Weakness

1. The paper lacks reporting on the computational cost of generating the memory. A complete efficiency analysis must include the system overhead of the structuring and retrieval processes, not just the output token length. It remains unclear how PLUGMEM's wall-clock latency and API call costs compare to baseline models.
2. PLUGMEM's structuring and reasoning modules run Qwen2.5-32B in addition to GPT-4o, meaning the total inference capacity deployed is potentially substantially higher than for any baseline. In the experiments shown in Table 3 (LongMemEval), PLUGMEM uses Qwen2.5-32B and GPT-4o jointly, while all baselines (including A-Mem and Vanilla Retrieval) use GPT-4o alone.
3. PMI Metric Operationalization Is Underspecified for Complex Action Spaces
The information density metric is theoretically well-motivated, but its practical computation for WebArena (where actions are free-form, combinatorial UI interactions) is not clearly explained. How is P_base(a* | s) estimated when the action space is not a fixed vocabulary? For WebArena's success rate metric, the "optimal action" a* is a task-level binary outcome , how does this map to the step-level PMI formulation?
4. In Table 4 (LongMemEval ablation), the "No Reasoning" variant achieves 78.0 accuracy, outperforming full PLUGMEM at 77.2, while consuming 38× more tokens (9187.5 vs 241.9). This finding directly contradicts the paper's claim that reasoning improves performance; at best, it suggests reasoning only improves efficiency. The authors briefly note that "reasoning determines efficiency" but do not address why removing it yields higher accuracy.
5. A notable concern is that the majority of cited works are unpublished arXiv preprints, including several key baselines and related systems the paper positions itself against (e.g., A-Mem, Zep, HippoRAG2, AWM, ReasoningBank, LiCoMemory, MemoS, MEMP). Only a small number of references appear in peer-reviewed venues (e.g., HotpotQA at EMNLP 2018, RAG at NeurIPS 2020, PropRAG at EMNLP 2025). This undermines both the related work positioning and the experimental comparisons. Since the methodology and results of these baselines have not been independently validated, the fairness of the comparison is difficult to assess.

---

> ### Author Rebuttal · Authors · 2026-03-27
>
> We appreciate your valuable and detailed feedback. We are encouraged that many strengths, including the novelty, principled memory design grounded in cognitive theory, information-theoretic evaluation framework, and strong cross-benchmark evaluation are recognized. From the review and scores, we understand that the primary concern relates to soundness, particularly regarding (i) the role of the reasoning module, (ii) the operationalization of the PMI metric, and (iii) computational cost and fairness. We address these points with additional analysis and clarifications as follows.
>
> ### Q1: Role of the reasoning module and accuracy trade-off
> We agree that there exist cases on LongMemEval where removing the reasoning module slightly improves accuracy. Our analysis shows that this is due to the compressive nature of the reasoning module (paper line 382-386), which significantly improves memory efficiency, but may omit fine-grained details required in tasks that demand strict information fidelity. Due to space constraints, we will provide a detailed case study in the final version.
>
> Importantly, **this reflects a controllable efficiency–accuracy trade-off rather than a fundamental limitation**. The reasoning module reduces memory tokens by 38×, substantially lowering context usage. Moreover, **its behavior can be adjusted via prompting for reduced compression when high fidelity is required**. **We intentionally avoid heavy task-specific prompt-tuning to preserve a task-agnostic design, which may expose this trade-off on certain subsets of LongMemEval** (e.g., compression-induced temporal misalignment, where a payment made within the queried window is excluded due to confusion between payment date and event date). In contrast, on tasks such as HotpotQA and WebArena, we do not observe negative impact, as these tasks rely less on **exhaustive** detail retention. We will revise the paper to clarify this trade-off.
>
> ### Q2: PMI metric in WebArena
> We clarify that WebArena actions follow a structured schema (e.g., `click [{element_id}]`) rather than being fully free-form, and can be deterministically executed. Due to the interactive setting and inconsistent step scopes, we compute PMI at the trajectory level. Specifically, **P_{base} and P_{mem} are approximated as the agent's task success rates without and with memory**. This formulation measures information gain in terms of task completion probability. We find this formulation stable and aligned with benchmark evaluation, and will clarify the PMI computation in the paper.
>
> ### Q3: Computational cost and fairness
> Thank you for raising this important question. We clarify that all baselines are evaluated under equal or stronger settings using GPT-4o throughout. PlugMem improves efficiency via **adaptive LLM routing**, using smaller models (Qwen-32B) for simpler queries and stronger models only when needed.
>
> Using token usage as a proxy for cost, we observe: (i) under all-open-source-model settings, PlugMem has comparable token usage to baselines; (ii) with routing, PlugMem uses fewer closed-model tokens; and (iii) on simpler benchmarks, routing reduces closed-model usage by 2–3 orders of magnitude. Due to space constraints, we refer you to our response to Reviewer 8ueg for the full cost analysis tables across all benchmarks. These results indicate that **PlugMem does not incur higher effective cost and is often more efficient**. We will include this analysis in the main paper.
>
> ### Q4: Generalization in WebArena
> We believe there is a misunderstanding regarding the roles of the online and offline settings. The online setting evaluates **cold-start behavior**, where **all methods begin without prior memory** and thus achieve similar performance (20.0 vs. 20.0 SR), which is expected. The offline setting evaluates cross-task generalization by fixing the learned memory graph and testing knowledge reuse on new task instances. Under this protocol, PlugMem improves success rate from 15.8 to 21.6, demonstrating effective transfer of accumulated procedural knowledge. We will clarify this distinction in the paper.
>
> ### Q5: Baselines and reproducibility
> We clarify that many baselines are accepted or widely adopted (e.g., A-mem by NeurIPS 2025, HippoRAG2 by ICML 2025), while recent studies are included to reflect current progress. We distinguish between reported (“*”) vs. reproduced results. For reproduced baselines, we fully follow original protocols, ensuring a fair comparison. We will make this distinction more explicit.
>
> ### Additional clarifications and limitations
> We will strengthen the discussion of module roles and expand limitations to include the efficiency–accuracy trade-off and privacy and bias risks with mitigation discussion.
>
> We hope these clarifications address the concerns, and would greatly appreciate your reconsideration of the current assessment based on our responses. We also welcome further discussion and are happy to provide additional analysis if helpful.

---

> > ### Author Rebuttal · Reviewer_qunr · 2026-04-08
> >
> > The rebuttal improves clarity and partially strengthens the empirical support, but it does not fully resolve concerns about the uniqueness of the structural contribution and the sufficiency of comparisons with closely related abstraction-based memory methods. Overall, I will maintain my score.

---

> > > ### Author Response · Authors · 2026-04-08
> > >
> > > We thank you for the follow-up:
> > >
> > > > The rebuttal improves clarity and partially strengthens the empirical support, but it does not fully resolve concerns about the uniqueness of the structural contribution and the sufficiency of comparisons with closely related abstraction-based memory methods. Overall, I will maintain my score.
> > >
> > > We would like to note that **these questions were not raised in the initial review, but was introduced in the second-round comment posted less than four hours before the rebuttal deadline**. Fortunately, we find that these questions have **substantial overlap** with those raised by other reviewers, and have therefore can be **fully addressed** by our responses below.
> > >
> > > ---
> > >
> > > **(1) On “the uniqueness of the structural contribution”**
> > >
> > > We understand this as a question about whether the structuring module provides **meaningful and reliable gains** with its unique design beyond existing approaches.
> > >
> > > As addressed in our response to **Reviewer g3ba**, we conducted a detailed analysis of the **distribution of successful vs. failed constructions**. The key findings are:
> > >
> > > * On knowledge-centric benchmarks (HotpotQA, LongMemEval), **misleading cases are extremely rare (<1–2%)**, and most “imperfect” cases arise from **strict completeness criteria rather than incorrect extraction**.
> > > * Importantly, these imperfect cases are **largely recoverable via retrieval aggregation**, and **do not lead to incorrect final decisions** in practice.
> > > * On more challenging procedural tasks (WebArena), while errors increase due to foundation model limitations, the system remains **robust through fallback mechanisms**, and performance is not bottlenecked by the structuring module.
> > >
> > > Overall, these results show that our structuring module with two distinctive type of knowledge extractions (i.e., propositional knowledge extraction and prescriptive knowledge extraction, which has never been proposed and combined organically in any existing work), hierarchical orgnization, and fallback mechanism is **both reliable and practically effective**, allowing a single task-agnostic memory module to perform well across heterogenous benchmarks, and thus supporting its contribution beyond existing memory baselines.
> > >
> > > ---
> > >
> > > **(2) On “the sufficiency of comparisons with closely related abstraction-based memory methods”**
> > >
> > > We understand this as a request for stronger comparison with **high-level abstracted memory approaches** and clearer evidence of advantage.
> > >
> > > As detailed in our response to **Reviewer RFWh**, we include comparisons with:
> > >
> > > * **A-Mem** (flat summarization-based abstraction), and
> > > * **RAPTOR** (hierarchical abstraction),
> > >
> > > covering two major paradigms of abstraction-based memory.
> > >
> > > The key takeaway is:
> > >
> > > > The performance difference is not due to abstraction alone, but to how memory is structured and retrieved.
> > >
> > > Through multiple case studies, we show that abstraction-based methods tend to **omit or dilute task-critical details**, while our approach **preserves fine-grained, query-relevant information** via structured knowledge extraction and representation, leading to improved downstream performance.
> > >
> > > ---
> > >
> > > Given the strong overlap between the concerns raised here and those from other reviewers, and due to space limitations, we **kindly refer you to our detailed responses** to those reviewers for the full experimental results and case studies.
> > >
> > > We believe these additions address the questions raised. We respectfully request you to reconsider your evaluation in light of the additional evidence.

---

### Official Review · Reviewer_8ueg · 2026-03-13

**Soundness:** 2
**Presentation:** 2
**Significance:** 2
**Originality:** 3
**Overall Recommendation:** 5
**Confidence:** 3

**Summary:**

This paper presents PLUGMEM, a task-agnostic plugin memory module for LLM agents. The idea draws from cognitive science, that decision-relevant information is best represented as abstract knowledge rather than raw episodic traces. PLUGMEM takes raw agent interactions and transforms them into a structured memory graph. At inference time, a retrieval module searches over the graph, and a reasoning module compresses retrieved content into actionable guidance. The system is evaluated across three benchmarks (LongMemEval, HotpotQA, WebArena) to show consistently outperforming of both task-agnostic and task-specific baselines. The authors also introduce a memory information density metric and demonstrates PLUGMEM's high information density memory compared to other baselines.

**Compliance With Llm Reviewing Policy:**

Affirmed.

**Final Justification:**

The authors have addressed the concerns in my review and I have increased my score as a result.

**Key Questions For Authors:**

1. There is some model asymmetry in the results with Plugmem using Qwen32B + GPT-4o. What is the reasoning for this and what is the result if PlugMem uses only a single model, or if other baselines use two models?

2. Multiple LLM calls are required for the framework in structuring, retrieval, and reasoning. What is the impact of these calls in terms of token cost and inference time, compared to other baseline methods? Is there a trade-off cost for improved performance?

3. What does the negative information density mean for some baselines? How does negative information density arise, what are the implications, and how can memory frameworks avoid this negative information density?

**Limitations:**

Yes.

**Strengths And Weaknesses:**

# Soundness

**Strengths**
The paper is generally technically sound. The three components of the architecture (structuring, retrieval, reasoning) are all clearly motivated and well explained. There is good coverage of other task-specific and task-agnostic memory system. A number of performance metrics and average tokens are presented in the results.

**Weaknesses**
- Some baselines on WebArena get negative information density, this should be further analysed and explained what it means for the baselines to get negative information density.
- Some baselines only use a single model while PlugMem sometimes uses muliple (Qwen32b + GPT-4o). This should be explained why it still makes for a fair comparison, and input and output token costs should be reported not just the final tokens used for memory.
- Some other memory-based papers such as "G-Memory: Tracing Hierarchical Memory for Multi-Agent Systems" show a large number of memory frameworks for comparison. These memory frameworks should also be acknowledged and potentially compared against.
- Statistical significance of results should be reported as some results show similar or close performance that could be due to variance.

# Presentation

**Strengths**
Paper is generally well-organised, with a strong narrative from motivation to the architecture and experiments. Table 1 shows a good comparison and taxonomy of other memory systems.

**Weaknesses**
- Appendix C.4 is missing
- Results tables have some clarity issues. Some tables are quite small. Consider bold for the headers and removing the background for sub-category headings.
- Figures 2 and 3 are densely packed with information and examples, making it difficult to follow. Consider making the figures bigger with more detailed descriptions in the figure captions.

# Significance

**Strengths**
The problem of task-agnostic memory is a useful problem to solve, though there are other existing frameworks addressing the same problem. The information density metric is a novel and interesting approach that could be very practical in evaluation of memory-based agents.

**Weaknesses**
The paper's claim that task-agnostic memory can consistently outperform task-specific designs is a strong one; it holds in this paper's experimental regime, but the domains chosen (conversational QA, Wikipedia-based retrieval, web navigation) share structural similarities in the types of knowledge they demand, which may limit how broadly this conclusion generalizes.

# Originality
The information theory used to build the density metric is novel. The use of knowledge units is interesting and different from typical graph-based knowledge memory, representing a novel contribution.

---

> ### Author Rebuttal · Authors · 2026-03-27
>
> We thank you for the constructive feedback. We are encouraged that you find the paper technically sound, well-motivated, and recognize the novelty of the information density metric. Below we address the main concerns.
>
> ---
>
> ## Response to Key Questions
>
> ### (Q1–Q2) Model asymmetry & cost
>
> We thank you for raising this important concern. Our use of multiple models is motivated by cost–utility trade-offs rather than performance gains. We observe that many sub-modules (e.g., structuring, intermediate reasoning) are lightweight, and using a frontier model for all steps would be over-provisioned and inefficient.
>
> To address this, PlugMem adopts **an adaptive LLM routing mechanism**, **dynamically assigning models based on task complexity**. This avoids unnecessary use of strong models while maintaining performance.
>
> Regarding fairness, when PlugMem uses multiple models (e.g., a combination of a smaller open-source model Qwen-32B and a stronger closed-source model GPT-4o), **we evaluate baselines using the stronger model only** (e.g., GPT-4o, which consistently performs better across reasoning and factual benchmarks than Qwen-32B). This ensures that baselines are evaluated under strong and competitive settings, rather than being disadvantaged by weaker models.
>
> We further conducted cost analysis across benchmarks and find:
> (i) In all-open-source-model settings, PlugMem has comparable token usage to baselines;
> (ii) With adaptive routing, PlugMem consistently uses the least closed-model tokens;
> (iii) On simpler benchmarks, routing reduces closed-model usage by 2–3 orders of magnitude.
>
> We report cost analysis results below, evaluated on randomly sampled subsets of each benchmark. The subsets are sufficiently large to ensure reliable estimates.
> `*` indicates results taken from previous work
>
> **LongMemEval**:
> | Method | Q32_{in/out} | GPT-4o_{in/out} | Latency |
> |-----|-----|-----|-----|
> | Zep | - | 2545/1189 | `*` 2.58s|
> | LiCoMemory | - | 585/217 | `*` 1.52s |
> | PlugMem | 1604/418 | 9/0.3 | 0.66s + 2 * Q32 calls + 1 * gpt-4o call |
>
> **HotpotQA**:
> | Method | Q32_{in/out} | Latency |
> |-----|-----|-----|
> | A-Mem | 1778/261 | 1.19s |
> | HippoRAG2 | 1350/490 | 2.32s |
> | PlugMem | 1421/195 |  1.31s + 8 * Q32 |
>
> **WebArena**:
> | Method | Q32_{in/out} | GPT-4o_{in/out} | Latency |
> |-----|-----|-----|-----|
> | A-Mem | - | 10397/1821 | 0.91s |
> | PlugMem | 12203/1312 | 4424/412 | 0.71s + 2 * Q32 + 1 * gpt-4o |
>
> We will include detailed results in the final version.
>
> ### (Q3) Negative information density
>
> Information density is defined as information gain / token count (denominator > 0).
> A negative value indicates that the memory introduces noisy or misleading tokens, leading to worse decisions than without memory. Thus, negative density reflects a failure mode where memory harms performance due to noise. We provide a detailed interpretation in L1157–1161 and will further clarify this in the revision.
>
> ---
>
> ## Response to Weaknesses
>
> ### Related work
> We thank you for the suggestion. Our evaluation focuses on recent and competitive baselines, while several methods included in G-Memory are earlier systems that have been surpassed on overlapping benchmarks. Specifically, on HotpotQA, MemoryBank (the strongest non-G-Memory baseline in G-Memory) has already been compared with A-Mem and shows weaker performance. Given that PlugMem substantially outperforms A-Mem in our experiments, this transitive comparison provides a strong indication of its advantage over several memory frameworks considered in G-Memory.
>
> We will clarify this connection in the revision and expand the discussion of G-Memory and its baselines. Where feasible, we will also include additional direct comparisons.
>
> ### Statistical significance
> Thank you for the suggestion. For all the major comparisons, we will run three independent trials for both our method and the baselines, and report the mean and standard deviation. For this rebuttal, we conducted three runs on HotpotQA for the closest baseline. Our method consistently outperforms it, with improvements remaining within the margin of variance. We will add such statistical significance test results for all the major comparisons.
> | Method | EM | F1-Score |
> |---|---|---|
> | HippoRAG2 | 59.533 (±0.125) | 72.880 (±0.157) |
> | PlugMem | 61.233 (±0.123)  | 74.190 (±0.137) |
>
> ### Generalization claim
> While benchmarks appear related, they require different knowledge types: factual (QA), long-context knowledge reasoning (retrieval), and procedural skills (WebArena). Existing methods typically focus on one type; **we are not aware of prior work performing well across both**. We will clarify and moderate the claim.
>
> ### Presentation
> We will add Appendix C.4 and improve table/figure formatting for clarity.
>
> We thank you again for the helpful feedback. We will incorporate the suggested improvements in the final version to further strengthen clarity and evaluation, and hope these clarifications address your main concerns.

---

> > ### Author Rebuttal · Reviewer_8ueg · 2026-04-03
> >
> > Thank you for the response, my questions and concerns have been well addressed.

---

> > > ### Author Response · Authors · 2026-04-08
> > >
> > > We sincerely appreciate your positive feedback and are very glad to hear that all your questions and concerns have been well addressed! Your suggestions have been very helpful, and we will incorporate the corresponding clarifications and improvements into the final revision to further strengthen the paper.
> > >
> > > We also note that the acknowledgement option you selected indicates that, when concerns are fully resolved, this may warrant reconsidering the score. Of course, we fully respect that this is entirely at your discretion, but we would be grateful if you would consider whether an updated score would be appropriate. Thank you!

---

### Decision · Program_Chairs · 2026-04-30

**Decision:**

Accept (regular)

**Comment:**

Overall, the submission presents a well-motivated and technically sound approach to task-agnostic memory for LLM agents, with a clear architectural design grounded in cognitive principles and a novel information-theoretic evaluation. Multiple reviewers acknowledge the strength of the formulation, empirical consistency across heterogeneous benchmarks, and practical significance of improving memory efficiency and transferability. While concerns remain regarding computational cost transparency, fairness of model usage, and the novelty relative to prior abstraction-based memory frameworks, the authors provide largely convincing rebuttals, clarifying the efficiency–accuracy trade-off, detailing cost analysis, and strengthening empirical support. Some issues such as incomplete comparisons and limited analysis of failure modes slightly weaken the overall contribution, but they do not fundamentally undermine the validity or usefulness of the proposed method. Considering the balance of strengths, the cross-task generalization evidence, and the potential impact on agent memory design, I recommend acceptance with the expectation that the final version incorporates the clarified analyses and improves positioning against closely related work.